# Incremental Transformer Neural Processes

**Philip Mortimer** [* 1]  **Cristiana Diaconu** [* 1]  **Tommy Rochussen** [2 3 4]  **Bruno Mlodozeniec** [1]  **Richard E. Turner** [1 5]

## Abstract

Neural Processes (NPs), and specifically Transformer Neural Processes (TNPs), have demonstrated remarkable performance across tasks ranging from spatiotemporal forecasting to tabular data modelling. However, many of these applications are inherently sequential, involving continuous data streams such as real-time sensor readings or database updates. In such settings, models should support cheap, incremental updates rather than recomputing internal representations from scratch for every new observation—a capability existing TNP variants lack. Drawing inspiration from Large Language Models, we introduce the Incremental TNP (`incTNP`). By leveraging causal masking, Key-Value (KV) caching, and a data-efficient autoregressive training strategy, `incTNP` matches the predictive performance of standard TNPs while reducing the computational cost of updates from quadratic to linear time complexity. We empirically evaluate our model on a range of synthetic and real-world tasks, including tabular regression and temperature prediction. Our results show that, surprisingly, `incTNP` delivers performance comparable to—or better than—non-causal TNPs while unlocking orders-of-magnitude speedups for sequential inference. Finally, we assess the consistency of the model's updates—by adapting a metric of "implicit Bayesianness", we show that under a one-at-a-time streaming protocol, `incTNP` retains a prediction rule as implicitly Bayesian as standard non-causal TNPs, demonstrating that `incTNP` achieves the computational benefits of causal masking without sacrificing the consistency required for streaming inference.[1]

[*]Equal contribution  [1]University of Cambridge [2]Helmholtz AI [3]Technical University of Munich [4]Munich Center for Machine Learning [5]Alan Turing Institute. Correspondence to: Philip Mortimer <pm846@cam.ac.uk>, Cristiana Diaconu <cdd43@cam.ac.uk>.

*Proceedings of the $43^{rd}$ International Conference on Machine Learning*, Seoul, South Korea. PMLR 306, 2026. Copyright 2026 by the author(s).

[1]Our code: https://github.com/philipmortimer/incTNP-code.

## 1. Introduction

The dominant paradigm in machine learning involves training models on static datasets collected up to a fixed point in time. While effective in offline settings, this approach faces significant challenges in real-world streaming scenarios where data arrive sequentially and continuously. In such cases, a model should be able to quickly integrate new observations to improve predictions. This necessitates architectures that support **cheap, incremental** updates, allowing for immediate adaptation without the computational overhead of reprocessing the entire history.

Amongst models that output predictions conditioned on past data (the context set), approaches such as Neural Processes (NP; Garnelo et al. (2018a;b); Nguyen & Grover (2022)) and the closely-related Prior-Fitted Networks (PFNs; Müller et al. (2022)) have shown great promise. They have been successfully applied to tasks such as weather modelling (Andersson et al., 2023; Allen et al., 2025) and tabular data prediction (Hollmann et al., 2025). Crucially, these domains are inherently suited for streaming workflows, for example integrating real-time measurements from weather sensors or handling continuous updates in active databases. However, existing NP and PFN implementations are not well-suited to handle such live data streams efficiently, as they lack the ability to perform cheap incremental updates. The most popular variants rely on the transformer architecture (Vaswani et al., 2017), which, when applied naïvely, incurs a quadratic computational cost for every new observation, making high-frequency updates prohibitively expensive.

In this work, we address this computational bottleneck by leveraging advancements from causally masked Large Language Models (LLMs) to introduce an incrementally-updateable TNP, the `incTNP`. By incorporating causal masking into the self-attention mechanism and utilising Key-Value (KV) caching, `incTNP` enables efficient incremental context updates with linear time complexity. We also introduce a dense autoregressive training strategy (referred to as `incTNP-Seq`) that amortises learning across all context sizes in a single forward pass. Our experiments show that when trained with this data-efficient scheme, `incTNP` often outperforms standard TNP models trained on a compute-matched budget, while unlocking the ability to perform high-frequency updates in real-time streaming scenarios.

This finding is interesting, indicating that causal masking has surprisingly little effect on the quality of the predictions compared to the full bi-directional attention used in standard TNPs.

We further investigate the extent to which causal masking compromises permutation invariance with respect to the context set, a key property of NPs. We adapt the metric of *implicit Bayesianness* proposed in Mlodozeniec et al. (2024), applying it for the first time to NPs in order to quantify this effect. Comparing the prediction rules of `incTNP` against the standard TNP (`TNP-D`; Nguyen & Grover 2022) under a one-at-a-time streaming protocol, our analysis reveals that, despite its causal structure, `incTNP` exhibits a degree of implicit Bayesianness comparable to the permutation-invariant baseline, confirming that `incTNP` secures the computational benefits of causal masking without compromising probabilistic consistency.

We validate our approach across both factorised and autoregressive (AR) deployment modes (see Figure 1). The latter is particularly relevant when we expect correlations between target points, yielding state-of-the-art performance (Bruinsma et al., 2023). However, AR prediction creates a computational bottleneck: standard TNPs must re-process the entire context history for every generated target. In a streaming setting, where this expensive inference loop repeats at every update step, standard TNPs become prohibitively slow. We demonstrate that `incTNP` overcomes this barrier, rendering high-fidelity AR inference feasible even in streaming scenarios. We summarise our key contributions:

1. We propose the Incremental TNP (`incTNP`), a Transformer NP variant that leverages causal masking and KV caching to enable linear-time context updates, unlocking orders-of-magnitude speedups for streaming predictions tasks, in both factorised and AR deployment mode.
2. We demonstrate on both synthetic benchmarks and complex real-world tasks—including environmental temperature prediction and tabular regression—that by employing a dense autoregressive training strategy, `incTNP` matches or exceeds the predictive performance of standard TNPs under equal training budgets.
3. We investigate the impact of causal masking on predictive consistency using a novel implicit Bayesianness metric. We show empirically that `incTNP` retains a prediction rule comparable in Bayesian consistency to the standard permutation-invariant `TNP-D`, confirming that the model achieves computational scalability without sacrificing probabilistic coherency.

## 2. Background

We consider a supervised meta-learning setting with input and output spaces $\mathcal{X} = \mathbb{R}^{D_x}$ and $\mathcal{Y} = \mathbb{R}^{D_y}$, where the model needs to adapt to new prediction tasks on the target set $\mathcal{D}^t$ based on some observed, context data $\mathcal{D}^c$. Let $|\mathcal{D}^c| = N_c$ and $|\mathcal{D}^t| = N_t$ points, and $\mathbf{X}^c \in \mathcal{X}^{N_c}, \mathbf{Y}^c \in \mathcal{Y}^{N_c}$ and $\mathbf{X}^t \in \mathcal{X}^{N_t}, \mathbf{Y}^t \in \mathcal{Y}^{N_t}$ denote the inputs and outputs for $\mathcal{D}^c$ and $\mathcal{D}^t$. Within the meta-learning setup, a single task is defined as $\xi = (\mathcal{D}^c, \mathcal{D}^t) = ((\mathbf{X}^c, \mathbf{Y}^c), (\mathbf{X}^t, \mathbf{Y}^t))$.

### 2.1. Neural Processes

Neural processes (NPs; Garnelo et al. 2018a;b) define a family of neural-network-based mappings from context sets $\mathcal{D}^c = \{(\mathbf{x}_n^c, \mathbf{y}_n^c)\}_{n=1}^{N_c}$ to predictive distributions at target locations $\mathbf{X}^t$. Viewed through the lens of stochastic process modelling (Garnelo et al., 2018b), NPs are designed to respect consistency under marginalisation and permutation *given a fixed context set*. However, as noted by Xu et al. (2023), this design implies that NP variants typically lack *conditional consistency*: the distribution of a sequence $\mathbf{y}_{1:N_t}^t$ conditioned on an initial context need not match the distribution obtained if the points were predicted sequentially and appended to the context after each iteration (Kim et al., 2019). This distinction is particularly relevant in streaming settings, where the context set is not fixed but grows dynamically over time.

NPs are also tightly linked to Prior-Fitted Networks (PFNs; Müller et al. 2022), which have proven effective in domains such as tabular data prediction (Hollmann et al., 2025). While PFNs share the same meta-learning framework with NPs, they are generally trained on synthetic data and deployed on real-world tasks, necessitating robust sim-to-real capabilities.

**Conditional Neural Processes (CNPs)** Within the NP family, we focus on Conditional NPs (CNPs; Garnelo et al. 2018a), which approximate the predictive distribution by assuming factorisation over target points given the context representation:

$$p(\mathbf{Y}^t|\mathbf{X}^t, \mathcal{D}^c) = \prod_{n=1}^{N_t} p_\theta(\mathbf{y}_n^t|\mathbf{x}_n^t, \mathcal{D}^c) \qquad (1)$$

CNPs are trained in a meta-learning setup by maximising the expected log-likelihood over tasks: $\mathcal{L}_{\mathrm{ML}}(\theta) = \mathbb{E}_{p(\xi)}\left[\sum_{n=1}^{N_t} \log p_\theta(\mathbf{y}_n^t|\mathbf{x}_n^t, \mathcal{D}^c)\right]$. While efficient, the factorisation assumption prevents CNPs from modelling correlations between target points.

**Autoregressive Deployment** To capture dependencies without altering the training objective, CNPs can be deployed in AR mode at test time (Bruinsma et al., 2023)—the model predicts one target point at a time, samples a value, and appends it to the context set for the next prediction:

$$p(\mathbf{Y}^t|\mathbf{X}^t; \mathcal{D}^c) = \prod_{n=1}^{N_t} p_\theta(\mathbf{y}_n^t|\mathbf{x}_n^t, \mathcal{D}^c \cup (\mathbf{x}_j^t, \mathbf{y}_j^t)_{j=1}^{n-1}) \quad (2)$$

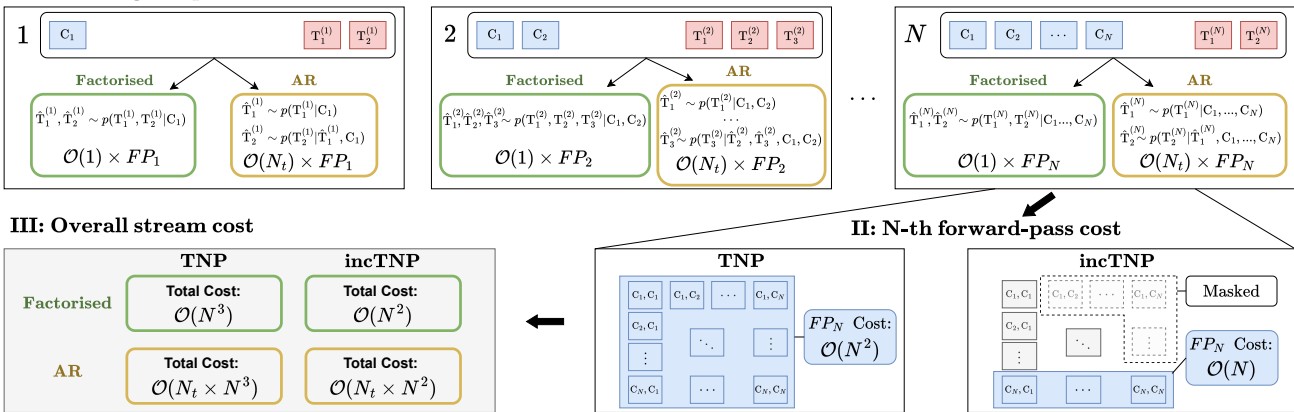

*Figure 1.* Computational complexity analysis in the streaming setting (see theoretical analysis in Appendix F). **Top**: As the data stream grows (steps $1, \ldots N$), the model updates its context $C$ and predicts targets $T$. In AR mode, the model must perform $N_t$ forward passes (FP) at every step, amplifying the computational burden. **Bottom Right**: A comparison of the FP mechanics. Standard `TNP` must re-encode the entire context history at every step (cost $\mathcal{O}(N^2)$), whereas `incTNP` leverages KV caching to process only the incremental update (cost $\mathcal{O}(N)$). **Bottom Left**: The cumulative cost over a stream of length $N$. Because `TNP` recomputes the full attention matrix repeatedly, its total cost scales cubically ($\mathcal{O}(N^3)$). `incTNP` reduces this to quadratic scaling ($\mathcal{O}(N^2)$), making it the only viable option for long streams, particularly when combined with the expensive AR decoding loop.

While empirically strong, this approach breaks consistency under permutation (Bruinsma et al., 2023) and incurs a high computational cost—a full forward pass for each target point—making it prohibitive for real-time streaming.

## 2.2. Transformer Neural Processes (TNPs)

Transformer-based architectures (Nguyen & Grover, 2022; Müller et al., 2024) have emerged as a powerful backbone for NPs due to their natural handling of set-valued inputs. In a standard TNP, context and target inputs are tokenised into representations $\mathbf{Z}_0^c$ and $\mathbf{Z}_0^t$. Each context token corresponds to an observed pair $(\mathbf{x}_i^c, \mathbf{y}_i^c)$, while each target token corresponds to a target input $\mathbf{x}_i^t$ whose output is to be predicted. These context and target tokens are processed via alternating layers of Multi-Head Self-Attention (MHSA) on the context and Multi-Head Cross-Attention (MHCA) from context to targets: $\mathbf{Z}_l^t = \mathrm{MHCA}(\mathbf{Z}_{l-1}^t, \mathrm{MHSA}(\mathbf{Z}_{l-1}^c))$. The Diagonal TNP (`TNP-D`; Nguyen & Grover 2022) factorises the predictive distribution similar to a CNP. However, because the self-attention mechanism $\mathrm{MHSA}(\mathbf{Z}^c)$ scales quadratically with the number of context points and must be recomputed whenever the context changes, `TNP-D` is ill-suited for incremental learning or streaming data. The Autoregressive TNP (`TNP-A`; Nguyen & Grover 2022) models the joint distribution over targets autoregressively, but still incurs the same contextual self-attention bottleneck.

## 2.3. Efficient Incremental Updates via Causal Masking

Outside the NP literature, Large Language Models (LLMs) achieve efficient incremental processing through KV caching and causal masking. KV caching stores the key ($K$)

and value ($V$) matrices of past tokens, allowing each new token to attend to the cached history and reducing per-step inference cost from $\mathcal{O}(N^2)$ to $\mathcal{O}(N)$ (Pope et al., 2022). This requires causal masking: in bidirectional attention, new tokens alter previous representations and invalidate the cache, whereas causal attention keeps past representations static. In this work, we adapt these mechanisms to the NP framework to enable efficient streaming updates.

## 3. Related Work

### 3.1. Architectures for Neural Processes

While early NP architectures relied on efficient aggregation mechanisms like DeepSets (Zaheer et al., 2017; Garnelo et al., 2018a) or convolutions (Gordon et al., 2020; Bruinsma et al., 2023), transformers have become the state-of-the-art for modelling complex relationships in data (Nguyen & Grover, 2022; Ashman et al., 2025). Despite their representational power, standard TNPs suffer from the quadratic cost of attention. Various works have proposed linear-time or sub-quadratic approximations for NPs (Feng et al., 2023; Lee et al., 2019; Ashman et al., 2025), often by summarising the context representation into a reduced set of pseudo-tokens. A prominent example is the Latent Bottlenecked Attention NP (`LBANP`; Feng et al. 2023), which is used as a baseline in this work. However, these methods typically require re-encoding the full context for each update or are designed for static context sets, rendering them inefficient for streaming scenarios where the context grows continuously.

### 3.2. Incremental Updates in Neural Processes

The challenge of efficient autoregression in NPs has recently been addressed by Hassan et al. (2025). Their work introduces a causal autoregressive buffer to enable cheaper AR sampling. However, their motivation differs fundamentally from ours: they focus on correlated target prediction rather than streaming data adaptation. Consequently, their architecture separates a fixed initial context from a dynamic buffer, a design that becomes unnatural in streaming settings where all data arrive sequentially. To handle infinite streams, their buffer would require periodic merging and re-encoding, reintroducing computational bottlenecks. In contrast, our `incTNP` applies causal masking uniformly across the entire stream, enabling $\mathcal{O}(N)$ updates indefinitely without buffers. Furthermore, we provide the first quantitative analysis of how causal masking affects the consistency of the prediction rule implied by the model.

### 3.3. Streaming Prediction and Implicit Bayesianness

In streaming settings, maintaining coherent probabilistic updates is crucial. Gaussian Processes (GPs) (Rasmussen & Williams, 2006) offer a gold standard for coherent Bayesian updates but scale poorly ($\mathcal{O}(N^3)$). While sparse GP methods (Bui et al., 2017; Stanton et al., 2021) allow for online updates, they often sacrifice consistency due to their approximations—such as the structured kernel interpolation approximation in Stanton et al. 2021. Furthermore, they lack the meta-learning capabilities of NPs, often struggle in higher-dimensional settings, and generally incur higher inference costs. Since NPs with dynamic contexts (or causal masking) are not strictly consistent stochastic processes, it is vital to assess how coherent their predictions are. Mlodozeniec et al. (2024) recently proposed a metric of *implicit Bayesianness*, quantifying how closely a prediction rule adheres to exchangeability—a proxy for being implicitly Bayesian (and hence rational in updating posterior beliefs based on new evidence). This is achieved by permuting the data uniformly at random and computing the variance of the log-joint predictive distribution over a set of target variables across these permutations. We extend this analysis by proposing an alternative metric, and apply it for the first time in the context of NPs to assess the impact of the causal masking mechanism on implicit Bayesianness.

### 3.4. Foundation Models for Tabular Data

Our work is also relevant to the growing field of Foundation Models for structured data. Prior-Fitted Networks (PFNs; Müller et al. 2022; Hollmann et al. 2025) and time-series foundation models (Moroshan et al., 2025) share the in-context learning paradigm of NPs. While highly effective on small datasets, these models often struggle with large historical contexts due to the quadratic attention costs.

In real-world applications where data accumulate continuously—such as clinical monitoring or financial forecasting—our efficient incremental architecture offers a path to scaling these foundation models to massive, evolving datasets without discarding historical information.

## 4. Incremental Transformer Neural Processes

In this section, we outline the architecture of the proposed Incremental Transformer Neural Process (`incTNP`). This novel member of the NP family introduces a causal structure to the standard TNP, enabling linear-time context updates via KV caching while preserving the expressivity of softmax-based attention.

### 4.1. Causally Masked Self-Attention

To enable efficient updates without recomputing the entire context, we introduce causal masking into the TNP encoder. By enforcing a lower-triangular causal attention mask $M^{\text{causal}}$ (where token $i$ attends only to tokens $j \leq i$) and utilising KV caching to store past representations, we decouple historical data from future observations. This allows `incTNP` to update the context by processing *only* the newly arrived datapoint. The update at layer $l$ becomes:

$$\mathbf{Z}_l^t = \text{MHCA}\left(\mathbf{Z}_{l-1}^t, \text{M-MHSA}(\mathbf{Z}_{l-1}^c, M^{\text{causal}})\right), \quad (3)$$

where M-MHSA denotes Masked Multi-Head Self-Attention. This mechanism reduces the marginal computational cost of an incremental update from $\mathcal{O}(N_c^2)$ to $\mathcal{O}(N_c)$, enabling scalable streaming inference.

### 4.2. Efficient training strategy

Standard CNPs are trained via a meta-learning objective that approximates the expected predictive log-probability over random tasks: $\mathbb{E}_{\xi \sim p(\xi)}\left[\sum_{n=1}^{N_t} \log p_\theta(\mathbf{y}_n^t | \mathbf{x}_n^t, \mathcal{D}^c)\right]$. In this regime, gradients are computed for a fixed context size $N_c$ per task, potentially leading to high variance and inefficient sample usage. Inspired by LLM training, we propose a dense autoregressive training strategy, applied in the model termed `incTNP-Seq`. Instead of partitioning data into disjoint context and target sets, we treat the data as a single sequence $\mathcal{D}^{\text{seq}} = [(\mathbf{x}_1, \mathbf{y}_1), \ldots, (\mathbf{x}_N, \mathbf{y}_N)]$. We construct two parallel streams for the transformer differentiated through a binary flag: a "context" stream $\mathbf{Z}^{c,\text{seq}}$ and a "target" stream $\mathbf{Z}^{t,\text{seq}}$. This change requires application of causal masking not only in the self-attention blocks (as in Equation 3) but also in the cross-attention blocks (MHCA). The target representation update at layer $l$ becomes:

$$\mathbf{Z}_l^t = \text{M-MHCA}\left(\mathbf{Z}_{l-1}^t, \text{M-MHSA}(\mathbf{Z}_{l-1}^c, M^{\text{causal}}), M^{\text{causal}}\right).$$
$$(4)$$

By masking the MHCA, we enforce that the prediction for the $n$-th target $\mathbf{y}_n$ relies solely on the context history

$(\mathbf{x}_{1:n}, \mathbf{y}_{1:n})$, effectively computing the loss for *every possible prefix context* simultaneously in a single forward pass. This dense supervision signal amortises the cost of training across all context sizes, improving data efficiency and generalisation. Such a training paradigm is only possible under `incTNP`'s causal masking structures; attempting to train a standard `TNP-D` via this approach would fail since each target token could 'cheat' by simply attending to its corresponding context token one step later in the stream.

## 4.3. Test-time Computational Complexity

**Runtime Cost** We demonstrate the efficiency gains of the `incTNP` family by analysing a streaming scenario with a growing context of instantaneous size $N_s$ and $N_t$ target locations. In the factorised deployment, existing TNPs must re-encode the full history at every step, incurring an $\mathcal{O}(N_s^2)$ cost per update. In contrast, `incTNP` processes only the new token, reducing update latency to $\mathcal{O}(N_s)$. These gains are magnified in AR deployment, where predictions are generated sequentially over the targets to capture dependencies. Whilst this regime is prohibitively expensive for standard models—which scale as $\mathcal{O}(N_t \cdot N_s^2)$ per update—`incTNP` leverages KV caching to reduce to an $\mathcal{O}(N_t \cdot N_s)$ cost[2]. This linear scaling makes high-fidelity autoregressive inference computationally feasible in real-time streaming applications with large histories.

**KV Cache Memory Footprint** Caching the streamed key-value history incurs a persistent memory cost of $\mathcal{O}(LD_zN_s)$ for factorised predictions, where $L$ is the number of transformer layers and $D_z$ is the embedding dimension. AR prediction has a peak KV cache memory cost of $\mathcal{O}(SLD_z(N_s + N_t))$ over $S$ sample unrolls. When accounting for peak transient activation memory to evaluate the overall peak memory footprint under standard attention kernels, `incTNP` scales linearly with respect to the context stream size whilst `TNP-D` scales quadratically due to its full self-attention computation over the context. However, `TNP-D` can also obtain linear peak-memory scaling through the use of memory-efficient attention kernels. Practically, `incTNP`'s KV caching may present a memory bottleneck at extreme scale, although we encountered no issues in our experiments. We provide an expanded analysis in Appendix G.

## 4.4. Theoretical trade-offs

While offering significant computational gains, the `incTNP` architecture sacrifices a core property of standard Neural Processes: permutation invariance with respect to the context set. Unlike `TNP-D`, `incTNP` defines a distribution that is sensitive to the order of context set. However, we argue that in settings where the context is not fixed (e.g.,

---

streaming data or autoregressive prediction)—and where the *conditional* inconsistency inherent to `TNP-D` and all other CNPs becomes particularly noticeable— strictly enforcing invariance with respect to the context constrains model design without necessarily guaranteeing a more "rational" prediction rule. To quantify the extent and impact of this violation, we adopt a metric of *implicit Bayesianness*.

**Theoretical Framework** Defining $\mathbf{z}_i := (\mathbf{x}_i, \mathbf{y}_i)$, Mlodozeniec et al. (2024) consider a prediction rule in a general unsupervised setting as a sequence of functions $q_n$ that map a history of observations $\mathbf{z}_{1:n} := (\mathbf{z}_1, \ldots, \mathbf{z}_n)$ taking values in some space $\mathcal{Z}^n := \mathcal{X}^n \times \mathcal{Y}^n$ to a predictive distribution over the next observation $q(\mathbf{z}_{n+1}|\mathbf{z}_{1:n})$. The sequence of predictions $q_{1:n}$ defines a joint distribution on $\mathcal{Z}^n$. We transform this joint into a (finitely) exchangeable distribution $\hat{q}_{1:n}$ by averaging over the set $\Pi_n$ of all possible permutations:

$$\hat{q}_{1:n}(\mathbf{z}_1, \ldots, \mathbf{z}_n) = \frac{1}{n!} \sum_{\pi \in \Pi_n} q_{1:n}(\mathbf{z}_{\pi(1)}, \ldots, \mathbf{z}_{\pi(n)}) \quad (5)$$

where we refer to the output of such a transformation, $\hat{q}$, as an *exchangeabilified* version of $q$.

**Proposition 4.1.** *Let $p : \mathcal{Z}^n \to \mathbb{R}$ be the true (joint) data-generating distribution, which we assume to be exchangeable. Then, the following decomposition holds:*

$$D_{KL}(q_{1:n} \| p) = D_{KL}(\hat{q}_{1:n} \| p) + D_{KL}(q_{1:n} \| \hat{q}_{1:n}) \quad (6)$$

*and, in particular, $D_{KL}(q_{1:n} \| p) \geq D_{KL}(\hat{q}_{1:n} \| p)$ with equality if and only if $q_{1:n}$ is exchangeable.*

While Proposition 4.1 applies to general unsupervised settings, we adapt it to the more relevant regression setting as follows.

**Proposition 4.2.** *Let the true data-generating distribution $p$ be **conditionally** exchangeable, i.e., for the true **conditional** data-generating distribution $p_{Y_{1:n}|X_{1:n}} : \mathcal{Y}^n \times \mathcal{X}^n \to \mathbb{R}$ the following holds:*

$$p(\mathbf{y}_{1:n} \mid \mathbf{x}_{1:n}) = p(\mathbf{y}_{\pi_{1:n}} \mid \mathbf{x}_{\pi_{1:n}}) \quad \forall \pi, \mathbf{x}_{1:n}, \mathbf{y}_{1:n}, \quad (7)$$

*and the true **marginal** data-generating distribution over inputs $p_{X_{1:n}} : \mathcal{X}^n \to \mathbb{R}$ be i.i.d. Next, for any conditional density $q_{Y_{1:n}|X_{1:n}} : \mathcal{Y}^n \times \mathcal{X}^n \to \mathbb{R}$, obtain the exchangeabilified version of it as*

$$\hat{q}_{Y_{1:n}|X_{1:n}}(\mathbf{y}_{1:n}|\mathbf{x}_{1:n}) = \frac{1}{n!} \sum_{\pi \in \Pi_n} q_{Y_{1:n}|X_{1:n}}(\mathbf{y}_{\pi_{1:n}}|\mathbf{x}_{\pi_{1:n}}).$$

$$(8)$$

*Then, the following decomposition holds:*

$$\mathbb{E}_{p_{X_{1:n}}} \left[ D_{KL} \left[ q_{Y_{1:n}|X_{1:n}} \| p_{Y_{1:n}|X_{1:n}} \right] \right]$$
$$= \mathbb{E}_{p_{X_{1:n}}} \left[ D_{KL} \left[ \hat{q}_{Y_{1:n}|X_{1:n}} \| p_{Y_{1:n}|X_{1:n}} \right] \right]$$
$$+ \underbrace{\mathbb{E}_{p_{X_{1:n}}} \left[ D_{KL} \left[ q_{Y_{1:n}|X_{1:n}} \| \hat{q}_{Y_{1:n}|X_{1:n}} \right] \right]}_{KL\ Gap} \quad (9)$$

---

[2]Analysis assumes $N_s \gg N_t$

*and, in particular,* $\mathbb{E}_{p_{X_{1:n}}} \left[ D_{KL} \left[ q_{Y_{1:n}|X_{1:n}} \| p_{Y_{1:n}|X_{1:n}} \right] \right] \geq \mathbb{E}_{p_{X_{1:n}}} \left[ D_{KL} \left[ \hat{q}_{Y_{1:n}|X_{1:n}} \| p_{Y_{1:n}|X_{1:n}} \right] \right]$ *with equality for all* $n$ *if and only if* $q_{Y_{1:n}|X_{1:n}}$ *is conditionally exchangeable.*

Informally, Proposition 4.2 implies that, for any true data-generating mechanism, we can 'gain' $\mathbb{E}_{p_{X_{1:n}}} \left[ D_{KL} \left[ q_{Y_{1:n}|X_{1:n}} \| \hat{q}_{Y_{1:n}|X_{1:n}} \right] \right]$ in performance by replacing the non-exchangeable prediction rule with its exchangeable counterpart. We term this quantity the *KL gap*; it acts as a proxy for the performance lost due to the conditional non-exchangeability of $q_{Y_{1:n}|X_{1:n}}$ and quantifies the deviation from Bayesian behaviour. A KL gap of zero implies that $q_{Y_{1:n}|X_{1:n}}$ is perfectly exchangeable and thus implicitly Bayesian. We provide proofs for Propositions 4.1 and 4.2 in Appendix C.1. The approach can also be extended to non-iid inputs (Appendix C.2).

**Implicit Bayesianness for NPs** Since we are concerned with NPs applied to streaming data, we implement the proposed metric as follows. For a particular dataset of $N$ observations $\{\mathbf{x}_i, \mathbf{y}_i\}_{i=1}^{N}$, we define our NP prediction rule as an autoregressive factorisation of the joint predictive distribution: $p_{\text{NP}}(\mathbf{y}_{1:N}|\mathbf{x}_{1:N}) = \Pi_{i=1}^{N} p_{\text{NP}}(\mathbf{y}_i|\mathbf{y}_{1:i-1}, \mathbf{x}_{1:i})$—simulating how at each update we predict the next point conditioned on all past history[3]. We construct the exchangeable prediction rule $\hat{p}_{\text{NP}}$ following Equation 8, and compute the expected value of the KL gap over a set of $N_{\text{perm}}$ joint permutations of the input-output observations through Monte Carlo averaging (see Appendix E.3.1 for further details).

This empirical KL gap is computed under a teacher-forced protocol in which ground-truth observations are sequentially appended to the context, as is typical in streaming settings. We therefore interpret it as a measure of implicit Bayesianness under the *one-at-a-time* streaming setting considered here. Alternative empirical setups could be used to study implicit Bayesianness under different protocols such as batched streaming, as well as to investigate length generalisation or the compounding of prediction errors in streaming setups. As argued in Mlodozeniec et al. (2024), implicit Bayesianness is a desirable property, but one that can be trivially achieved by uninformative prediction rules. As such, in our empirical investigation we jointly report both the KL gap, as well as a measure of performance in terms of the average negative log-likelihood.

# 5. Experiments

We evaluate our approach on synthetic, tabular, and environmental prediction tasks. Our analysis proceeds in three

---

[3]Because NPs are not usually trained to support the null context scenario, we condition on a few fixed datapoints $p_{\text{NP}}(\mathbf{y}_{1:N}|\mathbf{x}_{1:N}, \mathcal{D}^{\text{fixed}}) = \Pi_{i=1}^{N} p_{\text{NP}}(\mathbf{y}_i|\mathbf{y}_{1:i-1}, \mathbf{x}_{1:i}, \mathcal{D}^{\text{fixed}})$, but this does not change the arguments above.

stages: 1) We assess the impact of causal masking in static settings; 2) We measure probabilistic consistency via implicit Bayesianness; and 3) We validate the scalability of `incTNP` in streaming scenarios using both factorised and autoregressive (AR) deployment. Detailed data generation protocols and hyperparameters are provided in Appendix D. We consider the following domains:

- **1D GP Regression:** Synthetic tasks generated from Gaussian Processes (GPs) with randomised radial basis function (RBF) kernel hyperparameters, trained on up to 64 context points. This standard benchmark tests the model's ability to meta-learn flexible function approximations.
- **Tabular Data Regression:** Following Hollmann et al. (2023), we train on synthetic data generated via structural causal models (SCMs) and evaluate on both held-out synthetic tasks and four real-world UCI regression benchmarks (Skillcraft, Powerplant, Elevators, Protein).
- **Station Temperature Prediction:** A real-world spatiotemporal task involving temperature prediction across stations in Africa and Europe (see Figure 9). We train the models on a spatiotemporal interpolation task, and evaluate performance on both the interpolation scenario, as well as a forecasting one, where targets are sampled strictly from future time windows relative to the context.

## 5.1. Static Context Performance

**Causal masking maintains predictive performance.** As shown in Table 1 (see Table 2 for standard errors), when evaluated in the static context setting, both in factorised (Table 1(a)), as well as AR mode (Table 1(b)), the `incTNP` variants achieve competitive if not better performance relative to the non-incremental `TNP-D` baseline. This is important to establish, as it shows that `incTNP` achieves computational efficiency and scalability without compromising predictive performance. We observe the following key trends:

**Impact of Training Strategy:** While causal masking inherently restricts the receptive field—evident in the slight performance drop of the vanilla `incTNP` compared to the full-context `TNP-D`—our data-efficient training strategy in `incTNP-Seq` effectively counteracts this limitation. Consequently, `incTNP-Seq` closes the gap to `TNP-D` on simpler tasks (1D GP) and outperforms it on more complex benchmarks (Tabular and Temperature) by leveraging the richer training signal.

**Autoregressive Efficiency:** In AR mode, `incTNP-Seq` exceeds the predictive performance of `TNP-D` while significantly reducing computational cost. By utilising linear-time $\mathcal{O}(N)$ context updates rather than re-encoding the full history ($\mathcal{O}(N^2)$), we achieve substantial speedups (e.g., $3.5\times$ faster on Temperature) without sacrificing accuracy. We also compare against the autoregressive `TNP-A`, with `incTNP-Seq` performing competitively whilst support-

*Table 1.* Average Test Log-Likelihoods (↑) for (a) Factorised and (b) AR deployment. We report the absolute performance of the reference non-incremental TNP-D. For all the other methods we report the difference (Δ) relative to the reference performance. Held-out task are shown in purple, while transfer scenarios, such as Sim-to-Real (Tabular) and Forecasting (Temperature) are coloured in orange. Best results (including ties within one SEM) are bolded, with standard errors provided in Table 2.

*(a)* **Factorised predictions**. `incTNP-Seq` achieves parity or outperforms `TNP-D` on held-out tasks, and dominates on transfer scenarios.

| | REFERENCE | OURS (Δ (↑) RELATIVE TO REF.) | | BASELINES (Δ (↑) RELATIVE TO REF.) | |
| DATASET | TNP-D | INCTNP | INCTNP-SEQ | CNP | LBANP |
|---|---|---|---|---|---|
| 1D GP | 0.431 | −0.013 | −0.002 | −0.230 | **+0.004** |
| TABULAR (SYNTHETIC) | 0.154 | −0.020 | **+0.007** | −0.330 | −0.058 |
| SKILLCRAFT | −0.954 | +0.002 | **+0.008** | −0.134 | −0.031 |
| POWERPLANT | −0.008 | +0.002 | **+0.003** | −0.269 | −0.024 |
| ELEVATORS | **-0.322** | −0.024 | −0.009 | −0.942 | −0.205 |
| PROTEIN | −1.152 | −0.028 | **+0.036** | −0.188 | −0.024 |
| TEMPERATURE (INTERP) | −1.703 | −0.011 | **+0.018** | −0.533 | −0.090 |
| TEMPERATURE (FORECAST) | −2.571 | +0.030 | **+0.690** | +0.181 | +0.268 |

*(b)* **AR predictions.** Cost (↓) is the ratio of inference time versus `incTNP-Seq` ($T_{model}/T_{incTNP-Seq}$) (> 1.0 indicates slower inference).

| | TNP-D (REF) | | INCTNP | | INCTNP-SEQ | | CNP | | LBANP | | TNP-A | |
| DATASET | LL (↑) | COST | Δ (↑) | COST | Δ (↑) | COST | Δ (↑) | COST | Δ (↑) | COST | Δ (↑) | COST |
|---|---|---|---|---|---|---|---|---|---|---|---|---|
| 1D GP | 0.767 | 1.21× | −0.007 | 1.01× | **+0.002** | 1.00× | −0.230 | 0.11× | −0.001 | 1.38× | **+0.004** | 1.04× |
| TEMPERATURE (INTERP) | −1.670 | 3.57× | −0.038 | 1.00× | +0.016 | 1.00× | −0.552 | 0.02× | −0.097 | 1.50× | **+0.031** | 0.81× |
| TEMPERATURE (FORECAST) | −1.749 | 3.57× | −0.005 | 1.00× | **+0.061** | 1.00× | −0.559 | 0.02× | −0.120 | 1.50× | +0.035 | 0.81× |

ing efficient incremental context updates. `TNP-A` attains strong performance and low static-context inference cost on Temperature tasks; however, it retains an $\mathcal{O}(N^2)$ context-update cost, which becomes prohibitively expensive at scale for streaming. For instance, `TNP-A` is 3.5× slower than `incTNP` on Temperature at $N_s = 100,000$; we discuss `TNP-A` runtime and additional costs in Appendix E.2.

**Comparison to Baselines:** Our method bridges the gap between flexibility and performance. Our best variant, `incTNP-Seq`, significantly outperforms `CNP`, the primary baseline supporting incremental updates, whose performance is limited by its capacity. Furthermore, `incTNP-Seq` outperforms or achieves parity with `LBANP` (Feng et al., 2023), a strong baseline that offers efficient forward passes via pseudo-tokens but does not support incremental updates. `incTNP` follows the same trend, with the exception of zero-shot temperature forecasting under factorised prediction, where the combination of highly correlated targets and train-test mismatch favours the `CNP` baseline over both `TNP-D` and `incTNP`, likely due to underfitting. Importantly, performance is recovered through AR deployment, which is better matched to the target structure of this task, or by using the densely trained `incTNP-Seq`.

**Dense autoregressive training improves hyperparameter robustness.** We assess the hyperparameter sensitivity of each model by sweeping over a range of learning rates and schedules (see Appendix E.1) on the 1D GP and Tabular benchmarks. After filtering for outliers, the performance dis-

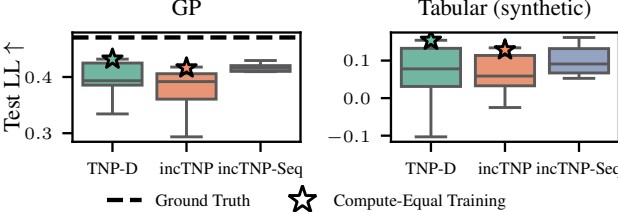

*Figure 2.* Test log-likelihoods (↑) on synthetic GP (RBF kernel, $N_c$ up to 64) and Tabular datasets across multiple training configurations. `incTNP-Seq` exhibits reduced variance compared to `TNP-D` and `incTNP`, demonstrating the robustness offered by its training strategy.

tributions in Figure 2 reveal that `incTNP-Seq` yields significantly lower variance than its counterparts. We attribute this stability to the dense autoregressive training objective, which likely reduces gradient variance (see Appendix E.1.3), with implicit regularisation and a smoother loss landscape also potentially contributing. Although this training strategy incurs a higher cost per forward pass ($\mathcal{O}((N_c + N_t)^2)$) compared to standard training ($\mathcal{O}(N_c^2 + N_c N_t)$), the resulting gradients are more informative. To test if this benefit is purely due to compute, we allocate to `TNP-D` and `incTNP` a compute budget equivalent to that of `incTNP-Seq` and report their peak performance, indicated by stars in Figure 2. Since standard training is cheaper per step, this budget allows these models to train for more iterations and observe more data. Crucially, even with this advantage, they fail to surpass the peak performance of `incTNP-Seq`.

## 5.2. Implicit Bayesianness

Next, we investigate the impact of causal masking on implicit Bayesianness—the degree to which the model behaves like a consistent Bayesian predictor during sequential updates. While standard TNPs guarantee permutation invariance for a fixed context set, they do not inherently satisfy conditional consistency as the context grows. incTNP further sacrifices permutation invariance with respect to the context set to enable efficient streaming. We therefore ask: Does this sensitivity to context order degrade the model's ability to approximate a valid Bayesian posterior in this setting compared to the TNP-D, a model that is permutation invariant with respect to the context set?

We evaluate this on the 1D GP benchmark using the KL Gap metric described in Section 4.4. We condition all models on a fixed initial context ($N_c = 20$) and evaluate predictions on $N_t = 40$ targets. Inference is performed sequentially under a teacher-forced protocol: the model predicts a target, observes the ground truth, updates its context, and proceeds to the next point. We simulate multiple permutations of the same data stream, compute the KL Gap for each, and then average the results. Remarkably, as shown in Figure 3, the KL Gap for incTNP variants—particularly incTNP-Seq—is comparable to that of TNP-D. This finding suggests that the primary driver of inconsistency in streaming settings is the lack of conditional consistency (Xu et al., 2023) inherent to multiple NP variants, rather than the specific causal masking mechanism of incTNP. In practice, incTNP is no less "Bayesian" than the standard TNP-D under the empirical protocol considered here. We provide an additional example in Appendix E.3.2 which further supports this claim.

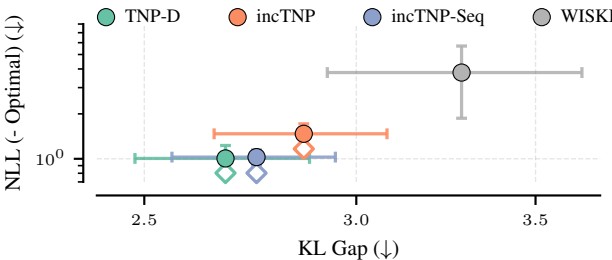

*Figure 3.* Performance (Joint NLL gap relative to optimal) versus implicit Bayesianness (KL Gap). incTNP-Seq achieves a KL gap similar to the non-causal TNP-D, whereas the streaming GP baseline (WISKI) performs worse in both metrics. Diamond markers ($\diamond$) denote the exchangeable versions of the models.

We also benchmark our method against WISKI (Stanton et al., 2021), a scalable streaming GP framework. While exact GP inference is perfectly Bayesian (yielding a KL gap of zero), WISKI relies on structured kernel interpolation approximations to achieve scalability. As shown in Figure 3, these induce a significant degradation of Bayesian consistency; WISKI exhibits a higher KL gap than both

TNP-D and the incTNP variants, alongside inferior predictive performance. This underscores incTNP's superior performance in streaming, maintaining greater Bayesian consistency and accuracy than established approximate GP baselines in this one-at-a-time streaming setting.

## 5.3. Streaming Performance

This last section focuses on a streaming setup, measuring performance on a fixed set of targets as observations accumulate. While all variants improve performance with new evidence, TNP-D becomes prohibitively expensive—re-encoding its full history during each update incurs a quadratic cost ($\mathcal{O}(N_s^2)$). In contrast, the incTNP variants leverage caching to achieve linear update scaling ($\mathcal{O}(N_s)$), all while matching or exceeding baseline accuracy.

**Streaming Tabular Data (Factorised)** Following the paradigm of tabular foundation models (Hollmann et al., 2025), we meta-train on a synthetic prior and evaluate in two distinct regimes: (1) held-out synthetic tasks sampled from the same prior used during training; and (2) real-world datasets, which involve sim-to-real transfer and test robustness to distributional shifts. As shown in Figures 4 and 5, incTNP-Seq consistently matches or outperforms TNP-D, which in turn outperforms the non-sequential incTNP. Crucially, we observe strong length generalisation: despite training on contexts of up to only $N_c = 1024$, the models seamlessly adapt to context streams exceeding $30,000$ points at inference without OOD degradation. However, as the stream grows, the quadratic update cost of TNP-D becomes prohibitive, whereas incTNP remains computationally tractable (Figure 5, right). We provide additional performance and cost comparisons against WISKI and LBANP in Appendix E.4.1.

**Streaming Temperature Prediction (AR)** Finally, we evaluate our models on the streaming version of the temperature prediction task. We consider two regimes of practical interest: Interpolation (e.g., reconstructing data from faulty or sparse sensors), and Forecasting (e.g, predicting future weather patterns based on historical data). Crucially, to capture spatiotemporal correlations, we employ AR decoding over targets at *every* streaming step. This creates a "nested" computational burden that renders standard TNP-D prohibitive due to its quadratic scaling (see Appendix F for computational cost plots). In contrast, incTNP handles this load efficiently. As shown in Figure 6, incTNP-Seq generally outperforms the other NP variants—including the non-incremental LBANP and the capacity-limited CNP—delivering the necessary performance for correlated, real-time predictions. We report results for TNP-A in Appendix E.4.2.

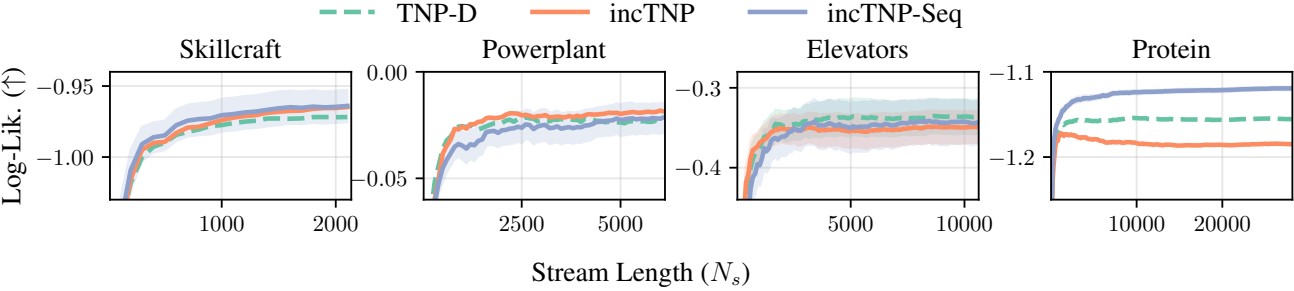

*Figure 4.* Test log-likelihood (↑) on Tabular (real-world datasets). Performance generally increases with stream length for all models. While `incTNP` and `TNP-D` remain competitive on smaller datasets, `incTNP-Seq` demonstrates greater robustness on Protein.

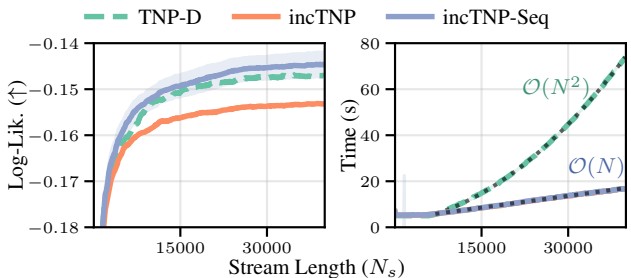

*Figure 5.* Test log-likelihood (↑) on the Tabular (Synthetic) dataset. (Left) Performance improves with more data, with `incTNP-Seq` consistently outperforming `TNP-D` and `incTNP`—even when evaluating beyond the training limit of 1024 points. (Right) While achieving similar accuracy, the incremental variants demonstrate significantly better scaling efficiency as context size grows.

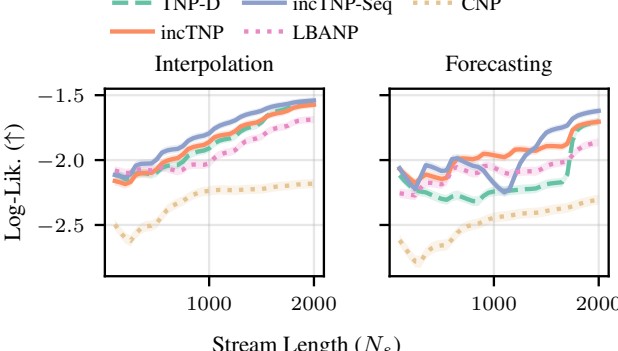

*Figure 6.* Test log-likelihood (↑) on the Temperature Prediction task (streaming AR mode).

## 6. Conclusion

This work introduces the `incTNP`, a new Neural Process variant designed for streaming settings where efficient context updates are imperative. This is achieved by incorporating causal masking and KV caching, and adapting the training strategy to a dense autoregressive objective (`incTNP-Seq`) which improves training efficiency and stability. Empirically, `incTNP` (especially `incTNP-Seq`) matches the predictive accuracy of the standard `TNP-D` across synthetic and real-world benchmarks, while reducing the computational update cost from $\mathcal{O}(N^2)$ to $\mathcal{O}(N)$. Furthermore, using our proposed "implicit Bayesianness" metric, we find that in one-at-a-time streaming scenarios where the context evolves sequentially, `incTNP` retains a prediction rule that is practically no less Bayesian than `TNP-D`.

Several promising avenues for future research remain. It would be valuable to further analyse `incTNP`'s robustness to distributional shifts and non-stationary environments, building on our preliminary experiment in Appendix I. Its linear-time updates and robust uncertainty estimates also make `incTNP` an attractive surrogate model for active learning scenarios. When observation order can be controlled,

context-order sensitivity may be exploited to improve performance; we propose a context ordering algorithm as part of our broader analysis of ordering sensitivity in Appendix H. Tabular foundation models such as TabPFN (Hollmann et al., 2025) could incorporate `incTNP`'s incremental framework to support streamed row and column updates. Having laid the foundations for analysing the implicit Bayesianness of NPs, future work developing and interpreting consistency measures could provide further insight into the probabilistic coherence of NP updates. Finally, although our tabular experiments show strong length generalisation, handling $N_s = 30,000$ after training on $N_c = 1024$, further testing is needed at extreme context scales. To address potential KV-cache memory bottlenecks, future work could replace attention with State-Space Models (SSMs) (Gu & Dao, 2024), whose constant-time updates and fixed state size could enable even larger context windows.

## Acknowledgements

Cristiana Diaconu is supported by the Cambridge Trust Scholarship. Richard E. Turner is supported by Google, Amazon, ARM, Improbable and the EPSRC Probabilistic AI Hub (EP/Y028783/1).

## Impact Statement

This paper presents work whose goal is to advance the field of Machine Learning. There are many potential societal consequences of our work, none of which we feel must be specifically highlighted here.

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

# A. Hardware specifications

Training and inference for synthetic GP regression, as well as the training of tabular models, were performed on a single NVIDIA GeForce RTX 2080 Ti GPU (20 CPU cores). For the evaluation of tabular models within the streaming context and for non-streaming evaluation on the Protein UCI dataset, we conduct experiments on a single NVIDIA RTX 6000 Ada Generation GPU (48 GB) with 56 CPU cores. For the computationally expensive temperature prediction task, we perform training and inference for all models on a single NVIDIA A100 80GB GPU with 32 CPU cores. We benchmark tabular model runtimes on the NVIDIA RTX 6000 Ada GPU and environmental model runtimes on the NVIDIA A100 80GB GPU, ensuring that FlashAttention (Dao, 2024) is used across both environments.

# B. Experiment Details

For our neural process baseline implementations, we adopt the same architectures as detailed in Ashman et al. (2025); see Appendix A from Ashman et al. (2025) for architectural diagrams of `TNP-D` and `LBANP` models. Full code for reproducing our experiments is provided in the supplementary material.

**Optimiser details**    For all experiments, we optimise parameters using AdamW (Loshchilov & Hutter, 2019) with gradient norms clipped to 0.5, using $\beta_1 = 0.9$, $\beta_2 = 0.999$ and a weight decay of $1 \times 10^{-2}$. We explore the use of both a fixed learning rate and a cosine decay learning rate schedule (Loshchilov & Hutter, 2017). The latter strategy consists of a linear warmup phase across the first 10% of training steps to the maximum learning rate, followed by a half-cosine curve decay to a minimum learning rate of $1 \times 10^{-6}$. For the tabular data and GP regression tasks, we explore both schedulers and a sweep of learning rates. For the computationally demanding temperature prediction task (HadISD), we use the cosine learning rate scheduler and a learning rate of $5 \times 10^{-4}$ for all trained models.

**Training seed**    Unless otherwise specified, all reported results are from a single training run per model, using a single training seed.

**Log-likelihood performance metric**    To evaluate the performance of our NP models, we report the average log-likelihood computed over $K$ test tasks. For the $i$-th task with context set $\mathcal{D}_c^{(i)}$ and $N_t$ target points $\{\mathbf{x}_{n,i}^{(t)}, \mathbf{y}_{n,i}^{(t)}\}_{n=1}^{N_t}$, the metric is defined as:

$$LL = \frac{1}{KN_t} \sum_{i=1}^{K} \sum_{n=1}^{N_t} \log \mathcal{N}\left(\mathbf{y}_{n,i}^{(t)}; \mu_\theta(\mathbf{x}_{n,i}^{(t)}, \mathcal{D}_c^{(i)}), \sigma_\theta^2(\mathbf{x}_{n,i}^{(t)}, \mathcal{D}_c^{(i)})\right), \tag{10}$$

where $\mu_\theta(\cdot)$ and $\sigma_\theta^2(\cdot)$ denote the predicted mean and variance parameters at the target location $\mathbf{x}_{n,i}^{(t)}$ conditioned on the context $\mathcal{D}_c^{(i)}$.

**Likelihood details**    All models parameterise Gaussian predictive likelihoods via a mean and variance. To ensure non-negativity of the variances, we pass the models' raw variance outputs through a softplus function. We also set a minimum noise level of $\boldsymbol{\sigma}_{\min}^2 = 1 \times 10^{-4}$ for tabular data experiments, and $\boldsymbol{\sigma}_{\min}^2 = 0$ for all other experiments. Thus, given a target representation $\mathbf{z}_t$, our decoder, $\varphi$, outputs:

$$\boldsymbol{\mu}_{\theta,t}, \, \log(\exp(\boldsymbol{\sigma}_{\theta,t} - \boldsymbol{\sigma}_{\min})^2 - 1) = \varphi(\mathbf{z}_t), \quad p(\mathbf{y}_t \mid \mathcal{D}_c, \mathbf{x}_t) = \mathcal{N}(\mathbf{y}_t; \boldsymbol{\mu}_{\theta,t}, \boldsymbol{\sigma}_{\theta,t}^2). \tag{11}$$

**Embedder / decoder details**    For all models, we embed each context point to a fixed dimensional representation in $\mathbb{R}^{128}$ (i.e., $D_z = 128$) using an MLP $\phi$ with two hidden layers. Similarly, all models use an MLP with two hidden layers to decode target predictions. All MLPs use 128 neurons per hidden layer (matching $D_z$).

**CNP details**    Context points $(\mathbf{x}_n^c, \mathbf{y}_n^c)$ are embedded into a representation $\mathbf{z}_n^c \in \mathbb{R}^{D_z}$ and aggregated together using $\mathbf{z}_c = \frac{1}{N_c} \sum_{n=1}^{N_c} \mathbf{z}_n^c$. This representation is concatenated with the target input $\mathbf{x}_t$ and fed through the decoder to produce the predictive likelihood parameters $\{\boldsymbol{\mu}_t, \boldsymbol{\sigma}_t\}$.

**TNP family shared backbone**    Each transformer block consists of an attention mechanism (either MHSA or MHCA) and an MLP, with Layer normalisation (LN) applied before each operation (pre-norm). We utilise $H = 8$ heads, each with

dimension $D_V = 16$ and $D_{QK} = D_Z$ throughout. Each block uses two residual connections. The update rule for a block is defined as:

$$
\begin{aligned}
\widetilde{\mathbf{Z}} &\leftarrow \mathbf{Z} + \text{Attention}(\text{LN}_1(\mathbf{Z})) \\
\mathbf{Z} &\leftarrow \widetilde{\mathbf{Z}} + \text{MLP}(\text{LN}_2(\widetilde{\mathbf{Z}})).
\end{aligned}
\tag{12}
$$

Across all transformer models, we use 5 layers.

**LBANP details**   For our `LBANP` implementation, we compress our context set into $L_{\text{LBANP}}$ latent vectors to obtain a latent representation $\mathbf{U} \in \mathbb{R}^{L_{\text{LBANP}} \times D_z}$, using a Perceiver (Jaegle et al., 2021) style encoder:

$$
\begin{aligned}
\mathbf{U} &\leftarrow \text{MHCA-layer}(\mathbf{U}, \mathbf{Z}_c) \\
\mathbf{U} &\leftarrow \text{MHSA-layer}(\mathbf{U}) \\
\mathbf{Z}_t &\leftarrow \text{MHCA-layer}(\mathbf{Z}_t, \mathbf{U}).
\end{aligned}
\tag{13}
$$

We use $L_{\text{LBANP}} = 32$ latent vectors for all GP experiments, $L_{\text{LBANP}} = 128$ for tabular data tasks and $L_{\text{LBANP}} = 256$ for temperature modelling, reflecting the increased complexity of the latter two tasks.

**Common incTNP transformer details**   `incTNP` class models use the same transformer architecture as TNP models, but substitute unmasked attention operations (MHSA/MHCA) with masked attention operations (M-MHSA/M-MHCA) where appropriate. All causal attention calculations use a causal mask $M^{\text{causal}}$:

$$
M_{i,j}^{\text{causal}} = \begin{cases} 0 & \text{if } i \geq j \\ -\infty & \text{otherwise.} \end{cases}
\tag{14}
$$

Incremental TNP models use updated masked attention layer operations where appropriate:

$$
\begin{aligned}
\widetilde{\mathbf{Z}} &\leftarrow \mathbf{Z} + \text{Attention}(\text{LN}_1(\mathbf{Z}), \text{mask} = M) \\
\mathbf{Z} &\leftarrow \widetilde{\mathbf{Z}} + \text{MLP}(\text{LN}_2(\widetilde{\mathbf{Z}})).
\end{aligned}
\tag{15}
$$

During inference, both `incTNP` and `incTNP-Seq` use causal masking $M^{\text{causal},c} \in \mathbb{R}^{N_c \times N_c}$ to encode the context set, and unmasked cross-attention to handle target-context MHCA. At training time, `incTNP-Seq` trains across the sequence $N = N_c + N_t$, imposing a contextual masking $M^{\text{causal},c} \in \mathbb{R}^{N \times N}$, and a causal target-context masking $M^{\text{causal},t} \in \mathbb{R}^{(N-1) \times N}$.

**incTNP architecture**   To outline the flow of information through `incTNP`, consider a context set $\{\mathbf{x}_n^c, \mathbf{y}_n^c\}_{n=1}^{N_c}$ and a target set $\{\mathbf{x}_n^t, \mathbf{y}_n^t\}_{n=1}^{N_t}$. We construct input representations by augmenting these points with a boolean source flag (0 for context, 1 for target). Context points are concatenated as $[\mathbf{x}_n^c, \mathbf{y}_n^c, 0]$. Target points are concatenated as $[\mathbf{x}_n^t, \mathbf{0}, 1]$, where $\mathbf{0}$ serves as a zero-vector placeholder for the unobserved target value. These vectors are passed through the MLP embedder to obtain the context and target embeddings $\mathbf{R}_{1:N_c}^{(c)}$ and $\mathbf{R}_{1:N_t}^{(t)}$. These representations are propagated through the $L = 5$ self-attention and cross-attention layers:

$$
\mathbf{Z}_l^{(c)} = \text{M-MHSA-layer}(\mathbf{Z}_{l-1}^{(c)}, M^{\text{causal}}) \quad l \in 1, \cdots, L \quad \text{where} \quad \mathbf{Z}_0^{(c)} = \mathbf{R}_{1:N_c}^{(c)}
\tag{16}
$$

$$
\mathbf{Z}_l^{(t)} = \text{MHCA-layer}(\mathbf{Z}_{l-1}^{(t)}, \mathbf{Z}_l^{(c)}) \quad l \in 1, \cdots, L \quad \text{where} \quad \mathbf{Z}_0^{(t)} = \mathbf{R}_{1:N_t}^{(t)}.
\tag{17}
$$

Finally, the MLP decoder maps the resulting representation $\mathbf{Z}_L^{(t)}$ to the parameters of the factorised predictive distribution $p_\theta(\mathbf{Y}_t \mid \mathcal{D}_c, \mathbf{X}_t)$. A visualisation of `incTNP`'s architecture can be seen in Figure 7.

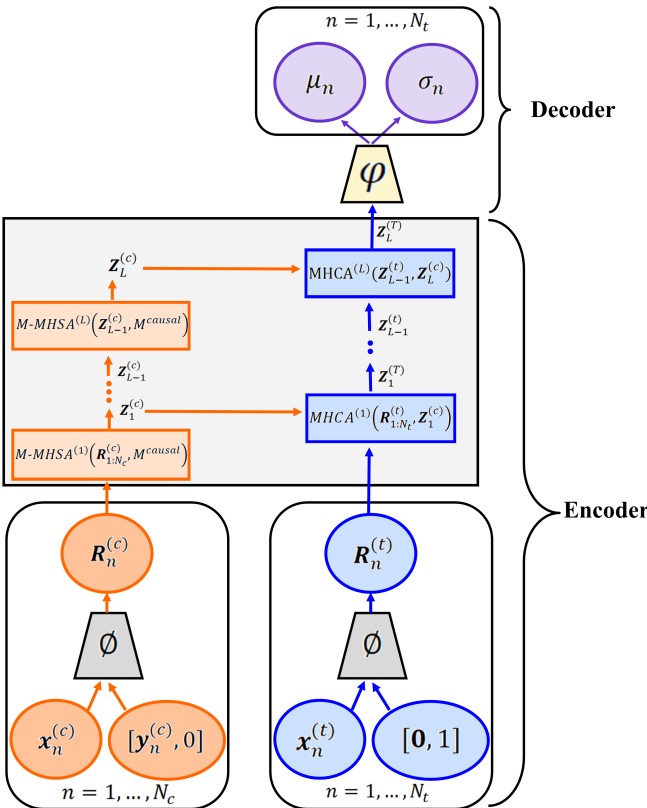

*Figure 7.* Architecture of the incremental Transformer Neural Process (`incTNP`), which uses interwoven M-MHSA and MHCA layers.

**incTNP-Seq data flow** At test time, `incTNP-Seq` behaves identically to `incTNP`. At training time, data is treated as a single joined sequence such that $\mathbf{X}^{(seq)} = [\mathbf{X}_c, \mathbf{X}_t]$ and $\mathbf{Y}^{(seq)} = [\mathbf{Y}_c, \mathbf{Y}_t]$. Two copies of this sequence are used to generate a context sequence $\mathbf{D}_c^{(seq)} = \phi([\mathbf{X}^{(seq)}, \mathbf{Y}^{(seq)}, \mathbf{0}])$ and a target sequence $\mathbf{D}_t^{(seq)} = \phi([\mathbf{X}^{(seq)}, \mathbf{0}, \mathbf{1}])$, where $\phi$ is the MLP embedder. Causally masked self-attention updates the context sequence representation as it does for `incTNP`. In addition to this, masked cross-attention is used to enforce a causal structure. That is a target token $(\mathbf{x}_i^{(seq)}, \mathbf{0}, 1)$, is only permitted to attend to preceding context tokens $(\mathbf{x}_j^{(seq)}, \mathbf{y}_j^{(seq)}, 0)$ for $1 \le j < i$, resulting in the following transformer update equations:

$$\mathbf{Z}_l^{(c)} = \text{M-MHSA-layer}(\mathbf{Z}_{l-1}^{(c)}, M^{\text{causal},c}) \quad l \in 1, \cdots, L \quad \text{where} \quad \mathbf{Z}_0^{(c)} = \mathbf{D}_c^{(seq)} \tag{18}$$

$$\mathbf{Z}_l^{(t)} = \text{M-MHCA-layer}(\mathbf{Z}_{l-1}^{(t)}, \mathbf{Z}_l^{(c)}, M^{\text{causal},t}) \quad l \in 1, \cdots, L \quad \text{where} \quad \mathbf{Z}_0^{(t)} = (\mathbf{D}_t^{(seq)})_{2:N_c+N_t}. \tag{19}$$

The data flow and masking configuration of `incTNP-Seq` during training are outlined in Figure 8.

**WISKI details** We utilise the Woodbury Inversion with SKI (`WISKI`) framework proposed by Stanton et al. (2021), using their official implementation [4]. `WISKI` facilitates fast online learning by approximating the kernel matrix using a set of $m$ inducing points placed on a structured grid (Structured Kernel Interpolation). The `WISKI` architecture requires a user-specified kernel function and optionally a learnable stem to reduce effective feature dimensionality. Training consists of two phases:

- **Pretraining:** `WISKI` is pretrained on an initial subset of contextual observations (of size $N_{\text{init}}$) over 200 epochs using the Adam optimiser. GP hyperparameters are updated using a learning rate of $lr_{\text{init}}^{(\text{GP})}$, and stem hyperparameters are updated using a learning rate of $lr_{\text{init}}^{(\text{stem})}$.

[4] Official `WISKI` implementation available at https://github.com/wjmaddox/online_gp

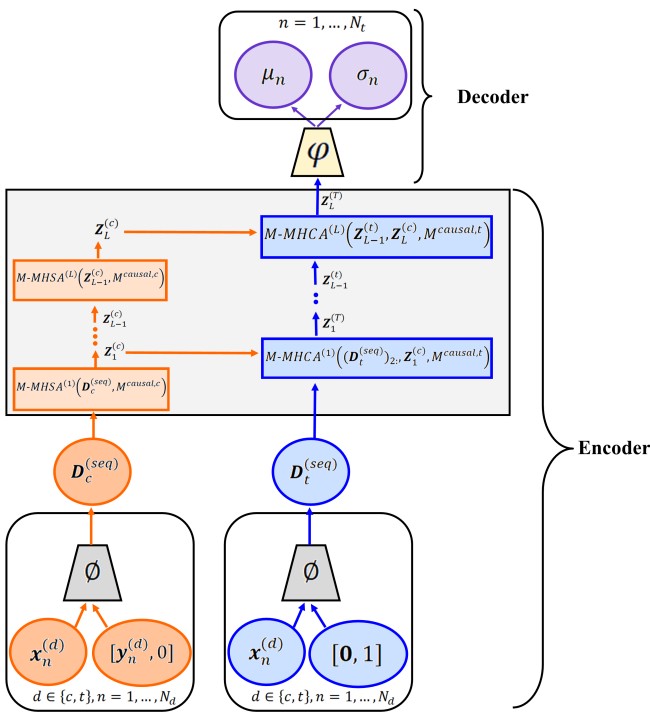

*Figure 8.* Schematic of `incTNP-Seq` during training. All M-MHSA and M-MHCA layers use causal masking.

- **Online Updates:** For the remaining data stream, the model updates sequentially, using a single Adam optimisation step per streamed point. GP hyperparameters are updated using a learning rate of $lr_{\text{stream}}^{(\text{GP})}$, and stem hyperparameters are updated using a learning rate of $lr_{\text{stream}}^{(\text{stem})}$.

For the tabular data task, we copy the setup from Stanton et al. (2021). Specifically, we learn an RBF-ARD GP kernel and set the grid resolution to 16 per dimension (resulting in $m = 256$ total inducing points). We employ a stem that projects data to $\mathbb{R}^2$ using a learned linear projection, a batch-norm operation, and a hyperbolic tangent activation in sequence. We define an initial learning rate for each task $lr_{(\text{base})}$ and set the learning rates as $lr_{\text{init}}^{(\text{GP})} = lr_{(\text{base})}$, $lr_{\text{init}}^{(\text{stem})} = 10^{-1} lr_{(\text{base})}$, $lr_{\text{stream}}^{(\text{GP})} = 10^{-1} lr_{(\text{base})}$ and $lr_{\text{stream}}^{(\text{stem})} = 10^{-2} lr_{(\text{base})}$. For the Powerplant and Skillcraft datasets, we set $lr_{(\text{base})} = 5 \times 10^{-2}$ and for all other tabular tasks we use $lr_{(\text{base})} = 1 \times 10^{-2}$. We pretrain tabular `WISKI` models using 5% of the total contextual data.

For GP regression tasks, we match the GP kernel to the same class as the underlying GP kernel and do not use a stem. We set $lr_{\text{init}}^{(\text{GP})} = 5 \times 10^{-2}$ and $lr_{\text{stream}}^{(\text{GP})} = 10^{-1} lr_{\text{init}}^{(\text{GP})}$, having tuned $lr_{\text{init}}^{(\text{GP})}$ on the GP dataset using a learning rate sweep. $m = 32$ inducing points are utilised for this task. In Section 5.2, we use $N_{\text{init}} = 20$ points to pretrain `WISKI`.

### B.1. 1D Gaussian Process Regression

**Training and testing details** We train NP models on the GP regression task for 16,000 samples with a batch size of 16. We train for 200 epochs on the RBF kernel and for 400 epochs on a more challenging mixed kernel by default. All samples within a batch are drawn from the same randomly sampled GP kernel. For each batch, we select $N_t = 128$ target points for evaluation and $N_c \sim \mathcal{U}[1, N_c^{(\text{max})}]$ context points, with model training lasting approximately 3 hours. We sample uniformly from an input range of $[-2, 2]$, with the standard deviation of observation noise set to $\sigma_{\text{obs}} = 0.1$ across all experiments. An outline of our training procedure is provided in Algorithm 1. We evaluate predictive performance across 80,000 random samples.

---

**Algorithm 1** 1D GP Regression Training

---

1: **Input:** Target count $N_t$, context range $[N_c^{(\min)}, N_c^{(\max)}]$, epochs $E$, samples per epoch $S_E$, batch size $B$, learning rate $l_r$, kernel family $k$, observation noise $\sigma_{\mathrm{obs}}$, NP model $\mathrm{NP}_\theta$.

2: **for** epoch $= 1, \ldots, E$ **do**

3:  **for** step $s = 1, \ldots, S_E/B$ **do**

4:   $k_s \sim k$ {Sample kernel hyperparameters}

5:   $N_c \sim \mathcal{U}[N_c^{(\min)}, N_c^{(\max)}]$

6:   $\mathcal{D}_c, \mathcal{D}_t \leftarrow \emptyset, \emptyset$

7:   **for** $b = 1, \ldots, B$ **do**

8:    $f^{(b)} \leftarrow \mathrm{SampleGP}(0, k_s)$

9:    $\mathbf{x}^{(b)} \sim \mathcal{U}[-2, 2]^{N_t + N_c}$

10:    $\boldsymbol{\epsilon}^{(b)} \sim \mathcal{N}(\mathbf{0}, I)$

11:    $\mathbf{y}^{(b)} \leftarrow f^{(b)}(\mathbf{x}^{(b)}) + \sigma_{\mathrm{obs}} \boldsymbol{\epsilon}^{(b)}$

12:    $\mathcal{D}_c \leftarrow \mathcal{D}_c \cup \{(\mathbf{x}_{1:N_c}^{(b)}, \mathbf{y}_{1:N_c}^{(b)})\}$

13:    $\mathcal{D}_t \leftarrow \mathcal{D}_t \cup \{(\mathbf{x}_{N_c+1:N_c+N_t}^{(b)}, \mathbf{y}_{N_c+1:N_c+N_t}^{(b)})\}$

14:   **end for**

15:   $\mathbf{X}_t, \mathbf{Y}_t \leftarrow \mathcal{D}_t$

16:   pred_dist $\leftarrow \mathrm{NP}_\theta(\mathbf{X}_t, \mathcal{D}_c)$

17:   $\mathcal{L} \leftarrow \mathrm{NLL}(\mathrm{pred\_dist}, \mathbf{Y}_t)$ {Negative log-likelihood}

18:   $\theta \leftarrow \mathrm{AdamW}(\theta, \nabla_\theta \mathcal{L}, l_r)$

19:  **end for**

20: **end for**

---

## B.2. Tabular Data Regression

**Training and testing details**   We train our models in exactly the same manner as for the GP regression task, with the key difference being that input features are multi-dimensional $\mathbf{x} \in \mathbb{R}^{20}$. When performing inference on real-world tabular tasks, datasets with fewer than 20 features are padded with zeros to be 20-dimensional. We train NP models on a synthetically generated dataset for 500 epochs, using 32,768 samples per epoch with a batch size of 128. For each training step, we select $N_t = 128$ target points and draw $N_c \in \mathcal{U}[10, 1024]$ context points. We evaluate synthetic model performance using 80,000 test samples. We adopt a random data ordering for all tabular tasks.

## B.3. Station Temperature Prediction

**Task overview**   For the temperature prediction task using HadISD data, we train models to perform spatiotemporal interpolation conditioned on contextual observations. Consequently, the model learns a spatiotemporal function mapping from time $t$, latitude $\phi$, longitude $\lambda$ and elevation $z$ to a predicted temperature $T$:

$$f \colon \mathbb{R}_t \times [-20^\circ, 60^\circ]_\phi \times [-10^\circ, 52^\circ]_\lambda \times \mathbb{R}_z \longrightarrow \mathbb{R}_T, \quad (t, \phi, \lambda, z) \mapsto T(t, \phi, \lambda, z). \tag{20}$$

**Fourier embeddings**   Raw inputs are 4-dimensional, but we employ Fourier embeddings to extract multi-scale features, resulting in input vectors $\mathbf{x} \in \mathbb{R}^{128}$ (using 32 dimensions per feature). This approach has been established for foundational NP weather forecasting models, and we adopt a similar setup to (Bodnar et al., 2025), where we Fourier encode each scalar feature $x_j$ into a $D$ dimensional vector:

$$\mathrm{FourierEncode}(x_j) = \left[\cos\left(\frac{2\pi x_j}{\Lambda_i}\right), \sin\left(\frac{2\pi x_j}{\Lambda_i}\right)\right] \quad \text{for } 0 \le i < D/2. \tag{21}$$

The wavelengths $\Lambda_i$ are logarithmically spaced between a minimum $\Lambda^{(\min)}$ and maximum $\Lambda^{(\max)}$ wavelength:

$$\Lambda_i = \exp\left(\log \Lambda^{(\min)} + i \frac{\log \Lambda^{(\max)} - \log \Lambda^{(\min)}}{D/2 - 1}\right). \tag{22}$$

We set $\Lambda^{(\min)}$ and $\Lambda^{(\max)}$ to match the physical scales we aim to capture in the normalised data. For elevation, we select $\Lambda_z^{(\min)} = 0.1$ and $\Lambda_z^{(\max)} = 8$, reflecting the limited impact of fine-scale elevation variations and a maximum normalised altitude just below 8. For latitude and longitude, we use $\Lambda^{(\min)} = 0.001$ and $\Lambda^{(\max)} = 2$. For time, we choose $\Lambda_t^{(\min)} = 1$ and $\Lambda_t^{(\max)} = 8760$ (representing the number of hours in a year) to capture seasonal variations.

**Timestamp sampling** To ensure observational diversity, we sample training window start times $t$ proportionally to the temporal density of observations. We precompute a count array $C$ containing the number of observations within each window starting at a given timestamp, and define a probability mass function over all valid timestamps $t_{\text{valid}}$:

$$\mathbf{w} \leftarrow \left[ w_i = \frac{C[i]}{\sum_{j \in t_{\text{valid}}} C[j]} \right]_{i \in t_{\text{valid}}} . \tag{23}$$

**Training and testing details** We train for 200 epochs using 32,000 samples per epoch and a batch size of 32. We train on a spatiotemporal interpolation task using $H = 8$ time windows stepped in increments of $\delta = 6$ hours, resulting in a 48-hour task window. For each sample, we select $N_t = 250$ target stations and sample $N_c \sim \mathcal{U}[100, 2100]$ context stations, randomly distributed in space-time. A temporal ordering is imposed on the context sequence. We evaluate factorised predictive performance across 80,000 random test samples and autoregressive (AR) predictive performance across 4,096 test samples.

## C. Proofs

### C.1. Exchangeability gap results for *i.i.d.* data

**Notation** For densities and conditional densities on some sequence space $\mathcal{X}^n$ (or $(\mathcal{X} \times \mathcal{Y})^n$ in the case of regression), we will indicate with a subscript what part of the product space each argument refers to. For example, we might write $p_{X_1,Y_1,\ldots,X_n,Y_n} : (\mathcal{X} \times \mathcal{Y})^n \to \mathbb{R}$ for a joint density on $(\mathcal{X} \times \mathcal{Y})^n$, $p_{X_1,\ldots,X_n}(x_1,\ldots,x_n) := \int p_{X_1,Y_1,\ldots,X_n,Y_n}(x_1,y_1,\ldots,x_n,y_n) dy_1 \ldots dy_n$ for the corresponding marginal density, $p_{Y_1,\ldots,Y_n|X_1,\ldots,X_n}(y_1,\ldots,y_n|x_1,\ldots,x_n) = \frac{p_{X_1,Y_1,\ldots,X_n,Y_n}(x_1,y_1,\ldots,x_n,y_n)}{p_{X_1,\ldots,X_n}(x_1,\ldots,x_n)}$ for the specific conditional density, e.t.c.. We will also often drop the subscript if it can be readily inferred from the arguments, e.g. writing $p(x_1,\ldots,x_n)$ instead of $p_{X_1,\ldots,X_n}(x_1,\ldots,x_n)$. Sometimes we'll also abbreviate $x_1,\ldots,x_n$ as $x_{1:n}$, freely asserting equivalences between spaces like $\mathcal{X}^n \times \mathcal{Y}^n$ and $\underbrace{\mathcal{X} \times \cdots \times \mathcal{X}}_{\times n} \times \underbrace{\mathcal{Y} \times \cdots \times \mathcal{Y}}_{\times n}$. For index permutations $\pi$ we will also write $\pi(1:n)$ for $\pi(1),\ldots,\pi(n)$.

C.1.1. PROOF OF PROPOSITION 4.1

We re-state Proposition 4.1 before providing the proof.

**Proposition.** *Let $p : \mathcal{Z}^n \to \mathbb{R}$ be the true (joint) data-generating distribution with full support[5], which we assume to be exchangeable. Then, the following decomposition holds:*

$$D_{KL}(q_{1:n} \parallel p) = D_{KL}(\hat{q}_{1:n} \parallel p) + D_{KL}(q_{1:n} \parallel \hat{q}_{1:n}) \tag{24}$$

*and, in particular, $D_{KL}(q_{1:n} \parallel p) \geq D_{KL}(\hat{q}_{1:n} \parallel p)$ with equality if and only if $q_{1:n}$ is exchangeable.*

---

[5]Else, the Kullbeck-Leibler Divergence might be ill-defined.

*Proof.* We can expand $D_{KL}(q_{1:n} \| p)$ in the following way:

$$D_{KL}(q_{1:n} \| p) := \int_{\mathcal{Z}^n} q_{1:n}(\mathbf{z}_1, \ldots, \mathbf{z}_n) \log \frac{q_{1:n}(\mathbf{z}_1, \ldots, \mathbf{z}_n)}{p(\mathbf{z}_1, \ldots, \mathbf{z}_n)} d\mathbf{z}_{1:n} \tag{25}$$

$$= \int_{\mathcal{Z}^n} q_{1:n}(\mathbf{z}_1, \ldots, \mathbf{z}_n) \left( \log \frac{q_{1:n}(\mathbf{z}_1, \ldots, \mathbf{z}_n)}{\hat{q}_{1:n}(\mathbf{z}_1, \ldots, \mathbf{z}_n)} + \log \frac{\hat{q}_{1:n}(\mathbf{z}_1, \ldots, \mathbf{z}_n)}{p(\mathbf{z}_1, \ldots, \mathbf{z}_n)} \right) d\mathbf{z}_{1:n} \tag{26}$$

$$= D_{KL}(q_{1:n} \| \hat{q}_{1:n}) + \int_{\mathcal{Z}^n} q_{1:n}(\mathbf{z}_1, \ldots, \mathbf{z}_n) \log \frac{\hat{q}_{1:n}(\mathbf{z}_1, \ldots, \mathbf{z}_n)}{p(\mathbf{z}_1, \ldots, \mathbf{z}_n)} d\mathbf{z}_{1:n} \tag{27}$$

*(introduce average over all permutations $\Pi_n$)* $\tag{28}$

$$= D_{KL}(q_{1:n} \| \hat{q}_{1:n}) + \frac{1}{n!} \sum_{\pi \in \Pi_n} \int_{\mathcal{Z}^n} q_{1:n}(\mathbf{z}_1, \ldots, \mathbf{z}_n) \log \frac{\hat{q}_{1:n}(\mathbf{z}_1, \ldots, \mathbf{z}_n)}{p(\mathbf{z}_1, \ldots, \mathbf{z}_n)} d\mathbf{z}_{1:n} \tag{29}$$

*(apply change of variables $(\mathbf{z}_1, \ldots, \mathbf{z}_n) \mapsto (\mathbf{z}_{\pi(1)}, \ldots, \mathbf{z}_{\pi(n)})$ within inner summation)* $\tag{30}$

$$= D_{KL}(q_{1:n} \| \hat{q}_{1:n}) + \frac{1}{n!} \sum_{\pi \in \Pi_n} \int_{\mathcal{Z}^n} q_{1:n}(\mathbf{z}_{\pi(1)}, \ldots, \mathbf{z}_{\pi(n)}) \log \frac{\hat{q}_{1:n}(\mathbf{z}_{\pi(1)}, \ldots, \mathbf{z}_{\pi(n)})}{p(\mathbf{z}_{\pi(1)}, \ldots, \mathbf{z}_{\pi(n)})} d\mathbf{z}_{1:n} \tag{31}$$

$$= D_{KL}(q_{1:n} \| \hat{q}_{1:n}) + \frac{1}{n!} \sum_{\pi \in \Pi_n} \int_{\mathcal{Z}^n} q_{1:n}(\mathbf{z}_{\pi(1)}, \ldots, \mathbf{z}_{\pi(n)}) \log \frac{\hat{q}_{1:n}(\mathbf{z}_1, \ldots, \mathbf{z}_n)}{p(\mathbf{z}_1, \ldots, \mathbf{z}_n)} d\mathbf{z}_{1:n} \tag{32}$$

$$= D_{KL}(q_{1:n} \| \hat{q}_{1:n}) + \int_{\mathcal{Z}^n} \left( \frac{1}{n!} \sum_{\pi \in \Pi_n} q_{1:n}(\mathbf{z}_{\pi(1)}, \ldots, \mathbf{z}_{\pi(n)}) \right) \log \frac{\hat{q}_{1:n}(\mathbf{z}_1, \ldots, \mathbf{z}_n)}{p(\mathbf{z}_1, \ldots, \mathbf{z}_n)} d\mathbf{z}_{1:n} \tag{33}$$

$$= D_{KL}(q_{1:n} \| \hat{q}_{1:n}) + \int_{\mathcal{Z}^n} \hat{q}_{1:n}(\mathbf{z}_1, \ldots, \mathbf{z}_n) \log \frac{\hat{q}_{1:n}(\mathbf{z}_1, \ldots, \mathbf{z}_n)}{p(\mathbf{z}_1, \ldots, \mathbf{z}_n)} d\mathbf{z}_{1:n} \tag{34}$$

$$= D_{KL}(q_{1:n} \| \hat{q}_{1:n}) + D_{KL}(\hat{q}_{1:n} \| p) \tag{35}$$

where the equality between Equation 31 and Equation 32 follows from both $\hat{q}_{1:n}$ and $p$ being exchangeable. Clearly, by known properties of the KL-divergence, the gap $D_{KL}(q_{1:n} \| \hat{q}_{1:n})$ will be 0 if and only if the two distributions are the same. The distributions will only be the same if $q_{1:n}$ is already exchangeable. This completes the proof. $\square$

### C.1.2. PROOF OF PROPOSITION 4.2

We re-state Proposition 4.2 before providing the proof.

**Proposition.** *Let the true data-generating distribution $p$ also be conditionally exchangeable, i.e., for the true conditional data-generating distribution $p_{Y_{1:n}|X_{1:n}} : \mathcal{Y}^n \times \mathcal{X}^n \to \mathbb{R}$ the following holds:*

$$p(\mathbf{y}_{1:n} \mid \mathbf{x}_{1:n}) = p(\mathbf{y}_{\pi_{1:n}} \mid \mathbf{x}_{\pi_{1:n}}) \quad \forall \pi, \mathbf{x}_{1:n}, \mathbf{y}_{1:n}, \tag{36}$$

*and the true marginal data-generating distribution over inputs $p_{X_{1:n}} : \mathcal{X}^n \to \mathbb{R}$ be i.i.d. Next, for any conditional density $q_{Y_{1:n}|X_{1:n}} : \mathcal{Y}^n \times \mathcal{X}^n \to \mathbb{R}$, obtain the exchangeabilified version of it as*

$$\hat{q}_{Y_{1:n}|X_{1:n}}(\mathbf{y}_{1:n}|\mathbf{x}_{1:n}) = \frac{1}{n!} \sum_{\pi \in \Pi_n} q_{Y_{1:n}|X_{1:n}}(\mathbf{y}_{\pi_{1:n}}|\mathbf{x}_{\pi_{1:n}}). \tag{37}$$

*Then, if the support of $p$ is within the support of $q$, the following decomposition holds:*

$$\mathbb{E}_{p_{X_{1:n}}} \left[ D_{KL} \left[ q_{Y_{1:n}|X_{1:n}} \| p_{Y_{1:n}|X_{1:n}} \right] \right] = \mathbb{E}_{p_{X_{1:n}}} \left[ D_{KL} \left[ \hat{q}_{Y_{1:n}|X_{1:n}} \| p_{Y_{1:n}|X_{1:n}} \right] \right]$$
$$+ \underbrace{\mathbb{E}_{p_{X_{1:n}}} \left[ D_{KL} \left[ q_{Y_{1:n}|X_{1:n}} \| \hat{q}_{Y_{1:n}|X_{1:n}} \right] \right]}_{KL\ Gap} \tag{38}$$

*and, in particular, $\mathbb{E}_{p_{X_{1:n}}} \left[ D_{KL} \left[ q_{Y_{1:n}|X_{1:n}} \| p_{Y_{1:n}|X_{1:n}} \right] \right] \geq \mathbb{E}_{p_{X_{1:n}}} \left[ D_{KL} \left[ \hat{q}_{Y_{1:n}|X_{1:n}} \| p_{Y_{1:n}|X_{1:n}} \right] \right]$ with equality if and only if $q_{Y_{1:n}|X_{1:n}}$ is conditionally exchangeable.*

*Proof.* We begin by expanding the expected Kullback-Leibler divergence by writing out its integral definition:

$$
\mathbb{E}_{X_{1:n} \sim p_{X_{1:n}}} \left[ \mathrm{D_{KL}} \left[ q_{Y_{1:n}|X_{1:n}}(\cdot|X_{1:n}) \mid p_{Y_{1:n}|X_{1:n}}(\cdot|X_{1:n}) \right] \right]
$$

$$
= \int_{\mathcal{X}^n} p_{X_{1:n}}(\mathbf{x}_{1:n}) \int_{\mathcal{Y}^n} q_{Y_{1:n}|X_{1:n}}(\mathbf{y}_{1:n}|\mathbf{x}_{1:n}) \log \frac{q_{Y_{1:n}|X_{1:n}}(\mathbf{y}_{1:n}|\mathbf{x}_{1:n})}{p_{Y_{1:n}|X_{1:n}}(\mathbf{y}_{1:n}|\mathbf{x}_{1:n})} \mathrm{d}\mathbf{y}_{1:n} \mathrm{d}\mathbf{x}_{1:n}
$$

*(multiply and divide by the exchangeable-ified density $\hat{q}_{Y_{1:n}|X_{1:n}}$ inside the logarithm)*

$$
= \int_{\mathcal{X}^n} p_{X_{1:n}}(\mathbf{x}_{1:n}) \int_{\mathcal{Y}^n} q_{Y_{1:n}|X_{1:n}}(\mathbf{y}_{1:n}|\mathbf{x}_{1:n}) \left( \log \frac{q_{Y_{1:n}|X_{1:n}}(\mathbf{y}_{1:n}|\mathbf{x}_{1:n})}{\hat{q}_{Y_{1:n}|X_{1:n}}(\mathbf{y}_{1:n}|\mathbf{x}_{1:n})} \right.
$$

$$
\left. + \log \frac{\hat{q}_{Y_{1:n}|X_{1:n}}(\mathbf{y}_{1:n}|\mathbf{x}_{1:n})}{p_{Y_{1:n}|X_{1:n}}(\mathbf{y}_{1:n}|\mathbf{x}_{1:n})} \right) \mathrm{d}\mathbf{y}_{1:n} \mathrm{d}\mathbf{x}_{1:n}
$$

$$
= \mathbb{E}_{X_{1:n} \sim p_{X_{1:n}}} \left[ \mathrm{D_{KL}} \left[ q_{Y_{1:n}|X_{1:n}}(\cdot|X_{1:n}) \mid \hat{q}_{Y_{1:n}|X_{1:n}}(\cdot|X_{1:n}) \right] \right]
$$

$$
+ \int_{\mathcal{X}^n} p_{X_{1:n}}(\mathbf{x}_{1:n}) \int_{\mathcal{Y}^n} q_{Y_{1:n}|X_{1:n}}(\mathbf{y}_{1:n}|\mathbf{x}_{1:n}) \log \frac{\hat{q}_{Y_{1:n}|X_{1:n}}(\mathbf{y}_{1:n}|\mathbf{x}_{1:n})}{p_{Y_{1:n}|X_{1:n}}(\mathbf{y}_{1:n}|\mathbf{x}_{1:n})} \mathrm{d}\mathbf{y}_{1:n} \mathrm{d}\mathbf{x}_{1:n} \tag{39}
$$

The first term in Equation 39 is exactly the conditional exchangeability gap from the proposition. We now focus on the second integral term.

By introducing an average over all permutations $\Pi_n$, which does not change the value of the integral (as it's a scalar with respect to $\pi \in \Pi_n$), we obtain:

$$
\frac{1}{n!} \sum_{\pi \in \Pi_n} \int_{\mathcal{X}^n} \int_{\mathcal{Y}^n} p_{X_{1:n}}(\mathbf{x}_{1:n}) q_{Y_{1:n}|X_{1:n}}(\mathbf{y}_{1:n}|\mathbf{x}_{1:n}) \log \frac{\hat{q}_{Y_{1:n}|X_{1:n}}(\mathbf{y}_{1:n}|\mathbf{x}_{1:n})}{p_{Y_{1:n}|X_{1:n}}(\mathbf{y}_{1:n}|\mathbf{x}_{1:n})} \mathrm{d}\mathbf{y}_{1:n} \mathrm{d}\mathbf{x}_{1:n}
$$

*(apply change of variables $(\mathbf{x}_{1:n}, \mathbf{y}_{1:n}) \mapsto (\mathbf{x}_{\pi(1:n)}, \mathbf{y}_{\pi(1:n)})$ within the summation)*

$$
= \frac{1}{n!} \sum_{\pi \in \Pi_n} \int_{\mathcal{X}^n} \int_{\mathcal{Y}^n} p_{X_{1:n}}(\mathbf{x}_{\pi(1:n)}) q_{Y_{1:n}|X_{1:n}}(\mathbf{y}_{\pi(1:n)}|\mathbf{x}_{\pi(1:n)}) \log \frac{\hat{q}_{Y_{1:n}|X_{1:n}}(\mathbf{y}_{\pi(1:n)}|\mathbf{x}_{\pi(1:n)})}{p_{Y_{1:n}|X_{1:n}}(\mathbf{y}_{\pi(1:n)}|\mathbf{x}_{\pi(1:n)})} \mathrm{d}\mathbf{y}_{1:n} \mathrm{d}\mathbf{x}_{1:n} \tag{40}
$$

Because both $\hat{q}_{Y_{1:n}|X_{1:n}}$ and $p_{Y_{1:n}|X_{1:n}}$ are conditionally exchangeable, they are invariant to joint permutations of their inputs and outputs. Furthermore, since $p_{X_{1:n}}$ is i.i.d. it is also exchangeable, $p_{X_{1:n}}(\mathbf{x}_{\pi(1:n)}) = p_{X_{1:n}}(\mathbf{x}_{1:n})$. Therefore, we can freely remove the permutations from these terms:

$$
= \frac{1}{n!} \sum_{\pi \in \Pi_n} \int_{\mathcal{X}^n} \int_{\mathcal{Y}^n} p_{X_{1:n}}(\mathbf{x}_{1:n}) q_{Y_{1:n}|X_{1:n}}(\mathbf{y}_{\pi(1:n)}|\mathbf{x}_{\pi(1:n)}) \log \frac{\hat{q}_{Y_{1:n}|X_{1:n}}(\mathbf{y}_{1:n}|\mathbf{x}_{1:n})}{p_{Y_{1:n}|X_{1:n}}(\mathbf{y}_{1:n}|\mathbf{x}_{1:n})} \mathrm{d}\mathbf{y}_{1:n} \mathrm{d}\mathbf{x}_{1:n} \tag{41}
$$

*(pull the summation inside the integral)*

$$
= \int_{\mathcal{X}^n} \int_{\mathcal{Y}^n} \left( p_{X_{1:n}}(\mathbf{x}_{1:n}) \frac{1}{n!} \sum_{\pi \in \Pi_n} q_{Y_{1:n}|X_{1:n}}(\mathbf{y}_{\pi(1:n)}|\mathbf{x}_{\pi(1:n)}) \right) \log \frac{\hat{q}_{Y_{1:n}|X_{1:n}}(\mathbf{y}_{1:n}|\mathbf{x}_{1:n})}{p_{Y_{1:n}|X_{1:n}}(\mathbf{y}_{1:n}|\mathbf{x}_{1:n})} \mathrm{d}\mathbf{y}_{1:n} \mathrm{d}\mathbf{x}_{1:n}
$$

*(apply the definition of $\hat{q}$)*

$$
= \int_{\mathcal{X}^n} \int_{\mathcal{Y}^n} p_{X_{1:n}}(\mathbf{x}_{1:n}) \hat{q}_{Y_{1:n}|X_{1:n}}(\mathbf{y}_{1:n}|\mathbf{x}_{1:n}) \log \frac{\hat{q}_{Y_{1:n}|X_{1:n}}(\mathbf{y}_{1:n}|\mathbf{x}_{1:n})}{p_{Y_{1:n}|X_{1:n}}(\mathbf{y}_{1:n}|\mathbf{x}_{1:n})} \mathrm{d}\mathbf{y}_{1:n} \mathrm{d}\mathbf{x}_{1:n}
$$

$$
= \mathbb{E}_{X_{1:n} \sim p_{X_{1:n}}} \left[ \mathrm{D_{KL}} \left[ \hat{q}_{Y_{1:n}|X_{1:n}}(\cdot|X_{1:n}) \mid p_{Y_{1:n}|X_{1:n}}(\cdot|X_{1:n}) \right] \right] \tag{42}
$$

Substituting Equation 42 into Equation 39, we arrive exactly at the stated decomposition. Clearly, by known properties of the KL-divergence, the gap $\mathbb{E}_{p_{X_{1:n}}} \left[ \mathrm{D_{KL}} \left[ q_{Y_{1:n}|X_{1:n}} \parallel \hat{q}_{Y_{1:n}|X_{1:n}} \right] \right]$ will be 0 if and only if the two conditional distributions are the same, in which case $q_{Y_{1:n}|X_{1:n}}$ will already be conditionally exchangeable. $\square$

## C.2. Exchangeability gap for non-*i.i.d.* data

Although this paper is primarily concerned with the case of regression data with *i.i.d.* inputs, we believe some readers might be interested in whether the framework applies in a non-*i.i.d.* inputs case as well. In this section, we answer this question in

the affirmative with the proposition below. In particular, we show a similar decomposition of the model's performance into a performance of a conditionally exchangeable predictor and an exchangeability gap, albeit the "exchangeabilified" predictor takes on a slightly weirder form.

**Proposition C.1.** *(Conditional exchangeability gap). Let $p_{X_1,\dots,X_n} : \mathcal{X}^n \to \mathbb{R}$ be a density (interpreted as a density for the 'true data generating' distribution for the 'inputs') on the sequence space $\mathcal{X}^n$. Let $q_{Y_{1:n}|X_{1:n}} : \mathcal{Y}^n \times \mathcal{X}^n \to \mathbb{R}$ be any conditional density on 'targets' $\mathcal{Y}^n$. Define the exchangeable-ified conditional density as:*

$$\hat{q}(y_{1:n}|x_{1:n}) := \frac{1}{n!} \sum_{\pi \in \Pi_n} q_{Y_{1:n}|X_{1:n}}(y_{\pi(1:n)}|x_{\pi(1:n)}) \frac{p_{X_{1:n}}(x_{\pi(1:n)})}{\hat{p}_{X_{1:n}}(x_{1:n})}, \quad where \; \hat{p}(x_{1:n}) := \frac{1}{n!} \sum_{\pi \in \Pi_n} p_{X_{1:n}}(x_{\pi(1:n)}),$$

*and where $\Pi_n$ is the space of all permutations of $n$ elements. $\hat{q}$ is clearly a conditional density and is conditionally exchangeable.*

*Then, for any conditionally exchangeable density $p_{Y_{1:n}|X_{1:n}}$ (as a reminder, a conditional density is conditionally exchangeable if $p_{Y_{1:n}|X_{1:n}}(y_{1:n}|x_{1:n}) = p_{Y_{1:n}|X_{1:n}}(y_{\pi(1:n)}|x_{\pi(1:n)})$ for all $\pi, x_{1:n}, y_{1:n}$), the following decomposition holds:*

$$\mathbb{E}_{X_{1:n} \sim p_{X_{1:n}}} \left[ D_{KL} \left[ q_{Y_{1:n}|X_{1:n}}(\cdot|X_{1:n}) \mid p_{Y_{1:n}|X_{1:n}}(\cdot|X_{1:n}) \right] \right] =$$
$$= \mathbb{E}_{X_{1:n} \sim p_{X_{1:n}}} \left[ D_{KL} \left[ \hat{q}_{Y_{1:n}|X_{1:n}}(\cdot|X_{1:n}) \mid p_{Y_{1:n}|X_{1:n}}(\cdot|X_{1:n}) \right] \right]$$
$$+ \underbrace{\mathbb{E}_{X_{1:n} \sim p_{X_{1:n}}} \left[ D_{KL} \left[ q_{Y_{1:n}|X_{1:n}}(\cdot|X_{1:n}) \mid \hat{q}_{Y_{1:n}|X_{1:n}}(\cdot|X_{1:n}) \right] \right]}_{Gap}.$$

*Proof.* We begin by expanding the expected Kullback-Leibler divergence by writing out its integral definition:

$$\mathbb{E}_{X_{1:n} \sim p_{X_{1:n}}} \left[ D_{KL} \left[ q_{Y_{1:n}|X_{1:n}}(\cdot|X_{1:n}) \mid p_{Y_{1:n}|X_{1:n}}(\cdot|X_{1:n}) \right] \right]$$

$$= \int_{\mathcal{X}^n} p_{X_{1:n}}(x_{1:n}) \int_{\mathcal{Y}^n} q_{Y_{1:n}|X_{1:n}}(y_{1:n}|x_{1:n}) \log \frac{q_{Y_{1:n}|X_{1:n}}(y_{1:n}|x_{1:n})}{p_{Y_{1:n}|X_{1:n}}(y_{1:n}|x_{1:n})} \mathrm{d}y_{1:n}\mathrm{d}x_{1:n}$$

*(multiply and divide by the exchangeable-ified density $\hat{q}_{Y_{1:n}|X_{1:n}}$ inside the logarithm)*

$$= \int_{\mathcal{X}^n} p_{X_{1:n}}(x_{1:n}) \int_{\mathcal{Y}^n} q_{Y_{1:n}|X_{1:n}}(y_{1:n}|x_{1:n}) \left( \log \frac{q_{Y_{1:n}|X_{1:n}}(y_{1:n}|x_{1:n})}{\hat{q}_{Y_{1:n}|X_{1:n}}(y_{1:n}|x_{1:n})} + \log \frac{\hat{q}_{Y_{1:n}|X_{1:n}}(y_{1:n}|x_{1:n})}{p_{Y_{1:n}|X_{1:n}}(y_{1:n}|x_{1:n})} \right) \mathrm{d}y_{1:n}\mathrm{d}x_{1:n}$$

$$= \mathbb{E}_{X_{1:n} \sim p_{X_{1:n}}} \left[ D_{KL} \left[ q_{Y_{1:n}|X_{1:n}}(\cdot|X_{1:n}) \mid \hat{q}_{Y_{1:n}|X_{1:n}}(\cdot|X_{1:n}) \right] \right]$$

$$+ \int_{\mathcal{X}^n} p_{X_{1:n}}(x_{1:n}) \int_{\mathcal{Y}^n} q_{Y_{1:n}|X_{1:n}}(y_{1:n}|x_{1:n}) \log \frac{\hat{q}_{Y_{1:n}|X_{1:n}}(y_{1:n}|x_{1:n})}{p_{Y_{1:n}|X_{1:n}}(y_{1:n}|x_{1:n})} \mathrm{d}y_{1:n}\mathrm{d}x_{1:n} \tag{43}$$

The first term in Equation 43 is exactly the conditional exchangeability gap from the proposition. We now focus on the second integral term.

By introducing an average over all permutations $\Pi_n$, which does not change the value of the integral (as it's a scalar with respect to $\pi \in \Pi_n$), we obtain:

$$\frac{1}{n!} \sum_{\pi \in \Pi_n} \int_{\mathcal{X}^n} \int_{\mathcal{Y}^n} p_{X_{1:n}}(x_{1:n}) q_{Y_{1:n}|X_{1:n}}(y_{1:n}|x_{1:n}) \log \frac{\hat{q}_{Y_{1:n}|X_{1:n}}(y_{1:n}|x_{1:n})}{p_{Y_{1:n}|X_{1:n}}(y_{1:n}|x_{1:n})} \mathrm{d}y_{1:n}\mathrm{d}x_{1:n}$$

*(apply change of variables $(x_{1:n}, y_{1:n}) \mapsto (x_{\pi(1:n)}, y_{\pi(1:n)})$ within the summation)*

$$= \frac{1}{n!} \sum_{\pi \in \Pi_n} \int_{\mathcal{X}^n} \int_{\mathcal{Y}^n} p_{X_{1:n}}(x_{\pi(1:n)}) q_{Y_{1:n}|X_{1:n}}(y_{\pi(1:n)}|x_{\pi(1:n)}) \log \frac{\hat{q}_{Y_{1:n}|X_{1:n}}(y_{\pi(1:n)}|x_{\pi(1:n)})}{p_{Y_{1:n}|X_{1:n}}(y_{\pi(1:n)}|x_{\pi(1:n)})} \mathrm{d}y_{1:n}\mathrm{d}x_{1:n} \tag{44}$$

Because both $\hat{q}_{Y_{1:n}|X_{1:n}}$ and $p_{Y_{1:n}|X_{1:n}}$ are conditionally exchangeable, they are invariant to joint permutations of their

inputs and targets. Therefore, we can simplify the argument inside the logarithm:

$$= \frac{1}{n!} \sum_{\pi \in \Pi_n} \int_{\mathcal{X}^n} \int_{\mathcal{Y}^n} p_{X_{1:n}}(x_{\pi(1:n)}) q_{Y_{1:n}|X_{1:n}}(y_{\pi(1:n)}|x_{\pi(1:n)}) \log \frac{\hat{q}_{Y_{1:n}|X_{1:n}}(y_{1:n}|x_{1:n})}{p_{Y_{1:n}|X_{1:n}}(y_{1:n}|x_{1:n})} \mathrm{d}y_{1:n} \mathrm{d}x_{1:n} \tag{45}$$

*(pull the summation inside the integral)*

$$= \int_{\mathcal{X}^n} \int_{\mathcal{Y}^n} \left( \frac{1}{n!} \sum_{\pi \in \Pi_n} p_{X_{1:n}}(x_{\pi(1:n)}) q_{Y_{1:n}|X_{1:n}}(y_{\pi(1:n)}|x_{\pi(1:n)}) \right) \log \frac{\hat{q}_{Y_{1:n}|X_{1:n}}(y_{1:n}|x_{1:n})}{p_{Y_{1:n}|X_{1:n}}(y_{1:n}|x_{1:n})} \mathrm{d}y_{1:n} \mathrm{d}x_{1:n}$$

Notice that, by the definition of $\hat{q}$ and $\hat{p}$ provided in the proposition, we have the equivalence:

$$\frac{1}{n!} \sum_{\pi \in \Pi_n} p_{X_{1:n}}(x_{\pi(1:n)}) q_{Y_{1:n}|X_{1:n}}(y_{\pi(1:n)}|x_{\pi(1:n)}) = \hat{p}_{X_{1:n}}(x_{1:n}) \hat{q}_{Y_{1:n}|X_{1:n}}(y_{1:n}|x_{1:n})$$

Substituting this back into our integral yields:

$$= \int_{\mathcal{X}^n} \int_{\mathcal{Y}^n} \hat{p}_{X_{1:n}}(x_{1:n}) \hat{q}_{Y_{1:n}|X_{1:n}}(y_{1:n}|x_{1:n}) \log \frac{\hat{q}_{Y_{1:n}|X_{1:n}}(y_{1:n}|x_{1:n})}{p_{Y_{1:n}|X_{1:n}}(y_{1:n}|x_{1:n})} \mathrm{d}y_{1:n} \mathrm{d}x_{1:n}$$

$$= \mathbb{E}_{X_{1:n} \sim \hat{p}_{X_{1:n}}} \left[ \mathrm{D}_{\mathrm{KL}} \left[ \hat{q}_{Y_{1:n}|X_{1:n}}(\cdot|X_{1:n}) \mid p_{Y_{1:n}|X_{1:n}}(\cdot|X_{1:n}) \right] \right] \tag{46}$$

To complete the proof, we must show that taking the expectation over $\hat{p}_{X_{1:n}}$ is equivalent to taking it over $p_{X_{1:n}}$. Let $f(x_{1:n}) := \mathrm{D}_{\mathrm{KL}} \left[ \hat{q}_{Y_{1:n}|X_{1:n}}(\cdot|x_{1:n}) \mid p_{Y_{1:n}|X_{1:n}}(\cdot|x_{1:n}) \right]$. Because both distributions in the KL divergence are conditionally exchangeable, the function $f$ evaluates equivalently under permutations: $f(x_{1:n}) = f(x_{\pi(1:n)})$.

Using the definition of $\hat{p}_{X_{1:n}}$, we can rewrite the expectation over $\hat{p}$ back into an expectation over $p$:

$$\int_{\mathcal{X}^n} \hat{p}_{X_{1:n}}(x_{1:n}) f(x_{1:n}) \mathrm{d}x_{1:n} = \int_{\mathcal{X}^n} \left( \frac{1}{n!} \sum_{\pi \in \Pi_n} p_{X_{1:n}}(x_{\pi(1:n)}) \right) f(x_{1:n}) \mathrm{d}x_{1:n}$$

$$= \frac{1}{n!} \sum_{\pi \in \Pi_n} \int_{\mathcal{X}^n} p_{X_{1:n}}(x_{\pi(1:n)}) f(x_{\pi(1:n)}) \mathrm{d}x_{1:n}$$

$$= \int_{\mathcal{X}^n} p_{X_{1:n}}(x_{1:n}) f(x_{1:n}) \mathrm{d}x_{1:n} \tag{47}$$

Applying Equation 47 to Equation 46, and combining it with the gap term from Equation 43, we arrive exactly at the stated decomposition. □

## D. Datasets

### D.1. 1D GP Synthetic Regression

In the main text we present results for models with up to $N_c^{(\mathrm{max})} = 64$ context points, using the RBF kernel. We present additional results in Appendix E.1.1 for more complex kernels, and higher context sizes (up to $N_c^{(\mathrm{max})} = 512$).

**Data generation**    For all experiments, input locations $x$ are sampled uniformly from the domain $[-2, 2]$. Target values are generated as $y = f(x) + \sigma_{\mathrm{obs}}\epsilon$, where observation noise is fixed at $\sigma_{\mathrm{obs}} = 0.1$.

**RBF kernel**    The radial basis function (RBF) kernel is defined as:

$$k_{\mathrm{rbf}}(x, x') = \exp\left( -\frac{(x - x')^2}{2\ell^2} \right). \tag{48}$$

To span a sensible range of complexity, we use logarithmic uniform sampling to sample the lengthscale $\ell$ such that $\log_{10}(\ell) \sim \mathcal{U}[\log_{10}(\ell_{\min}), \log_{10}(\ell_{\max})]$. We use $\ell_{\min} = 0.25$ and $\ell_{\max} = 1$.

**Matérn kernels** The Matérn kernel is a generalisation of the RBF kernel, with an additional smoothness parameter $\nu$. Within this work, kernels $\nu \in \{\frac{1}{2}, \frac{3}{2}, \frac{5}{2}\}$ are considered, referred to as *Matérn 1/2*, *Matérn 3/2* and *Matérn 5/2*. They vary in complexity and are given by:

$$k_{\text{m1/2}}(x, x') = \exp\left(-\frac{|x - x'|}{\ell}\right), \tag{49}$$

$$k_{\text{m3/2}}(x, x') = \left(1 + \frac{\sqrt{3}|x - x'|}{\ell}\right) \exp\left(-\frac{\sqrt{3}|x - x'|}{\ell}\right), \tag{50}$$

$$k_{\text{m5/2}}(x, x') = \left(1 + \frac{\sqrt{5}|x - x'|}{\ell} + \frac{5(x - x')^2}{3\ell^2}\right) \exp\left(-\frac{\sqrt{5}|x - x'|}{\ell}\right). \tag{51}$$

We sample $\ell$ using the same methodology as for the RBF kernel.

**Periodic kernel** The periodic kernel models periodic data by allowing the correlation between points to cycle over a period $p$, and is defined as:

$$k_{\text{per}}(x, x') = \exp\left(-2\frac{\sin^2(\pi|x - x'|/p)}{\ell^2}\right). \tag{52}$$

We use $p = 2$, allowing for two complete oscillations over the domain, with $\ell$ being sampled in the same fashion as for the RBF kernel.

**Mixed kernel** To introduce additional complexity, we train models using a mixture of GP kernels. In this *mixed kernel* setting, we randomly sample a stationary kernel family $k$ from the set $\{k_{\text{rbf}}, k_{\text{m1/2}}, k_{\text{m3/2}}, k_{\text{m5/2}}, k_{\text{per}}\}$ for each training batch, before sampling the specific hyperparameters. This requires the NP to implicitly infer not only the kernel parameters but also the underlying kernel family itself for each task.

### D.2. Tabular Data

**Synthetic prior** To encourage generalisation across diverse tabular distributions, we generate synthetic datasets using a structural prior adapted from the TabPFN v1.0 synthetic data pipeline (Hollmann et al., 2025)[6]. For each dataset instance, we sample hyperparameters and instantiate a randomly-weighted MLP-based generator, yielding input features $\mathbf{x}$ and regression targets $\mathbf{y}$. The MLP depth is sampled from a truncated normal distribution with a minimum depth of 2, with log-uniformly sampled mean and standard deviation in the range $[1, 6]$. The hidden width is sampled analogously with log-uniform mean/standard deviation in the range $[5, 130]$ and a minimum width of 4. Each MLP randomly selects an activation function from the set $\{\text{ReLU}, \text{Tanh}, \text{Sigmoid}, \text{ELU}, \text{Identity}\}$.

To standardize input dimensionality, we Z-score normalise features per dataset and zero-pad feature vectors to a fixed dimension $D_x = 20$. For regression targets, we first clip outliers using an interquartile range filter with factor $k = 3.0$, then apply a randomized normalisation chosen uniformly from Z-score, Min-Max, Max-Abs, or Robust scaling to induce robustness to varying output scales.

**Testing protocol** When testing on synthetically generated data, we test over 80,000 tasks with $N_c \sim \mathcal{U}[10, 1024]$ and $N_t = 128$. For real-world datasets of length $N$, we randomly sample $N_c \sim \mathcal{U}[0.5N, 0.8N]$ context points and use the remaining points as the targets $N_t = N - N_c$, averaging over 50 such tasks from each dataset. When evaluating model performance on real-world datasets in a streaming setting, we evaluate on $N_t = 0.2N$ randomly selected target points and use the remaining points as contextual information to be streamed, averaging our results across 10 random dataset permutations. We consider the Skillcraft, Powerplant, Elevator and Protein datasets from the UCI Machine Learning Repository for real-world tabular evaluation. We indicate the total number of datapoints $N$ in brackets for each UCI dataset.

---

[6]TabPFN's pipeline is available at https://github.com/PriorLabs/TabPFN/tree/v1.0.0/tabpfn/priors

**Skillcraft** ($N = 3338$) SkillCraft1 Master Table is a dataset of player rankings on the StarCraft 2 video game. We use this ranking (*LeagueIndex*) as our target output. We drop the *GameID* column (which is an index column) and log-transform all other features.

**Powerplant** ($N = 9568$) The Combined Cycle Power Plant dataset records the net hourly electrical output of a powerplant alongside ambient environmental features for that point in time. We use the hourly electrical output as our target output (which maps to the last column) and all other columns for our input features.

**Elevator** ($N = 16599$) For the Elevators dataset, a dataset monitoring the control of an aircraft elevator, we use the last column as our target and remove 2 feature columns that have a standard deviation below $1 \times 10^{-5}$.

**Protein** ($N = 44019$) The Physicochemical Properties of Protein Tertiary Structure is a large scale scientific dataset detailing protein decoy structures. For this task, we use the last column as the target column and drop all duplicate entries.

**Normalisation** We Z-normalise both features and targets, and clip all values to be in the range $[-5, 5]$. In the static context evaluation setting, we compute normalisation statistics on the context dataset and apply normalisation to both the target and context sets. For the streaming setting, we compute normalisation statistics on a calibration set of size $\max(200, 0.2N_c)$ which is excluded from evaluation and then stream the remaining data. We continuously update feature normalisation statistics in an online fashion as the stream arrives.

### D.3. Station Temperature Prediction

The HadISD dataset (Dunn et al., 2012; 2016) is a quality-controlled subset of the Integrated Surface Database (ISD) (Smith et al., 2011), comprising in situ observations from 9,948 weather stations. Key recorded variables include temperature, dewpoint temperature, cloud cover and wind speed. Due to computational constraints, we focus on the dry bulb air temperature at a 2-metre screen height across 2,805 stations situated in or near Africa and Europe. This corresponds to a latitude and longitude range of $[-20°, 60°]_\phi \times [-10°, 52°]_\lambda$.

As an off-the-grid meteorological dataset, station density varies geographically; weather station observations occur relatively sparsely in Africa, and much more frequently in Europe as seen in Figure 9, adding to the task complexity.

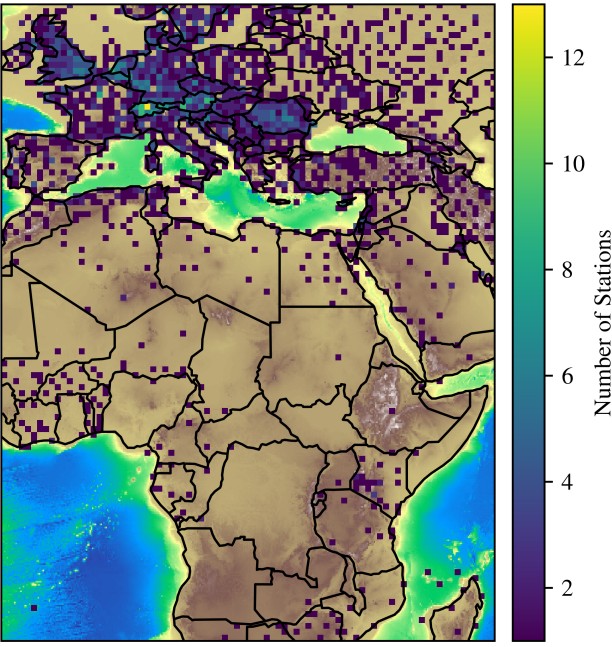

*Figure 9.* Distribution of the $2,805$ stations across the restricted latitude and longitude used for our station temperature prediction task, using $0.75° \times 0.75°$ patches.

**Data split**  We train our models on observations from 1980 to 2016, validate model performance on data from 2017, and test model performance on data from 2018–2019.

**Normalisation**  Latitude and longitude features are normalised to be in the range $[-1, 1]$, and we apply Z-normalisation to temperature and elevation. All normalisation statistics are computed on the training dataset.

**Spatiotemporal interpolation**  We train all NP models on a spatiotemporal interpolation task. Given a random subset of contextual observations at $H = 8$ windows separated by $\delta = 6$ hours, we train NP models to predict the temperature at randomly sampled locations across the complete time window. This corresponds to a total window of 48 hours. This training task forces NP models to learn complex underlying spatiotemporal temperature patterns, informing a foundational representation of weather modelling. Moreover, the spatiotemporal interpolation is a task of practical interest of its own, such as when having to estimate true readings at faulty or sparse sensors.

**Forecasting**  We test the ability of our NP models to generalise to weather forecasting, having only been trained on the markedly different task of spatiotemporal interpolation. For forecasting we take $H = 8$ windows separated by $\delta = 6$ hours and sample $N_c$ temporally ordered observations from the first $H - 1$ windows as our context set, leaving $N_t$ stations from the last time window to compose our target set.

## E. Additional Experimental Results

In addition to the experimental results provided in Section 5, here we present further evidence for the claims that we make in the main part of the paper. We also provide the results of further empirical investigations that we conduct into the performance of our models in certain scenarios.

### E.1. Static Context Performance

*Table 2.* Average Test Log-Likelihoods (↑) for (a) Factorised and (b) AR deployment. We report the absolute performance of the reference non-incremental TNP-D. For all the other methods we report the difference (Δ) relative to the reference performance. Held-out task are shown in purple, while transfer scenarios, such as Sim-to-Real (Tabular) and Forecasting (Temperature) are coloured in orange. The values indicate absolute or relative performance ± the standard error of the mean (SEM) of each method.

*(a)* **Factorised predictions**. `incTNP-Seq` achieves parity or outperforms `TNP-D` on held-out tasks, and dominates on transfer scenarios.

| | REFERENCE | OURS (Δ (↑) RELATIVE TO REF.) | | BASELINES (Δ (↑) RELATIVE TO REF.) | |
| --- | --- | --- | --- | --- | --- |
| **DATASET** | **TNP-D** | **INCTNP** | **INCTNP-SEQ** | **CNP** | **LBANP** |
| 1D GP | $0.4309 \pm 0.0019$ | $-0.0133 \pm 0.0019$ | $-0.0016 \pm 0.0019$ | $-0.2301 \pm 0.0021$ | $+0.0040 \pm 0.0019$ |
| TABULAR (SYNTHETIC) | $0.1540 \pm 0.0047$ | $-0.0201 \pm 0.0046$ | $+0.0075 \pm 0.0048$ | $-0.3304 \pm 0.0038$ | $-0.0576 \pm 0.0045$ |
| SKILLCRAFT | $-0.9543 \pm 0.0002$ | $+0.0025 \pm 0.0002$ | $+0.0078 \pm 0.0002$ | $-0.1336 \pm 0.0002$ | $-0.0310 \pm 0.0002$ |
| POWERPLANT | $-0.0076 \pm 0.0002$ | $+0.0022 \pm 0.0002$ | $+0.0028 \pm 0.0002$ | $-0.2692 \pm 0.0002$ | $-0.0244 \pm 0.0001$ |
| ELEVATORS | $-0.3219 \pm 0.0003$ | $-0.0235 \pm 0.0003$ | $-0.0086 \pm 0.0004$ | $-0.9417 \pm 0.0002$ | $-0.2048 \pm 0.0003$ |
| PROTEIN | $-1.1524 \pm 0.0001$ | $-0.0281 \pm 0.0001$ | $+0.0363 \pm 0.0001$ | $-0.1877 \pm 0.0000$ | $-0.0239 \pm 0.0001$ |
| HADISD (INTERP) | $-1.7025 \pm 0.0023$ | $-0.0112 \pm 0.0023$ | $+0.0182 \pm 0.0022$ | $-0.5331 \pm 0.0007$ | $-0.0904 \pm 0.0017$ |
| HADISD (FORECAST) | $-2.5708 \pm 0.0031$ | $+0.0302 \pm 0.0039$ | $+0.6897 \pm 0.0019$ | $+0.1806 \pm 0.0011$ | $+0.2681 \pm 0.0018$ |

*(b)* **AR predictions.**

| **DATASET** | **TNP-D** (REF) | **INCTNP** | **INCTNP-SEQ** | **CNP** | **LBANP** | **TNP-A** |
| --- | --- | --- | --- | --- | --- | --- |
| 1D GP | $0.767 \pm 0.003$ | $-0.007 \pm 0.003$ | $+0.002 \pm 0.003$ | $-0.230 \pm 0.011$ | $-0.001 \pm 0.003$ | $+0.004 \pm 0.003$ |
| HADISD (INTERP) | $-1.670 \pm 0.007$ | $-0.038 \pm 0.009$ | $+0.016 \pm 0.007$ | $-0.552 \pm 0.002$ | $-0.097 \pm 0.005$ | $+0.031 \pm 0.008$ |
| HADISD (FORECAST) | $-1.749 \pm 0.003$ | $-0.005 \pm 0.003$ | $+0.061 \pm 0.004$ | $-0.559 \pm 0.004$ | $-0.120 \pm 0.005$ | $+0.035 \pm 0.003$ |

In this task we evaluate the performance of the models when evaluated in the static context setting. This is to establish whether the causal masking introduced in the `incTNP` variants leads to significant performance drop when evaluated in the standard NP evaluation setting. For the 1D GP and Tabular (Synthetic) tasks the results are obtained over 80000 datasets sampled from the prior. For the real-world tabular tasks, the results are shown over 100 permutations of the datasets. For HadISD, factorised results are averaged over 80000 random samples whilst AR results are averaged over 4096 samples.

In Table 2, we expand on the results from Table 1 by including the standard error of the mean (SEM), providing a measure

of the variability across test datasets. In one of the only two settings for which `incTNP-Seq` is outperformed by `TNP-D`, the 1D GP task, the performance difference is within an SEM of `incTNP-Seq`'s performance. Otherwise, the variability is small enough to demonstrate that, when our models are the top performers, it is not an artefact of randomness.

### E.1.1. 1D GP SYNTHETIC REGRESSION

**Additional results**    We provide additional results for larger context set sizes ($N_c$ up to 512 during training and evaluation) as well as on more complicated kernels (mix of 5 kernels), to show that the conclusions from the main are indeed supported in a variety of experimental settings. These results are shown in Figure 10. In particular, `incTNP-Seq` continues to be the best-performing variant, followed by `TNP-D` and then `incTNP`, while also benefitting from robustness to hyperparameter setting. The differences between `incTNP-Seq` and `TNP-D` become even more pronounced with harder tasks.

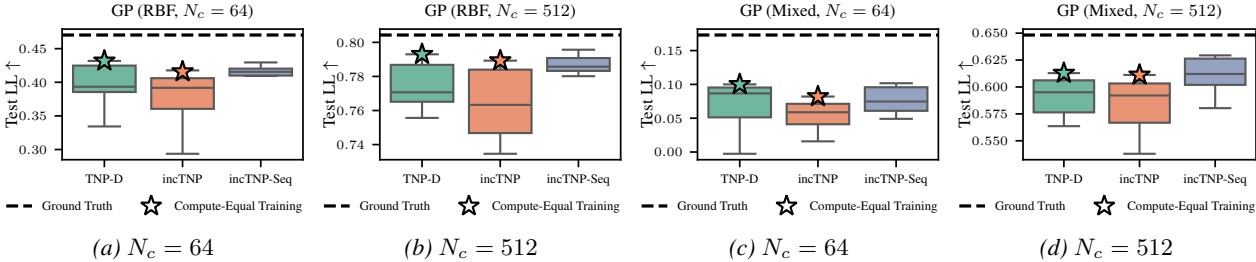

*Figure 10.* Mean Log-Likelihood boxplots for RBF and Mixed kernels and for different context sizes ($N_c$).

**Fairness of compute budgets**    For every task, we also train variants of `TNP-D` and `incTNP` where their training time is matched (i.e., increased) to that of `incTNP-Seq`. This was done to eliminate any bias towards `incTNP-Seq` in our fixed-number-of-epochs setting by it being slightly more compute-heavy an architecture. This benefits the faster-training models since they encounter a greater number of datasets during training than `incTNP-Seq`. In all cases, the results are presented for the best-performing variant within each family. We refer to the matched training-time variants as "compute-equal". Notably, even with compute-equal training, `incTNP-Seq` remains the top-performing variant across our experiments.

**Robustness to hyperparameter choice**    To verify robustness to hyperparameter selection, we iterate over learning rates (LRs) in [0.0001, 0.0003, 0.0005, 0.001, 0.002]. We perform each iteration twice; once with the LR held constant and again with the LR following a cosine annealing schedule (Loshchilov & Hutter, 2017). For each task and model, the best performing variant, LR, and annealing schedule are as follows:

- RBF, $N_c = 64$:
    - TNP-D: 220 epochs (compute-equal), cosine annealing scheduler with initial LR=0.0003
    - incTNP: 200 epochs, cosine annealing scheduler with initial LR=0.0003
    - incTNP-Seq: 200 epochs, cosine annealing scheduler with initial LR=0.0003

- RBF, $N_c = 512$:
    - TNP-D: 250 epochs (compute-equal), cosine annealing scheduler with initial LR=0.0001
    - incTNP: 250 epochs (compute-equal), cosine annealing scheduler with initial LR=0.0003
    - incTNP-Seq: 200 epochs, cosine annealing scheduler with initial LR=0.0003

- Mixed, $N_c = 64$:
    - TNP-D: 440 epochs (compute-equal), cosine annealing scheduler with initial LR=0.0003
    - incTNP: 440 epochs (compute-equal), cosine annealing scheduler with initial LR=0.0003
    - incTNP-Seq: 400 epochs, cosine annealing scheduler with initial LR=0.001

- Mixed, $N_c = 512$:
    - TNP-D: 500 epochs (compute-equal), cosine annealing scheduler with initial LR=0.0003
    - incTNP: 500 epochs (compute-equal), cosine annealing scheduler with initial LR=0.0005
    - incTNP-Seq: 400 epochs, cosine annealing scheduler with initial LR=0.0005

### E.1.2. Synthetic Tabular Data

**Additional results**    As with the synthetic GP data, we conduct a further experiment to investigate the effect of larger context sets in the synthetic tabular data setting. These results can be found in Figure 11. They demonstrate the same performance pattern yet again; `incTNP-Seq` leads and is followed by `TNP-D` and then `incTNP`. `incTNP-Seq`'s narrower boxplot once more demonstrates superior robustness to the hyperparameter setting. The only difference between these results and those in Figure 2 is that the upper bound of performance is somewhat increased in this setting. This is to be expected since the models are provided with more context data.

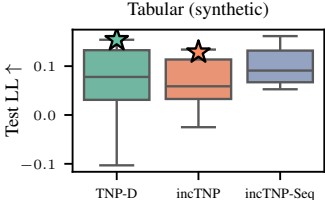

*Figure 11.* Mean Log-Likelihood boxplots for tabular data prediction with $N_c$ up to 1024.

As with the GP regression task, we perform a learning rate sweep across both a fixed learning rate and cosine learning rate scheduler, searching across (initial) learning rates in $[0.0001, 0.0003, 0.0005, 0.001, 0.003, 0.005]$. We also compare against compute-equal variants of `TNP-D` and `incTNP`.

The best trained variants were found to be:

- TNP-D: 680 epochs (compute-equal), cosine annealing scheduler with initial LR=0.003.

- incTNP: 500 epochs, cosine annealing scheduler with initial LR=0.005.

- incTNP-Seq: 500 epochs, cosine annealing scheduler with initial LR=0.003.

### E.1.3. Hyperparameter Robustness Analysis

**Outlier Filtering**    When conducting our hyperparameter sensitivity analysis in Figure 2, we use Seaborn's default filtering functionality and filter out training runs that fail to converge. Figure 12 shows our hyperparameter robustness analysis without any outlier filtering. In the GP case, `incTNP` and `TNP-D` fail to converge when using the largest learning rate without a scheduler, and `incTNP` fails to converge without a scheduler when using the largest learning rate for the tabular setting.

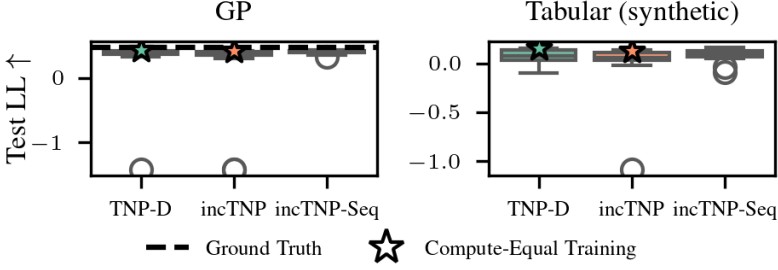

*Figure 12.* Test log-likelihoods ($\uparrow$) on synthetic GP (RBF kernel, $N_c$ up to 64) and Tabular datasets across multiple training configurations without any outlier removal. The highest learning rate we used led to failure of some models (`TNP-D` and `incTNP` in the GP case, and `incTNP` in the tabular case) to converge, resulting in outliers.

**Measured Gradient Variance**    `incTNP-Seq`'s autoregressive training strategy leads to a reduced sensitivity to hyperparameter choice (as shown in Figure 2). We hypothesise that this strategy results in lower gradient variance, with implicit regularisation from the denser supervision signal and a smoother loss landscape also potentially contributing to hyperparameter robustness. To probe this, we measure the gradient-noise-to-signal ratio (GNSR) for both `incTNP` and `incTNP-Seq` during GP training in Figure 13. Specifically, we compute gradients on a fixed collection of mini-batches,

estimate the signal as the squared norm of the mean gradient, and estimate the noise as the variance of the mini-batch gradients around this mean. `incTNP-Seq` exhibits lower GNSR throughout training, providing exploratory evidence to support the hypothesis of reduced training gradient variance.

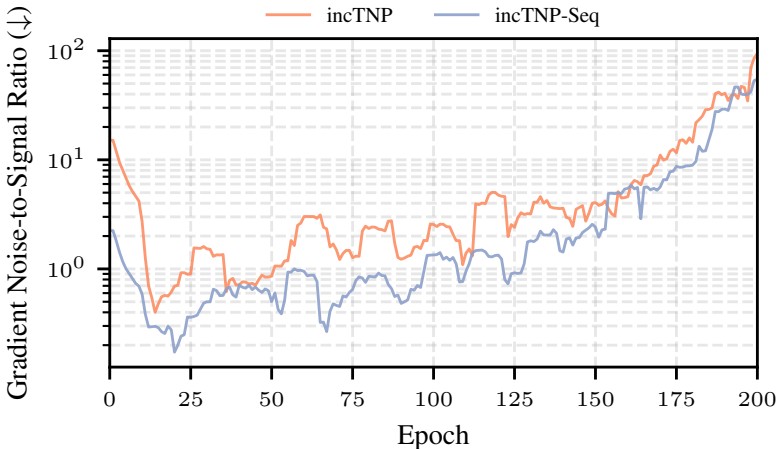

*Figure 13.* The gradient noise-to-signal ratio (GNSR) for `incTNP` and `incTNP-Seq` as training progresses on the RBF kernel with $N_c^{(\max)} = 64$. For each checkpoint, we compute gradients over a fixed collection of mini-batches, estimate the signal as the squared norm of the mean gradient, and estimate the noise as the variance of mini-batch gradients around this mean. We estimate this using 80k samples with batch size 16.

### E.1.4. STATION TEMPERATURE PREDICTION

We train all NP models on the temperature prediction task (HadISD) using $H = 8$ time windows spread $\delta = 6$ hours apart, resulting in a 48 hour training window. For the experiment performed in the main part of the paper, we train the models to perform spatiotemporal interpolation. We then evaluate their performance on the same interpolation scenario, as well as a forecasting scenario in which the targets are sampled strictly from future time windows relative to the context.

To further assess generalisation, here we extend our investigation by evaluating performance over a *24 hour* testing window. We evaluate on spatiotemporal interpolation with factorised predictions (where targets are less correlated), as well as forecasting with AR predictions (where targets are highly correlated, making AR predictions especially suitable). Since models are trained on 48 hour windows, we consider 24 hour window evaluation to be an out-of-distribution setting. Table 3 summarises NP model performance, with our `incTNP` and `incTNP-Seq` matching or exceeding the performance of `TNP-D`, demonstrating robust generalisation capabilities.

*Table 3.* Average Test Log-Likelihoods ($\uparrow$) for (a) Factorised and (b) AR deployment. We report the absolute performance of the reference non-incremental TNP-D. For all the other methods we report the difference ($\Delta$) relative to the reference performance. All tasks are considered out-of-distribution and coloured in orange. The values indicate absolute or relative performance $\pm$ the standard error of the mean (SEM) of each method.

*(a)* **Factorised predictions**. `incTNP-Seq` and `incTNP` outperform or achieve parity with `TNP-D` on OOD interpolation.

| | **REFERENCE** | **OURS** ($\Delta$ ($\uparrow$) RELATIVE TO REF.) | | **BASELINES** ($\Delta$ ($\uparrow$) RELATIVE TO REF.) | |
| --- | --- | --- | --- | --- | --- |
| **TIME WINDOW (INTERP)** | **TNP-D** | **INCTNP** | **INCTNP-SEQ** | **CNP** | **LBANP** |
| 24 HOURS | $-1.6695 \pm 0.0022$ | $+0.0064 \pm 0.0022$ | $+\mathbf{0.0312} \pm \mathbf{0.0022}$ | $-0.5024 \pm 0.0007$ | $-0.0501 \pm 0.0017$ |

*(b)* **AR predictions.** `incTNP` matches `TNP-D`'s performance, whilst `incTNP-Seq` significantly outperforms it.

| **TIME WINDOW (FORECAST)** | **TNP-D** (REF) | **INCTNP** | **INCTNP-SEQ** | **CNP** | **LBANP** | **TNP-A** |
| --- | --- | --- | --- | --- | --- | --- |
| 24 HOURS | $-1.8108 \pm 0.0026$ | $+0.0356 \pm 0.0030$ | $+\mathbf{0.1536} \pm \mathbf{0.0071}$ | $-0.4390 \pm 0.0040$ | $-0.0526 \pm 0.0050$ | $+0.0931 \pm 0.0031$ |

### E.2. Discussion on TNP-A

In addition to `TNP-D`, Nguyen & Grover (2022) introduce `TNP-A`, a TNP variant designed to predict the joint distribution over targets autoregressively. Whilst our primary focus is on the streaming setting rather than pure autoregressive (AR) generation, `TNP-A` represents a significant baseline in the literature that warrants discussion. We report results for `TNP-A` on AR prediction tasks in Table 2(b) and Table 3(b).

`TNP-A` processes a context set $\{x_n^c, y_n^c\}_{n=1}^{N_c}$ and target input locations $\{x_n^t, 0\}_{n=1}^{N_t}$, as well as a sequence of observed targets $\{x_n^t, y_n^t\}_{n=1}^{N_t}$ to model the joint distribution over targets autoregressively. As with `TNP-D`, context points attend to all other context points. However, target input locations and observed targets attend to all context points as well as preceding observed targets (as is visualised in Figure 2 of (Nguyen & Grover, 2022)), inducing a dependency on the ordering of targets. At prediction time, where the true target observations are not available, `TNP-A` samples from its predictive distribution sequentially to build up a set of observations, and averages predictions over multiple sample unrolls $S$.

By allowing unmasked attention between context points, `TNP-A` achieves context permutation invariance but cannot incrementally update its context representation efficiently, incurring the same computational bottleneck as `TNP-D`.

`TNP-A` combines the context set, target locations and target observations into a single sequence that is processed through masked self-attention layers, resulting in an $\mathcal{O}(SN_t(N_c + N_t)^2)$ runtime complexity. We observe that Nguyen & Grover (2022)'s `TNP-A` implementation can be sped up through the use of KV caching to $\mathcal{O}(S(N_c + N_t)^2)$. We term our implementation `TNP-A (Cached)`, and refer to the original as `TNP-A (Original)`. We note that `TNP-A`'s reliance on a unified self-attention stack combined with its hybrid masking strategy makes the implementation of KV caching non-trivial. We also note that runtime comparisons are not fully architecture-controlled: our `TNP-D` implementation interleaves self-attention and cross-attention layers, whereas `TNP-A` uses a unified self-attention stack, resulting in fewer `TNP-A` parameters when the transformer layer count is equalised.

In Figure 14, we explore the AR streaming cost of TNP variants relative to `incTNP` on the HadISD task. We observe that `TNP-A (Original)` is prohibitively expensive to deploy in streaming environments, scaling less favourably than AR `TNP-D`. Conversely, our optimised `TNP-A` model scales more capably. Ultimately, the inability to efficiently incrementally update its context representation means that `TNP-A` is ill-suited for AR streaming tasks, and cannot compete with the computational advantages `incTNP` offers, which are exacerbated for tasks where the frequency of context observations exceeds the number of inference calls.

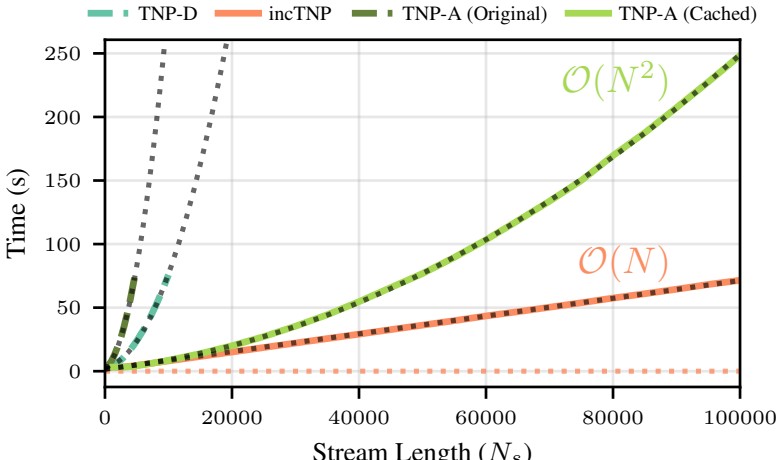

*Figure 14.* Cost of each AR mode deployment step as weather observations arrive sequentially on the HadISD task. At each step, models condition on all streamed data $N_s$ and perform AR prediction on a fixed target set of $N_t = 250$ using $S = 50$ sample unrolls. The dotted orange line represents the cost of updating `incTNP`'s context representation. Both the original unoptimised TNP-A model from Nguyen & Grover (2022), `TNP-A (Original)`, and our optimised version with KV caching implemented, `TNP-A (Cached)`, are compared. We only run `TNP-A (Original)` over 2,000 context points and AR `TNP-D` over 5,000 context points due to their expensive scaling, but fit quadratic functions to them (with dotted black lines) to show their expected scaling. We also fit cost curves to `TNP-A (Original)` and `incTNP` with dotted black lines to highlight their respective quadratic and linear per-step AR scaling. At $N_s = 100,000$, `TNP-A (Cached)` is $3.5\times$ slower than `incTNP` despite having approximately half as many parameters.

Our findings are echoed by Hassan et al. (2025), who show that `TNP-A`'s design is ill-suited for efficient autoregressive deployment. They show that the use of two target sets ($\{x_n^t, 0\}_{n=1}^{N_t}$ and $\{x_n^t, y_n^t\}_{n=1}^{N_t}$) imposes a significant computational burden during AR decoding, particularly when the target set is large. Additionally, they show that `TNP-A`'s bespoke masking strategy prevents the use of highly optimised kernels such as FlashAttention (Dao, 2024) during training, resulting in a prohibitively expensive training cost. Thus, whilst `TNP-A` achieves strong AR predictive performance (as shown in Table 2(b) and Table 3(b)), it is computationally demanding—particularly during training. By contrast, `incTNP-Seq` utilises a standard causal structure that is compatible with high-performance attention kernels and maintains a linear-time context update per step, making it a more practical candidate for the large-scale streaming tasks explored in this work.

## E.3. Implicit Bayesianness

We now provide additional details about our empirical investigation into the implicit Bayesianness of the models' implied prediction rules under our specific streaming evaluation protocol, as well as further results. These additional results further support our claim that `TNP-D` and `incTNP-Seq` exhibit similar degrees of consistency in their model updates.

### E.3.1. METRIC IMPLEMENTATION DETAILS

We detail the empirical quantity computed in our implicit Bayesianness experiments. Consider an evaluation task with a fixed context set $\mathcal{D}^{\text{fixed}}$ and $N$ target observations $\mathcal{D}^t = \{(\mathbf{x}_i^t, \mathbf{y}_i^t)\}_{i=1}^N$. For a given permutation $\pi \in \Pi_N$, the metric evaluates the model using a teacher-forced autoregressive factorisation. The context available prior to the $k$-th prediction is

$$H_{k,\pi}^{\text{TF}} = \left(\mathcal{D}^{\text{fixed}}, (\mathbf{x}_{\pi(1)}^t, \mathbf{y}_{\pi(1)}^t), \dots, (\mathbf{x}_{\pi(k-1)}^t, \mathbf{y}_{\pi(k-1)}^t)\right).$$

This induces the ordering-specific predictive density

$$p_{\theta,\pi}^{\text{TF}}(\tilde{\mathbf{y}}_{1:N}) = \prod_{k=1}^N p_\theta\left(\tilde{\mathbf{y}}_{\pi(k)} \mid \mathbf{x}_{\pi(k)}^t, H_{k,\pi}^{\text{TF}}\right).$$

Monte Carlo samples $\tilde{\mathbf{y}}_{\pi(k)}$ drawn from the model are used to evaluate the log-density of the current predictive factor, but are not added to the context. Subsequent predictions condition on the corresponding observed target values in $H_{k,\pi}^{\text{TF}}$.

The exchangeabilified predictive distribution is approximated by a finite mixture over $G$ random orderings $\{\rho_1, \dots, \rho_G\}$:

$$\hat{p}_{\theta,G}^{\text{TF}}(\tilde{\mathbf{y}}_{1:N}) = \frac{1}{G} \sum_{g=1}^G p_{\theta,\rho_g}^{\text{TF}}(\tilde{\mathbf{y}}_{1:N}).$$

In the implementation, each mixture component is represented by the predictive means and standard deviations obtained along its teacher-forced stream. These component parameters are then mapped back to the canonical target order before evaluating the mixture density.

The KL gap is estimated by Monte Carlo averaging over $J$ evaluation orderings $\pi_j$ and $S_{\text{MC}}$ samples:

$$\hat{g}_{\text{TF}} = \frac{1}{J S_{\text{MC}}} \sum_{j=1}^J \sum_{s=1}^{S_{\text{MC}}} \left[\log p_{\theta,\pi_j}^{\text{TF}}(\tilde{\mathbf{y}}^{(s,j)}) - \log \hat{p}_{\theta,G}^{\text{TF}}(\tilde{\mathbf{y}}^{(s,j)})\right], \qquad \tilde{\mathbf{y}}^{(s,j)} \sim p_{\theta,\pi_j}^{\text{TF}}.$$

The final metric is obtained by averaging $\hat{g}_{\text{TF}}$ across evaluation tasks.

### E.3.2. 1D GP REGRESSION

In the example provided in Section 5.2, we condition on 20 initial fixed context points, and predict over a set of 40 targets. The number of initial context points is chosen to ensure stable pre-training of `WISKI` so that it indeed represents a fair baseline. The factorisation we are using is:

$$p_{\text{NP}}(\mathbf{y}_{1:40}^t | \mathbf{x}_{1:40}^t, \mathcal{D}^{\text{fixed}}) = \Pi_{i=1}^{40} p_{\text{NP}}(\mathbf{y}_i^t | \mathbf{y}_{1:i-1}^t, \mathbf{x}_{1:i}^t, \mathcal{D}^{\text{fixed}}).$$

To compute the metric, we evaluate over 2000 such datasets. We use 256 different permutations to compute the permutation-invariant prediction map $\hat{p}_{\text{NP}}$. To compute the expectation of the KL gap, we average over 128 different permutations.

Finally, we use 32 Monte Carlo samples to approximate the KL divergence between $p_{\mathrm{NP}}$ (Gaussian) and $\hat{p}_{\mathrm{NP}}$ (mixture of Gaussians). Because `WISKI` is more expensive to run than the TNP-based variants, we compute the metric on 128 datasets, using 64 permutations for constructing $\hat{p}_{\mathrm{NP}}$, and 32 permutations for computing the expectation of the KL gap. Finally, we use 20 Monte Carlo samples for computing the KL divergence.

To verify the robustness of our findings regarding implicit Bayesianness, we replicate the analysis in a sparser initial context setting: conditioning on an initial fixed context of 3 points and evaluating over 100 target points. We omit `WISKI` from this comparison, as the method proved numerically unstable given such a small fixed context size. The results, presented in Figure 15, corroborate our earlier conclusions: `incTNP-Seq` exhibits a KL gap only marginally higher than the baseline `TNP-D`, while achieving superior NLL. Similarly, although `incTNP` shows a slightly larger KL gap and lower predictive performance, it remains highly comparable to the non-incremental baseline.

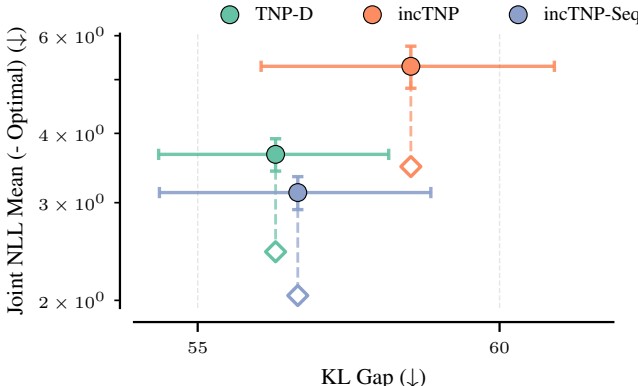

*Figure 15.* Performance (Joint NLL gap relative to optimal) versus implicit Bayesianness (KL Gap). Similarly to Figure 3, `incTNP-Seq` achieves a KL gap similar to the non-causal `TNP-D` and a better performance. In contrast, `incTNP`'s performance is poorer, and the KL gap is slightly larger, but within errors. Diamond markers (◇) denote the permutation-invariant versions of the models.

## E.4. Streaming Setting

### E.4.1. STREAMING TABULAR DATA

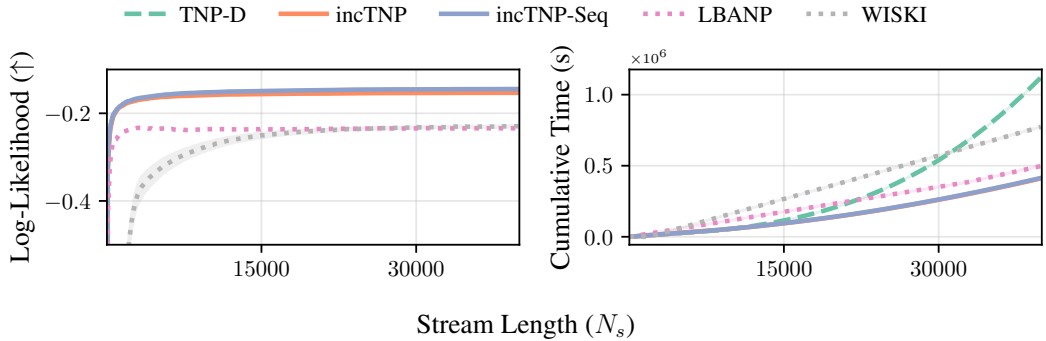

*Figure 16.* Test log-likelihood (↑) on the Tabular (Synthetic) dataset including baselines (`LBANP` and `WISKI`). (Left) The TNP variants show significantly better performance than the baselines. (Right) Although the TNP-based models show better performance, the TNP-D is more costly than the baselines at large streaming lengths. In contrast, the `incTNP` variants have both the best performance and lowest computational cost.

In the main part of the paper, we only compare TNP family members since the goal of the investigation is to expose any trade-offs introduced by our causal masking mechanism. However, a natural question to ask is how well our models perform in comparison to other baselines that practitioners might consider. We include `WISKI` as a baseline since it is a well-established stochastic process model suitable for streaming-data settings, and also the `LBANP` to represent the state-of-the-art of sub-quadratic and general-purpose TNP-based models. These additional results are shown in Figures 16 and 17 for synthetic and real-world scenarios, respectively. Although we also included a `CNP` as a baseline, its performance

was so poor that it would not even show up on the same scale in our figures.

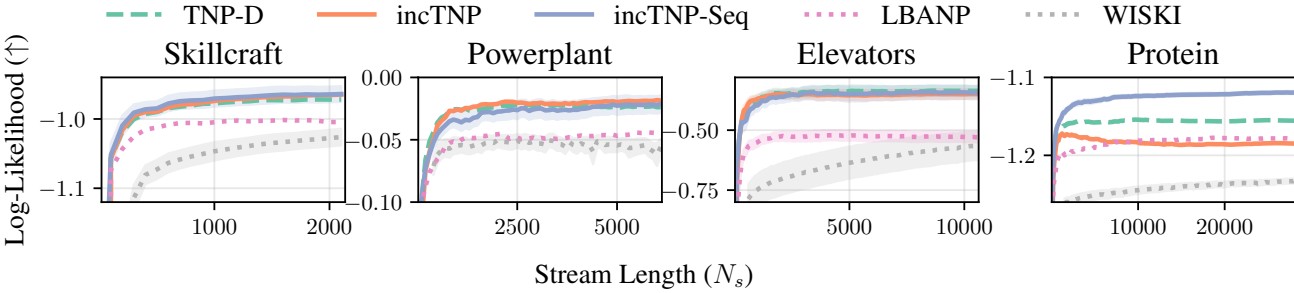

*Figure 17.* Test log-likelihood (↑) on Tabular (real-world datasets). The TNP-based variants show better performance on the real-world datasets too.

In both synthetic and real-world environments, the three TNP-type models considered in the main portion of the paper significantly outperform the extra baselines `LBANP` and `WISKI`. So, not only are the three TNP variants simply more performant in the straightforward synthetic setting (Figure 16), but there are also more efficient at overcoming the difficult sim-to-real transfer than the additional baselines are (Figure 17). Also note that, for all baselines, the predictive performance improves as the number of context points is increased, as expected.

### E.4.2. STREAMING TEMPERATURE AR PREDICTION

We evaluate the streamed performance of AR predictions for both spatiotemporal interpolation and forecasting on the temperature prediction task in Figure 18, including `TNP-A` as an additional baseline. In the interpolation setting (left), all TNP and incremental TNP variants improve their predictive performance as context arrives, consistently outperforming the `CNP` and `LBANP` baselines. Notably, `incTNP` closely matches `TNP-D`'s predictive capabilities across all context sizes, supporting the notion that causal masking of context points incurs minimal detriment to model performance. `incTNP-Seq` provides the strongest performance throughout the majority of the stream, ultimately converging to a similar log-likelihood as `TNP-A` at $N_s = 2000$. While `TNP-A` performs poorly when context is limited, it adapts effectively as the context set grows.

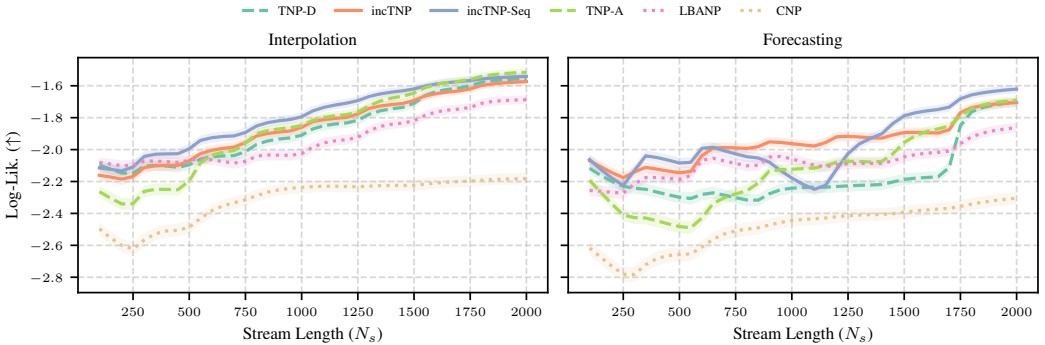

*Figure 18.* Test log-likelihood (↑) on the temperature prediction task using AR predictions, reporting streamed performance for both spatiotemporal interpolation (left) and forecasting (right). Results are averaged over 2,000 samples, with the shaded areas representing the standard error of the mean (SEM).

Figure 18 (right) showcases streamed AR performance for temperature forecasting. In this setup, observations are streamed from $H = 7$ time windows (spaced in steps of $\delta = 6$ hours), and performance is evaluated on the final window (48 hours from the first window). Consequently, the model must initially forecast 48 hours into the future, a task that becomes a 6-hour forecast by the end of the stream. Given the difficulty of this evaluation, performance improves more gradually, with significant boosts occurring as observations from the final time window arrive. Both `incTNP` and `incTNP-Seq` outperform `TNP-D` throughout the stream, with `incTNP-Seq` achieving superior final performance to all baselines. This highlights the effectiveness of the dense training strategy employed by `incTNP-Seq`, and the strong generalisation capabailites of incremental TNPs more broadly. Whilst `incTNP-Seq` experiences a transient performance drop for $N_s \in [600, 1100]$,

we attribute this to the inherent difficulty of forecasting at target locations that are multiple time windows into the future; similar drops are observed for `TNP-D` and `TNP-A`. Crucially, all models recover as they observe the final context window, with `incTNP-Seq` exhibiting the strongest final generalisation.

### E.5. AR Target Ordering Sensitivity

AR prediction induces a sensitivity to target ordering, and we impose a random target ordering across AR experiments unless otherwise stated following Bruinsma et al. (2023). We measure the standard deviation in log-likelihood per datapoint across random target orderings in Figure 19, following Figure 9 in Bruinsma et al. (2023). Our results show that `incTNP-Seq` is about as sensitive to target ordering as `TNP-D`, whilst `incTNP` is slightly more sensitive. For all models, a larger initial context set and larger target sets reduce the standard deviation in log-likelihood per datapoint, aligning with the findings of Bruinsma et al. (2023). Future work may wish to examine AR target ordering sensitivity further.

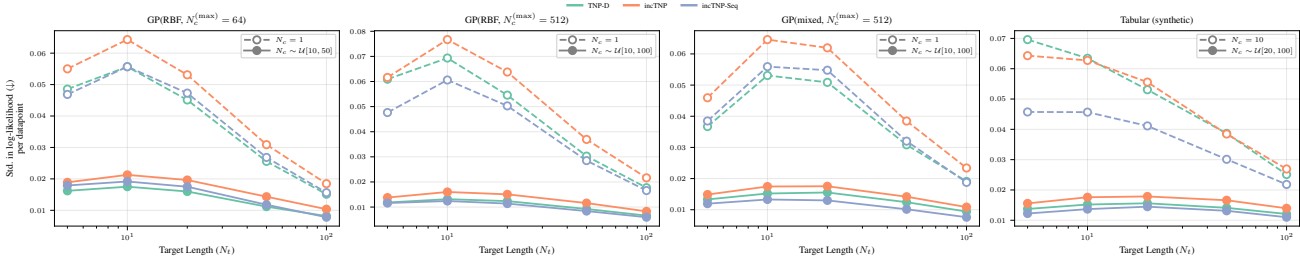

*Figure 19.* Standard deviation of the log-likelihood over different target orderings (following Figure 9 in Bruinsma et al. (2023)). Empty markers indicate small, fixed-size contexts (1 or 10 tabular datapoints), while filled markers represent larger, variable contexts sampled uniformly per the legend. We observe similar behaviour between `TNP-D` and `incTNP-Seq`, while `incTNP` tends to show slightly higher sensitivity to target ordering.

## F. Computational Complexity Analysis

In this section, we discuss the computational cost of `incTNP`, `incTNP-Seq` and `TNP-D` in the streaming setting. Within this environment, we evaluate on $N_t$ target points and a streamed context set $\mathcal{D}^s$ of size $N_s$. For AR mode predictions, we aggregate over $S$ separate sample unrolls. For simplicity—and to align with typical streaming setups—, we consider the case of $\mathcal{O}(1)$ new contextual observations for each streamed update step. Our analysis can easily be extended to deal with alternative context update arrangements.

**MHSA and MHCA cost**  Our cost calculations derive from the fact that self-attention on a sequence of length $A$ costs $\mathcal{O}(A^2)$ and that cross-attention between two sequences of lengths $A$ and $B$ costs $\mathcal{O}(AB)$. If causally masked self-attention has already been performed on a sequence of length $A$ and an additional point arrives at the end of that sequence, the cost of self-attention for that point falls to $\mathcal{O}(A)$ through KV caching. However, if self-attention is bidirectional, then the cost of self-attention remains $\mathcal{O}(A^2)$. We apply these facts to our NP models.

**Conditioning cost**  Early-stage NP variants such as the `CNP` support incrementally conditioning on contextual information as it arrives in $\mathcal{O}(N_s)$, allowing for a separation between conditional updating and target querying. As `TNP-D` models use bidirectional self-attention to encode the context set, they fail to support efficient incremental updating; each new context point requires the entire context representation to be recomputed in $\mathcal{O}(N_s^2)$. However, `incTNP` models use a causal masking within the context set. This avoids the need to recompute the contextual representations of previous context points, supporting efficient conditioning in $\mathcal{O}(N_s)$ through KV caching. This speedup ultimately makes `incTNP` models strong choices for streaming tasks and renders `TNP-D` prohibitively expensive.

**Querying cost (factorised)**  Having conditioned on the context set, both `TNP-D` and `incTNP` class models employ cross attention between embedded targets and context points, incurring an $\mathcal{O}(N_s N_t)$ cost.

**Querying cost (AR)**  Having conditioned on the context set, AR mode deployment involves iteratively predicting one target point at a time and concatenating a sample from that distribution to the context set:

$$p(\mathbf{Y}^t|\mathbf{X}^t;\mathcal{D}^s) = \prod_{n=1}^{N_t} p_\theta(\mathbf{y}_n^t|\mathbf{x}_n^t, \mathcal{D}^s \cup (\mathbf{x}_j^t, \mathbf{y}_j^t)_{j=1}^{n-1}). \tag{53}$$

This process is repeated over $S$ unrolls. At each AR update step, `TNP-D` has to recompute its entire contextual representation. Thus each step $i$ requires computing self-attention for a context set of size $N_s + i - 1$ (costing $\mathcal{O}((N_s + i - 1)^2)$), as well as computing cross attention between the single target point and this expanded context set (costing $\mathcal{O}(N_s + i - 1)$). Thus the total cost for AR mode prediction of `TNP-D` is:

$$S \sum_{i=1}^{N_t} \left[ \underbrace{\mathcal{O}((N_s + i - 1)^2)}_{\text{MHSA}} + \underbrace{\mathcal{O}(N_s + i - 1)}_{\text{MHCA}} \right]$$
$$= \mathcal{O}(SN_t(N_s + N_t)^2).$$

Both `incTNP` and `incTNP-Seq` dramatically speed up AR mode deployment as they allow for efficient incremental conditioning. Thus, the cost of updating the contextual representation at each step $i$ using the causally masked self-attention layer drops from a quadratic cost to a linear one: $\mathcal{O}((N_s + i - 1))$. Thus the total AR cost reduces to:

$$S \sum_{i=1}^{N_t} \left[ \underbrace{\mathcal{O}((N_s + i - 1))}_{\text{M-MHSA}} + \underbrace{\mathcal{O}(N_s + i - 1)}_{\text{MHCA}} \right]$$
$$= \mathcal{O}(SN_t(N_s + N_t)).$$

### F.1. Per-Update Cost

Table 4 summarises the per step cost of incremental TNP models relative to TNP. Within the streaming setting, causal masking enables computationally viable incremental context updates, enabling models to handle long streamed sequences.

*Table 4.* Summary of per-step streamed conditioning and querying cost for NP models, using both factorised and AR predictions. The training step cost for $N_c$ context points and $N_t$ targets is also included.

| Model | Update Cost (Step) | Querying Cost (Step) | | Training Cost |
| --- | --- | --- | --- | --- |
| | | Factorised | Autoregressive | |
| `incTNP` | $\mathcal{O}(N_s)$ | $\mathcal{O}(N_s N_t)$ | $\mathcal{O}(SN_t(N_s + N_t))$ | $\mathcal{O}(N_c^2 + N_c N_t)$ |
| `incTNP-Seq` | $\mathcal{O}(N_s)$ | $\mathcal{O}(N_s N_t)$ | $\mathcal{O}(SN_t(N_s + N_t))$ | $\mathcal{O}((N_c + N_t)^2)$ |
| `TNP-D` | $\mathcal{O}(N_s^2)$ | $\mathcal{O}(N_s N_t)$ | $\mathcal{O}(SN_t(N_s + N_t)^2)$ | $\mathcal{O}(N_c^2 + N_c N_t)$ |

Consider a streaming update step requiring NP models to have conditioned on all $N_s$ streamed points and compute the target distribution for $N_t$ target points. Predicting a factorised distribution with `TNP-D` costs $\mathcal{O}(N_s^2 + N_s N_t)$ per streaming step, increasing to $\mathcal{O}(SN_t(N_s + N_t)^2)$ for AR predictions. Our incremental TNP models scale much more attractively, performing in $\mathcal{O}(N_s N_t)$ and $\mathcal{O}(SN_t(N_s + N_t))$ for factorised and AR mode respectively.

### F.2. Cumulative Cost

Next, we explore the *cumulative* cost across $N_s$ streaming steps. `TNP-D` suffers a quadratic conditioning cost at each step resulting in a cubic cumulative complexity:

$$\sum_{i=1}^{N_s} \mathcal{O}(i^2) = \mathcal{O}(N_s^3).$$

`incTNP` and `incTNP-Seq` only suffer a linear conditioning cost per step, leading to an overall quadratic complexity:

$$\sum_{i=1}^{N_s} \mathcal{O}(i) = \mathcal{O}(N_s^2).$$

For our HadISD weather forecasting task, the streaming setup is such that the target set is fixed, but the predictions are updated as new context observations arrive in sequence. In particular, the observations arrive in a temporal ordering, allowing us to update our temperature predictions for a fixed set of $N_t = 250$ target stations. In such cases, the cumulative querying cost of factorised predictions for all models becomes:

$$\sum_{i=1}^{N_s} \mathcal{O}(iN_t) = \mathcal{O}(N_s^2 N_t).$$

In the case of AR mode deployment, `TNP-D` suffers from a cumulative querying cost of:

$$\sum_{i=1}^{N_s} \mathcal{O}(SN_t(i + N_t)^2) = \mathcal{O}(SN_t N_s(N_s + N_t)^2).$$

`incTNP` and `incTNP-Seq` offer a much more scalable cumulative querying cost of:

$$\sum_{i=1}^{N_s} \mathcal{O}(SN_t(i + N_t)) = \mathcal{O}(SN_t N_s(N_s + N_t)).$$

Thus for cases where $N_t << N_s$ and $N_t \in \mathcal{O}(1)$, the cumulative cost for `incTNP` models is quadratic, whilst it is cubic for `TNP-D`. An example of this cumulative scaling on the HadISD task is shown in Figure 20, where `TNP-D` is 3.5 times more computationally expensive after 2000 context points, with the gap ever expanding. For AR streaming tasks where the number of target points is large, this gap becomes significantly larger still. In short, imposing causal masking across the context set allows for transformer neural processes to be used in streaming tasks for both factorised and AR predictions, something which is simply not computationally viable for `TNP-D`.

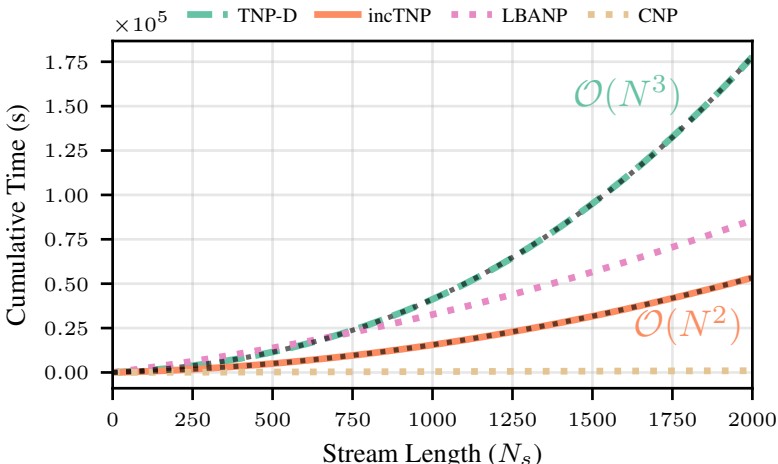

*Figure 20.* Cumulative runtime cost (measured in seconds) on the HadISD dataset of deploying in AR mode as the context set grows ($N_s$). Runtime measured as station observations arrive in a streamed fashion, resulting in updated predictions at $N_t = 250$ stations using $S = 50$ sample unrolls across 32 batches. We fit quadratic and cubic cumulative cost functions to `incTNP` and `TNP-D` (plotted with dotted black lines), to highlight the cumulative AR scaling cost of both approaches.

Cumulative runtime performance is summarised in Table 5.

*Table 5.* Summary of cumulative streamed conditioning and querying cost for NP models, using both factorised and AR predictions. Both `incTNP` and `incTNP-Seq` share the same streamed cumulative scaling and are presented together.

| Model | Update Cost (Cumulative) | Querying Cost (Cumulative) | |
|---|---|---|---|
| | | Factorised | Autoregressive |
| `incTNP(-Seq)` | $\mathcal{O}(N_s^2)$ | $\mathcal{O}(N_s^2 N_t)$ | $\mathcal{O}(SN_t N_s(N_s + N_t))$ |
| `TNP-D` | $\mathcal{O}(N_s^3)$ | $\mathcal{O}(N_s^2 N_t)$ | $\mathcal{O}(SN_t N_s(N_s + N_t)^2)$ |

## G. Memory Complexity Analysis

`incTNP` and `incTNP-Seq` achieve a substantial runtime speed-up over `TNP-D` at inference time through the use of KV caching. This section considers the memory complexity of both incremental approaches (which share an identical memory footprint at inference time). We assess the total memory cost, which can be decomposed into persistent and transient cost, for $N_t$ target points on a streamed context set of size $N_s$. We aggregate over $S$ sample unrolls for AR mode predictions, which we generalise to a batch size $B$ ($B = 1$ for factorised streaming prediction and $B = S$ for AR prediction). As discussed in Appendix B, all transformer-based NPs use $L$ layers, with $H$ heads and an embedding dimension $D_z$. As in Appendix F, we consider the case of $N_s' \in \mathcal{O}(1)$ new contextual observations per streaming update, though our analysis can be easily extended to consider alternative context update patterns.

### G.1. Transient Memory Cost

Within this subsection, we focus exclusively on the transient (i.e. non-persistent) memory cost of `incTNP` and `TNP-D`.

**Embedder and Decoder**   We make use of an embedder and decoder MLP for `incTNP` and `TNP-D`. Both models pay an $\mathcal{O}(BN_t D_z)$ cost to decode target predictions. Embedding cost scales linearly with the total number of new tokens $N_{\text{new}} = N_s' + N_t$ to be embedded, $\mathcal{O}(BN_{\text{new}}D_z)$. Thus the transient cost of periphery NP components is $M_{\text{peri}} \in \mathcal{O}(BN_{\text{new}}D_z)$. In the case of `TNP-D`, $N_s' = N_s$ which results in a cost of $M_{\text{peri}}^{\text{TNP-D}} \in \mathcal{O}(B(N_s + N_t)D_z)$. We focus on the case when `incTNP` processes $N_s' \in \mathcal{O}(1)$ new tokens per step, costing $M_{\text{peri}}^{\text{incTNP}} \in \mathcal{O}(BN_t D_z)$. In both cases, these terms are of the same order as, or dominated by, the transient cost of the transformer component.

**Transformer**   At inference time, computing multi-head attention on $N_q$ query tokens and $N_{kv}$ key/value tokens induces a transient memory cost $M_{\text{transformer}}(N_q, N_{kv})$. This cost arises from the dynamic allocation of intermediate tensors across three distinct stages of the transformer forward pass:

1. **Pre-Norm and Projections:** The inputs must be normalised and linearly projected into query, key, and value representations. Allocating these intermediate tensors scales linearly with the number of tokens being projected, costing $\mathcal{O}(B(N_q + N_{kv})D_z)$.

2. **Attention Kernel:** The memory required to compute attention weights for scaled-dot product attention (SDPA) $M_{\text{attn}}(N_q, N_{kv})$ varies based on the attention kernel used. Naive SDPA attention kernels materialise the full $N_q \times N_{kv}$ attention matrix across $H$ heads, resulting in a quadratic transient spike $M_{\text{attn}}^{\text{naive}}(N_q, N_{kv}) \in \mathcal{O}(BHN_q N_{kv})$. Conversely, memory-efficient attention kernels, such as FlashAttention (Dao, 2024), do not materialise the full similarity matrix, reducing the kernel's transient footprint to a linear one: $M_{\text{attn}}^{\text{flash}} \in \mathcal{O}(B(N_q + N_{kv})D_z)$. The availability of memory-efficient kernels tends to depend on the hardware used, with FlashAttention supported on high-end GPUs.

3. **MLP:** The output of the attention mechanism is passed through the transformer's shallow MLP, costing $\mathcal{O}(BN_q D_z)$.

At inference, we analyse peak transient memory during a forward pass, not cumulative memory allocated across all layers. Since transformer layers are executed sequentially and activations are not retained for backpropagation, this peak transient cost does not scale with the number of layers $L$. Thus, the transient memory cost with naive attention is

$$M_{\text{transformer}}^{\text{naive}}(N_q, N_{kv}) \in \mathcal{O}\left(B\left(D_z\left(N_q + N_{kv}\right) + HN_q N_{kv}\right)\right).$$

Memory-efficient attention comes at a reduced transient cost of

$$M_{\text{transformer}}^{\text{flash}}\left(N_q, N_{kv}\right) \in \mathcal{O}\left(BD_z\left(N_q + N_{kv}\right)\right).$$

We use $M_{\text{transformer}}(N_q, N_{kv}) = M_{\text{transformer}}^{\text{flash}}(N_q, N_{kv})$ unless otherwise specified within our cost-analysis.

Both `TNP-D` and `incTNP` make a MHCA call during querying, using $N_q = N_t$ queries and $N_{kv} = N_s$ key/value tokens at a cost of $M_{\text{transformer}}(N_t, N_s)$. Both models also make a MHSA call using $N_{kv} = N_s$ key/value tokens. Whilst `TNP-D` uses $N_q = N_s$ queries for each MHSA call, KV caching allows `incTNP` to only query new contextual observations $N_s' \in \mathcal{O}(1)$ such that $N_q = N_s'$. This results in a MHSA cost of $M_{\text{transformer}}(N_s, N_s)$ for `TNP-D` and a cost of $M_{\text{transformer}}(N_s', N_s)$ for `incTNP`.

**Factorised Predictions**   When making factorised predictions within the streaming setting, we set the batch size $B = 1$ and consider the case $N_s' \in \mathcal{O}(1)$.

For `incTNP`, updating the contextual representation has a transient memory footprint of:

$$\mathcal{O}(M_{\text{transformer}}(N_s', N_s)) = \mathcal{O}(D_z N_s).$$

`incTNP` performs cross-attention between the targets and context points at query time at a cost of:

$$\mathcal{O}(M_{\text{transformer}}(N_t, N_s)) = \mathcal{O}(D_z(N_s + N_t)).$$

Our `TNP-D` implementation does not support conditioning and thus performs self attention and cross attention sequentially when predicting at targets:

$$\mathcal{O}(M_{\text{transformer}}(N_s, N_s) + M_{\text{transformer}}(N_t, N_s)) = \mathcal{O}(D_z(N_s + N_t)).$$

**Autoregressive Predictions**   When making autoregressive predictions, transient memory usage peaks when appending the very last context point to the initial context set. For AR mode, we set $B = S$.

`TNP-D` recomputes self-attention over all previous points whilst also performing cross-attention to obtain predictions over the last target:

$$\mathcal{O}\left(\underbrace{M_{\text{transformer}}\left(N_t + N_s - 1, N_t + N_s - 1\right)}_{\text{MHSA}} + \underbrace{M_{\text{transformer}}\left(1, N_t + N_s - 1\right)}_{\text{MHCA}}\right)$$
$$= \mathcal{O}(SD_z(N_t + N_s)).$$

Whilst `incTNP` typically has a lower transient AR memory cost, it shares the same asymptotic behaviour as `TNP-D`:

$$\mathcal{O}\left(\underbrace{M_{\text{transformer}}\left(1, N_t + N_s - 1\right)}_{\text{M-MHSA}} + \underbrace{M_{\text{transformer}}\left(1, N_t + N_s - 1\right)}_{\text{MHCA}}\right)$$
$$= \mathcal{O}(SD_z(N_t + N_s)).$$

AR querying cost dominates the contextual updating costs, resulting in a total AR cost of $\mathcal{O}(SD_z(N_t + N_s))$.

*Table 6.* Summary of per-step streamed conditioning and querying transient memory costs for NP models, using both factorised and AR predictions. Cost is shown for the memory-efficient kernel $M_{\text{transformer}}^{\text{flash}}$(*Mem-Eff*) and standard attention kernel $M_{\text{transformer}}^{\text{naive}}$(*Standard*).

| Model | Update Cost | Querying Cost | | Attention |
|---|---|---|---|---|
| | | **Factorised** | **Autoregressive** | |
| incTNP | $\mathcal{O}(D_z N_s)$ | $\mathcal{O}(D_z(N_t + N_s))$ | $\mathcal{O}(SD_z(N_t + N_s))$ | Mem-Eff |
| TNP-D | — | $\mathcal{O}(D_z(N_t + N_s))$ | $\mathcal{O}(SD_z(N_t + N_s))$ | Mem-Eff |
| incTNP | $\mathcal{O}(N_s(D_z + H))$ | $\mathcal{O}(D_z(N_t + N_s) + HN_sN_t)$ | $\mathcal{O}(S(N_t + N_s)(D_z + H))$ | Standard |
| TNP-D | — | $\mathcal{O}(D_z(N_t + N_s) + H(N_s^2 + N_sN_t))$ | $\mathcal{O}(S(N_t + N_s)(D_z + H(N_t + N_s)))$ | Standard |

**Transient Cost Summary**   Table 6 summarises the transient memory costs of both incremental non-incremental transformer-based approaches.

### G.2. Persistent Memory Cost

**Weights**   Both incremental and standard TNPs share an identical parameter count, incurring a static memory footprint of $W \in \mathcal{O}(LD_z^2)$. Crucially, $W$ is independent of $N_s$ and $N_t$, acting simply as an $\mathcal{O}(1)$ constant.

**KV Cache**   `incTNP` supports incremental conditioning by storing all projected key and value representations from `incTNP`'s M-MHSA layers. This enables a substantial runtime speedup but comes at a persistent memory cost of $\mathcal{O}(BLD_zN_s)$. For factorised predictions this results in a footprint of $\mathcal{O}(LD_zN_s)$, whilst AR prediction has a peak cost of $\mathcal{O}(SLD_z(N_s + N_t))$.

### G.3. Total Cost

The peak live memory is given by the persistent memory plus the maximum transient memory incurred across the conditioning and querying stages. Table 7 outlines the overall memory scaling of our approach against `TNP-D` when using memory-efficient attention kernels and standard quadratic memory implementations. In the former case, both systems scale linearly with respect to the number of streamed context points $N_s$. `incTNP` also scales linearly with respect to the number of transformer layers $L$ as a result of its KV cache, which has limited impact on asymptotic scaling as we treat $L \in \mathcal{O}(1)$. When using quadratic memory attention kernels, `incTNP` still scales linearly with respect to the number of streamed context points $N_s$ whilst `TNP-D` scales quadratically. Consequently, `incTNP` scales as favourably or more favourably than `TNP-D` in terms of asymptotic memory cost within the streaming setting.

*Table 7.* Summary of per-step streamed maximum memory costs for NP models, using both factorised and AR predictions. Cost is shown for the memory-efficient kernels $M_{\text{transformer}}^{\text{flash}}$(*Mem-Eff*) and standard attention kernels $M_{\text{transformer}}^{\text{naive}}$(*Standard*).

| Model | Cost | | Attention |
|---|---|---|---|
| | **Factorised** | **Autoregressive** | |
| incTNP | $\mathcal{O}(D_z(N_t + LN_s) + W)$ | $\mathcal{O}(SD_zL(N_t + N_s) + W)$ | Mem-Eff |
| TNP-D | $\mathcal{O}(D_z(N_t + N_s) + W)$ | $\mathcal{O}(SD_z(N_t + N_s) + W)$ | Mem-Eff |
| incTNP | $\mathcal{O}(D_z(N_t + LN_s) + H(N_sN_t) + W)$ | $\mathcal{O}(S(N_t + N_s)(LD_z + H) + W)$ | Standard |
| TNP-D | $\mathcal{O}(D_z(N_t + N_s) + H(N_s^2 + N_sN_t) + W)$ | $\mathcal{O}(S(N_t + N_s)(D_z + H(N_t + N_s)) + W)$ | Standard |

### G.4. Measured Memory Costs

Figure 21 plots the measured peak memory of `incTNP` and `TNP-D` for factorised predictions on the temperature prediction task. When using FlashAttention in Figure 21a, memory requirements scale linearly for both models. `incTNP`'s KV cache contributes most to peak memory use, whilst `TNP-D`'s memory cost stems almost entirely from transient allocation. Whilst both approaches scale linearly with respect to the number of streamed points $N_s$, `incTNP` consistently uses more memory. This may present an issue for large scale models, however for this work we encountered no constraints even when going to one million streamed points. Running `TNP-D` at this scale is prohibitively slow ($\sim 650\times$ slower than `incTNP`), and thus

the limited memory-runtime tradeoff within this scenario is fully justified. Future work may wish to consider cache eviction and summarisation strategies, as well as use of constant-memory attention blocks (Feng et al., 2024).

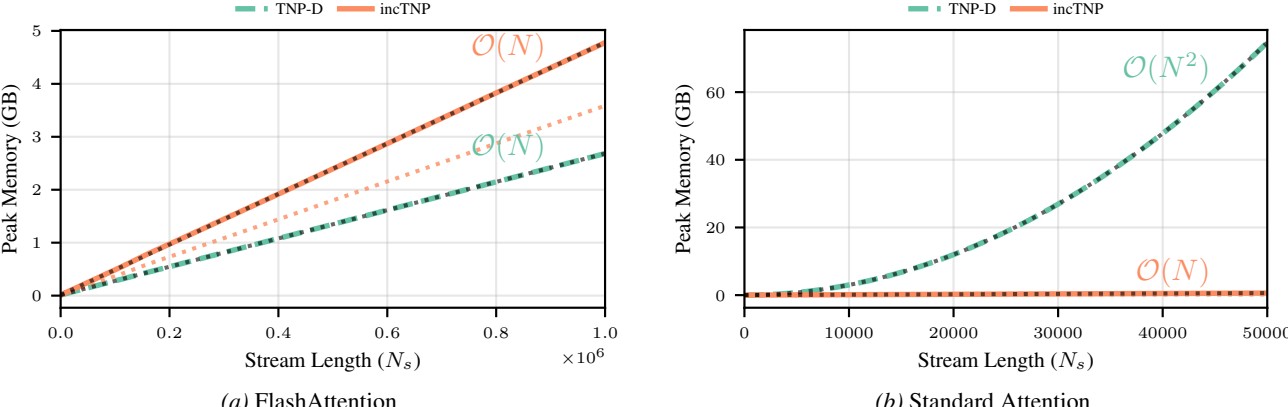

*(a)* FlashAttention                    *(b)* Standard Attention

*Figure 21.* Measured memory scaling of `TNP-D` and `incTNP` models on the HadISD temperature prediction task as the streamed context set $N_s$ grows for factorised predictions on $N_t = 250$ locations. Peak memory use is reported when using FlashAttention (left - $M_{\text{transformer}}^{\text{flash}}$) and when using standard quadratic memory attention (right - $M_{\text{transformer}}^{\text{naive}}$). A dotted orange line is used to indicate the persistent memory costs of `incTNP` (i.e. the memory cost of the KV cache and model weights), which contribute to the peak memory. Memory measured on a single NVIDIA A100 80GB GPU, using 16 bit floats and PyTorch's `torch.nn.attention.SDPBackend` FlashAttention and Math backends. Dotted black lines are fit to peak memory plots to highlight $\mathcal{O}(N)$ linear or $\mathcal{O}(N^2)$ quadratic memory scaling.

Using an attention kernel with a quadratic memory cost, as in Figure 21b, results in quadratic memory scaling for `TNP-D` as it recomputes the attention scores between all streamed tokens at each step. This quadratic scaling prevents `TNP-D` from streaming over $50,000$ tokens, whilst `incTNP` retains its linear complexity. For systems where efficient-memory attention kernels are not well supported, `incTNP` is clearly the superior choice in terms of memory and runtime cost. When linear-memory attention kernels are available, both approaches scale linearly with `incTNP`'s KV cache using more memory in practice.

### G.5. Additional Memory Considerations

**Raw Stream Storage**   Beyond model-state memory, practical streaming implementations may also need to retain raw inputs required for future predictions. In our implementation, `TNP-D` recomputes predictions from the full raw context set and therefore requires persistent storage of all previously observed context pairs, incurring an additional storage cost of $\mathcal{O}(N_s(D_x + D_y) + N_t D_x)$ at inference time, where $D_x$ is feature dimensionality and $D_y$ is output dimensionality. By contrast, `incTNP` encodes past context into a KV cache and therefore does not require the raw context stream to be retained, beyond the current streamed update of size $N_s' \in \mathcal{O}(1)$ and the target inputs used for querying. This results in a raw-data storage cost of $\mathcal{O}(N_t D_x + N_s'(D_x + D_y)) = \mathcal{O}(N_t D_x + D_y)$.

**Training Cost**   Our memory cost analysis primarily considers the footprint of `incTNP` at inference time, as a result of its use of KV caching and incremental updates. KV caching is not used during training, and thus `incTNP` and `TNP-D` share the same training memory profile of $\mathcal{O}(BLD_z(N_c + N_t) + W)$ with FlashAttention or $\mathcal{O}(BL(D_z(N_c + N_t) + H(N_c^2 + N_c N_t)) + W)$ with standard attention. Whilst `incTNP-Seq` behaves identically to `incTNP` at test time, it incurs a greater training cost of $\mathcal{O}(BL(D_z(N_c + N_t) + H(N_c + N_t)^2) + W)$ under standard attention. This asymptotic cost reduces back to the cost of `incTNP` training, $\mathcal{O}(BLD_z(N_c + N_t) + W)$, when using FlashAttention. Consequently, under memory-efficient attention, the training memory requirements of `incTNP`, `incTNP-Seq`, and `TNP-D` are of the same asymptotic order.

## H. Sensitivity to Context Ordering

In order to achieve linear-complexity streaming updates, incremental TNPs sacrifice a core property of standard Neural Processes: permutation invariance with respect to the context set. In the main text, we devise a metric of implicit Bayesianness to measure the consistency of the implied prediction rule. We find that `incTNP` is as consistent as `TNP-D` for streaming inference under this metric, despite lacking context permutation invariance.

Whilst consistency is the more relevant measure for *streaming* deployment, we analyse the degree of sensitivity to context permutation exhibited by `incTNP` and `incTNP-Seq` for *static* context sets within this section.

**Context permutation invariance** Formally, a model $p_\theta$ is context permutation invariant if and only if its predictive distribution is identical under all $\pi \in \Pi_{N_c}$ possible orderings of the context set $\mathcal{D}_\pi^c$:

$$\forall \pi \in \Pi_{N_c}, p_\theta(\mathbf{Y}^t|\mathbf{X}^t, \mathcal{D}^c) = p_\theta(\mathbf{Y}^t|\mathbf{X}^t, \mathcal{D}_\pi^c).$$

### H.1. Measure of Context Permutation Sensitivity

To measure the sensitivity of model prediction to permutation of the context set, we use an empirical diagnostic $g_{\text{ctx}}$, which we term the *context KL gap*.

For a specific target set and context ordering, we aim to measure the gap $g$ between the predictive distribution $q$ and the context permutation invariant construction $z$:

$$
\begin{aligned}
q &:= p_\theta(\mathbf{Y}^t|\mathbf{X}^t, \mathcal{D}^c), \\
z &:= \frac{1}{N_c!} \sum_{\pi \in \Pi_{N_c}} p_\theta(\mathbf{Y}^t|\mathbf{X}^t, \mathcal{D}_\pi^c), \\
g &= D_{\text{KL}}(q \parallel z).
\end{aligned}
$$

In practice, we approximate $z$ using $K < N_c!$ random context permutations and perform a Monte Carlo approximation using $S_{\text{MC}}$ samples to estimate the KL divergence between $q$ and $z$. We estimate the context KL gap $g_{\text{ctx}}$ by aggregating gaps across $R_{\text{draws}}$ random draws of context and target sets of fixed lengths $N_c$ and $N_t$.

By the properties of the KL divergence, we know that $g_{\text{ctx}} \geq 0$, and context permutation invariant models attain zero gap. Larger values indicate greater sensitivity to context ordering. Since context permutation invariance can be achieved trivially, we situate the context KL gap alongside predictive accuracy (log-likelihood); Mlodozeniec et al. (2024) opt for a similar approach in their analysis of implicit Bayesianness.

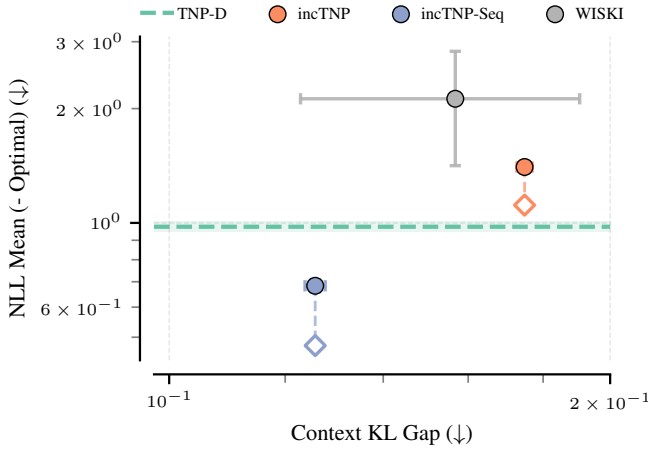

*Figure 22.* Performance (NLL gap relative to optimal for factorised predictions) versus context permutation sensitivity (context KL gap) for $N_c = 128$ context points and $N_t = 128$ target points. Diamond markers ($\diamond$) denote context permutation invariant model constructions. `TNP-D` is context permutation invariant (up to numerical noise) and thus is represented as a dotted line. We plot the standard error of the mean (SEM) of all models. `WISKI` is pretrained on the first 64 context points, and updated in a streaming manner for the remaining 64 context points. All NP models are trained with $N_c^{(\text{max})} = 512$.

We evaluate context sensitivity on 1D regression with the RBF kernel, considering both high and low context environments. We first consider context sensitivity for $N_c = 128$ context points, benchmarking our models against `TNP-D` and `WISKI`.

For all transformer-based NPs, we compute with $S_{\text{MC}} = 256$ Monte Carlo samples and $K = 256$ permutations per draw, averaging over $R_{\text{draws}} = 4096$ data draws. We use $S_{\text{MC}} = 20$, $K = 20$ and $R_{\text{draws}} = 512$ for `WISKI` due to its increased computational cost, pretraining on the first 64 context points and streaming the remaining 64 context observations. Figure 22 visualises our results, showcasing that incremental TNPs achieve comparable or favourable context sensitivity relative to `WISKI` whilst obtaining stronger modelling performance within this evaluation regime. `incTNP-Seq`'s dense autoregressive training strategy results in a lower sensitivity to context permutation than that of `incTNP` whilst also improving model performance within this test setting.

Figure 23 visualises context permutation sensitivity within a low context environment of $N_c = 8$ context points and $N_t = 10$ targets. We employ the same setup as the previous figure for measuring the context KL gap for transformer-based NPs, but use $S_{\text{MC}} = 128$, $K = 128$ and $R_{\text{draws}} = 1024$ for `TNP-A` due to its computational cost. Once again, `incTNP-Seq` exhibits a reduced sensitivity to context permutation compared to `incTNP` whilst offering superior modelling. Unsurprisingly, all factorised transformer-based NP models are outperformed by the autoregressive `TNP-A` models. Like `TNP-D`, `TNP-A` uses bidirectional attention to encode the context set and is thus context permutation invariant. However, during inference `TNP-A` samples autoregressively from target predictions to produce a Monte Carlo approximation of its true predictive distribution. Both incremental TNPs have a smaller measured $g_{\text{ctx}}$ than `TNP-A` with 50 sample unrolls, but exceed that of `TNP-A` with 100 samples unrolls. This helps to situate the measured sensitivity of our incremental TNPs relative to the baseline noise inherent to AR-style joint estimation with finite samples.

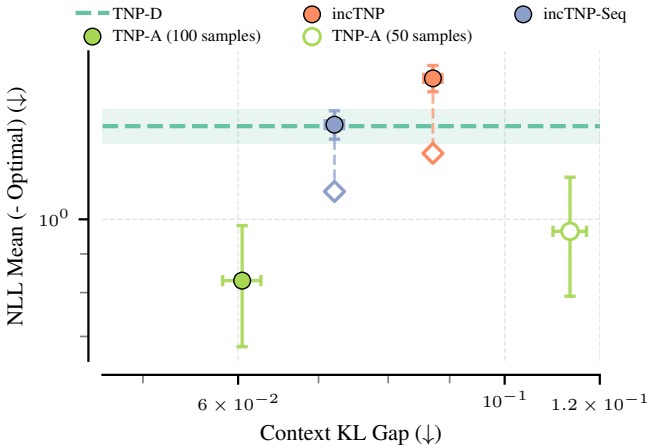

*Figure 23.* Performance (NLL gap relative to optimal for factorised predictions) versus context permutation sensitivity (context KL gap) for $N_c = 8$ context points and $N_t = 10$ target points. Diamond markers (◇) denote context permutation invariant model constructions. `TNP-D` is context permutation invariant (up to numerical noise) and thus is represented as a dotted line. We plot the standard error of the mean (SEM) of all models. We plot the test time characteristics of `TNP-A` when unrolling over $S = 50$ and $S = 100$ autoregressive sample unrolls. All NP models are trained with $N_c^{(\text{max})} = 64$.

## H.2. Test Set Performance with Context Permutation

Having quantified the sensitivity to context permutation of incremental TNP models, we now consider their implicit performance distribution over random contextual permutation. To do this we generate a test set consisting of 4096 task samples. We then randomly permute the ordering of context points for each sample, measuring the mean log-likelihood performance for each permuted test dataset to approximate the distribution of test set performance for each NP. This highlights both the mean performance and the expected variance across random context orderings.

We measure test set performance fluctuation across 1D regression (using the RBF kernel for $N_c^{(\text{max})} = 64$ and $N_c^{(\text{max})} = 512$) and synthetic tabular regression. We exclude HadISD temperature prediction from this analysis as models are trained with a temporal context ordering.

### H.2.1. 1D GP REGRESSION

Figure 24 showcases the performance distribution of incremental TNPs over random context permutations on 1D GP regression. Both `incTNP` and `incTNP-Seq` exhibit a relatively low average sensitivity to context ordering, as typified by

their large central density, allowing them to be reliably deployed on randomly ordered datasets.

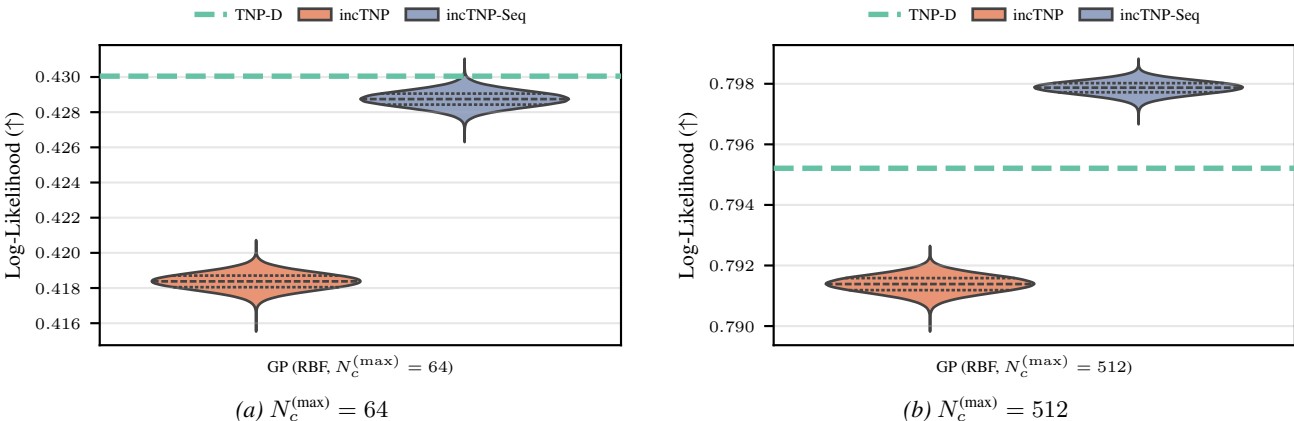

*(a)* $N_c^{(\text{max})} = 64$        *(b)* $N_c^{(\text{max})} = 512$

*Figure 24.* Distribution of test set log-likelihoods across random context permutations for `TNP-D`, `incTNP` and `incTNP-Seq`. We consider the distribution for both $N_c^{(\text{max})} = 64$ (left) and $N_c^{(\text{max})} = 512$ (right) RBF GP regression tasks. We approximate over $1,000,000$ test set permutations per model (processing $\sim 4$ billion samples per model) for both tasks. `TNP-D` is context permutation invariant and thus represented with a dotted line. For incremental NPs, we plot distribution density with a violin plot, using internal dashed lines to signify the distribution quartiles.

### H.2.2. SYNTHETIC TABULAR DATA

Figure 25 illustrates the distribution of average performance for synthetic tabular data. Across the tabular task, as well as both GP tasks (in Figure 24), `incTNP-Seq`'s performance is less sensitive to context ordering, whilst achieving significantly better modelling accuracy. This further underscores the strength of `incTNP-Seq`'s autoregressive training strategy, which consistently remains competitive with or outperforms `TNP-D` despite using a less expressive attention mechanism. This shows that whilst causal masking violates context permutation invariance, average variation is limited in practice and justified by the improved performance it affords.

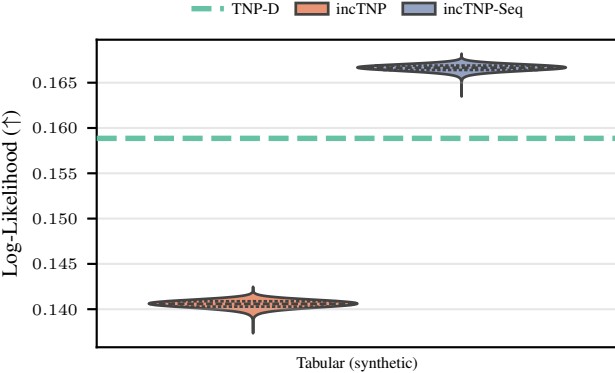

*Figure 25.* Distribution of test set log-likelihoods across random context permutations for `TNP-D`, `incTNP` and `incTNP-Seq` on synthetic tabular data. We approximate over $1,000,000$ test set permutations per model. `TNP-D` is context permutation invariant and thus represented with a dotted line. For incremental NPs, we plot distribution density with a violin plot, using internal dashed lines to signify the distribution quartiles.

### H.3. Context Ordering Algorithm

With incremental TNPs displaying limited average performance variation for random contextual shuffling, we now consider how to best order $N_c^{(\text{new})}$ new context points $\mathcal{D}_c^{(\text{new})}$ given an incremental model $\text{NP}_\theta$ already conditioned on $N_c^{(\text{init})}$ initial context points $\mathcal{D}_c^{(\text{init})}$. This is a particularly relevant consideration for batched streaming updates and active learning scenarios.

Inspired by the widely used *greedy variance* criterion for selecting inducing points of sparse Gaussian Processes, such as

in Burt et al. (2020), variance-based context ordering is explored. Let $\mathcal{D}_c$ denote the currently accumulated context set, initialized as $\mathcal{D}_c^{(\text{init})}$. For each new context point $d_i = (\mathbf{x}_i, \mathbf{y}_i) \in \mathcal{D}_c^{(\text{new})}$, we evaluate the incremental model at the input location $\mathbf{x}_i$ to obtain the predictive moments $\mu_i, \sigma_i^2 = \text{NP}_\theta(\mathbf{x}_i, \mathcal{D}_c)$. The variance $\sigma^2$ is used to select which point should be incorporated into the context set at that iteration. Maximum, median and minimum variance selection are all considered. We formalise our approach in Algorithm 2, incurring a runtime complexity of $\mathcal{O}((N_c^{(\text{new})})^2(N_c^{(\text{init})} + N_c^{(\text{new})}))$. This is well-suited towards batched streaming where $N_c^{(\text{new})}$ is often much smaller than $N_c^{(\text{init})}$.

---

**Algorithm 2** Greedy Context Data Ordering

---

1: **Input:** Initial context sequence $\mathcal{D}_c^{(\text{init})}$, new context points $\mathcal{D}_c^{(\text{new})}$, incremental TNP model $\text{NP}_\theta$, selection strategy $s \in \{\max, \text{median}, \min\}$.
2: $\mathcal{D}_c \leftarrow \mathcal{D}_c^{(\text{init})}$ {Initialize ordered context sequence}
3: **while** $\mathcal{D}_c^{(\text{new})} \neq \emptyset$ **do**
4:     **for each** point $d_i = (\mathbf{x}_i, \mathbf{y}_i) \in \mathcal{D}_c^{(\text{new})}$ **do**
5:         $\mu_i, \sigma_i^2 \leftarrow \text{NP}_\theta(\mathbf{x}_i, \mathcal{D}_c)$
6:     **end for**
7:     **if** $s = \max$ **then**
8:         $d^* \leftarrow \arg\max_{d_i \in \mathcal{D}_c^{(\text{new})}} \sigma_i^2$
9:     **else if** $s = \text{median}$ **then**
10:         $d^* \leftarrow \arg\text{median}_{d_i \in \mathcal{D}_c^{(\text{new})}} \sigma_i^2$
11:     **else**
12:         $d^* \leftarrow \arg\min_{d_i \in \mathcal{D}_c^{(\text{new})}} \sigma_i^2$
13:     **end if**
14:     $\mathcal{D}_c \leftarrow \mathcal{D}_c \oplus d^*$ {Append selected point to sequence}
15:     $\mathcal{D}_c^{(\text{new})} \leftarrow \mathcal{D}_c^{(\text{new})} \setminus \{d^*\}$
16: **end while**
17: **Return** $\mathcal{D}_c$

---

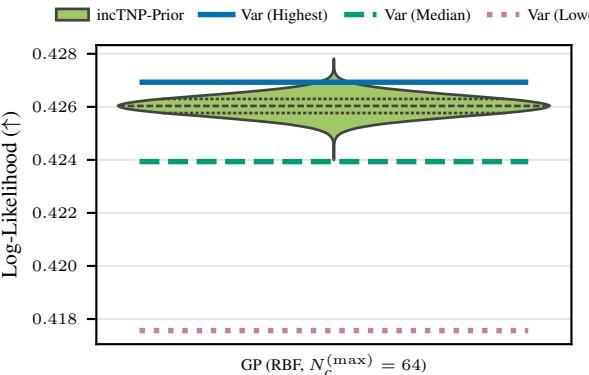

*Figure 26.* Distribution of test set log-likelihoods across random context permutations for an `incTNP-Seq` variant trained to condition on the null context set that we term `incTNP-Prior`. We measure performance over $1,000,000$ test set permutations for 1D GP regression using the RBF kernel and $N_c^{(\text{max})} = 64$. Our test set consists of 4096 samples. `incTNP-Seq`'s distribution density is shown using a violin plot, with internal dashed lines signifying distribution quartiles. We also benchmark `incTNP-Prior`'s modelling accuracy when using variance-based greedy contextual ordering as outlined in Algorithm 2. We consider all selection strategies $s \in \{\max, \text{median}, \min\}$, corresponding to Var (Highest), Var (Median) and Var (Lowest) respectively.

To best illustrate this ordering technique, we train a variant of `incTNP-Seq` that supports conditioning on the null set using a learned prior contextual token $\mathbf{R}^{(\text{prior})} \in \mathbb{R}^{D_z}$, which we term `incTNP-Prior`. We update Equation 18 and Equation 19 accordingly to:

$$\mathbf{Z}_l^{(c)} = \text{M-MHSA-layer}(\mathbf{Z}_{l-1}^{(c)}, M^{\text{causal},c}) \quad l \in 1, \cdots, L \quad \text{where} \quad \mathbf{Z}_0^{(c)} = \mathbf{R}^{(\text{prior})} \oplus \mathbf{D}_c^{(seq)} \tag{54}$$

$$\mathbf{Z}_l^{(t)} = \text{M-MHCA-layer}(\mathbf{Z}_{l-1}^{(t)}, \mathbf{Z}_l^{(c)}, M^{\text{causal},t}) \quad l \in 1, \cdots, L \quad \text{where} \quad \mathbf{Z}_0^{(t)} = (\mathbf{D}_t^{(seq)})_{1:N_c+N_t}, \tag{55}$$

where $\oplus$ denotes concatenation.

`incTNP-Prior`'s performance on greedily ordered context data is shown in Figure 26. Iteratively selecting context points with the highest predicted variance yields significantly improved performance over random context ordering, ranking in the top 1% of all random context set shuffles. Conversely, choosing context points with the lowest predicted variance significantly degrades the mean log-likelihood, with performance falling well below that seen across $1,000,000$ random test set orderings. Overall, greedily ordering the context set based on predicted variance is a principled task-independent approach that consistently extracts strong performance from incremental TNPs, exploiting their well-calibrated uncertainty estimates.

### H.4. Context Permutation Invariance through Aggregation

Having quantified the effect of sacrificing context permutation invariance for incremental TNPs, we now explore reintroducing this property approximately. As discussed in Appendix H.1, we can construct a context permutation invariant version $z$ of our model $p_\theta$ by constructing a Gaussian Mixture Model over all context orderings:

$$\frac{1}{N_c!} \sum_{\pi \in \Pi_{N_c}} p_\theta(\mathbf{Y}^t | \mathbf{X}^t, \mathcal{D}_\pi^c).$$

We can create an unbiased estimate of this permutation invariant construction using $K$ randomly drawn context orderings:

$$\frac{1}{K} \sum_{k=1}^{K} p_\theta(\mathbf{Y}^t | \mathbf{X}^t, \mathcal{D}_{\pi_k}^c) \quad \text{where} \quad \pi_k \sim \Pi_{N_c},$$

allowing for approximate context permutation invariance using a small, constant number of draws $K$. A similar technique is adopted by `TNP-A` to achieve approximate equivariance to the ordering of *target* datapoints.

*Table 8.* Average Test Log-Likelihoods ($\uparrow$) for factorised deployment. We report the absolute performance of the non incremental `TNP-D`. For `incTNP` and `incTNP-Seq` we report the difference ($\Delta$) relative to the reference performance when averaging model predictions over $K \in \{1, 2, 10, 100\}$ permutations of the context set. We also report the performance gain for $K = 100$ context permutations over a single permutation $K = 1$. Values indicate absolute or relative performance $\pm$ the standard error of the mean (SEM) of each method. We measure performance on 1D GP regression with an RBF kernel for both $N_c^{(\text{max})} = 64$ and $N_c^{(\text{max})} = 512$, as well as on the synthetic tabular dataset.

| | | REFERENCE | OURS ($\Delta$ ($\uparrow$) RELATIVE TO REF.) | | | | |
|---|---|---|---|---|---|---|---|
| DATASET | MODEL | TNP-D | $K = 1$ | $K = 2$ | $K = 10$ | $K = 100$ | GAIN ($1 \rightarrow 100$) |
| **1D GP (64)** | INCTNP | $0.4309 \pm 0.0070$ | $-0.0133 \pm 0.0070$ | $-0.0102 \pm 0.0070$ | $-0.0077 \pm 0.0070$ | $-0.0071 \pm 0.0069$ | $+0.0062$ |
| | INCTNP-SEQ | | $-0.0014 \pm 0.0070$ | $+0.0012 \pm 0.0069$ | $+0.0034 \pm 0.0069$ | $\mathbf{+0.0039 \pm 0.0069}$ | $+0.0053$ |
| **1D GP (512)** | INCTNP | $0.7930 \pm 0.0033$ | $-0.0037 \pm 0.0033$ | $-0.0028 \pm 0.0033$ | $-0.0022 \pm 0.0033$ | $-0.0020 \pm 0.0033$ | $+0.0017$ |
| | INCTNP-SEQ | | $+0.0027 \pm 0.0032$ | $+0.0033 \pm 0.0032$ | $+0.0038 \pm 0.0032$ | $\mathbf{+0.0039 \pm 0.0032}$ | $+0.0012$ |
| **TABULAR** | INCTNP | $0.1541 \pm 0.0054$ | $-0.0202 \pm 0.0054$ | $-0.0165 \pm 0.0054$ | $-0.0137 \pm 0.0053$ | $-0.0129 \pm 0.0053$ | $+0.0073$ |
| | INCTNP-SEQ | | $+0.0075 \pm 0.0054$ | $+0.0105 \pm 0.0054$ | $+0.0129 \pm 0.0053$ | $\mathbf{+0.0136 \pm 0.0053}$ | $+0.0060$ |

Table 8 showcases the log-likelihood performance improvements obtained for our incremental TNP models by averaging over $K \in \{1, 2, 10, 100\}$ permutations across GP and tabular regression tasks. Aggregation offers modest but consistent performance increases for `incTNP` and `incTNP-Seq`, with most of the improvement obtained after a small handful of permutations (at around $K = 10$). Thus aggregating predictions across context permutations offers approximate context permutation invariance and reliable performance boosts. This is particularly useful for `incTNP-Seq`, allowing it to be deployed as a standard Neural Process in a static context environment, offering performance benefits over `TNP-D`.

## I. Distribution Shift

In many real-world streaming tasks, the task distribution may evolve over time; such distribution shifts may occur subtly over a long period or happen suddenly. We investigate how `incTNP` models, as well as other NP baselines, handle this phenomenon within the context of GP regression. Concretely, we adapt a version of the change surface GP kernel proposed by Herlands et al. (2016) to smoothly blur between two GP kernels $k_1$ and $k_2$ over a number of context observations $t$:

$$k_{\text{cs}}\Big((x,t),(x',t')\Big) = \Big(1 - \omega(t)\Big)\Big(1 - \omega(t')\Big)k_1(x,x') + \omega(t)\omega(t')k_2(x,x'),$$

where $\omega(t)$ is the scaled and shifted sigmoid function:

$$\omega(t) = \sigma_{\text{sig}}\left(\frac{t - t_0}{\tau + \epsilon}\right).$$

and where $\sigma_{\text{sig}}(\cdot) := \frac{1}{1+e^{-\cdot}}$ denotes the standard sigmoid function.

The distribution shift is controlled by a central parameter $t_0$, dictating where both kernels are mixed equally, and a temperature $\tau$ controlling the transition rate between the kernels, with $\epsilon$ preventing zero division. For this work $k_1$ and $k_2$ are chosen from the same kernel family but have different randomly sampled hyperparameters, and we aggregate our results over 4,096 tasks. We consider performance for NP models trained up to $N_c^{(\text{max})} = 64$ context points on both the mixed and RBF kernels (using the training procedure from Appendix B.1). Crucially, these models were trained on stationary kernels where the kernel hyperparameters remain fixed throughout each task. *They have never been exposed to distribution shifts during training.*

Figure 27 illustrates the streaming performance of NP models as the distribution shifts between different kernels sampled randomly from the mixed kernel (using a moderate shift of $t_0 = 20, \tau = 10$). A primary finding is that, despite being trained exclusively on static tasks where the underlying distribution remains fixed, the TNP-based models (`TNP-D` and `incTNP` variants) successfully adapt with the non-stationary environment. These models recover and achieve strong predictive performance as more data is observed from the new regime $k_2$. This demonstrates the strong generalisation capabilities of these models because they have never been exposed to shifting distributions during training.

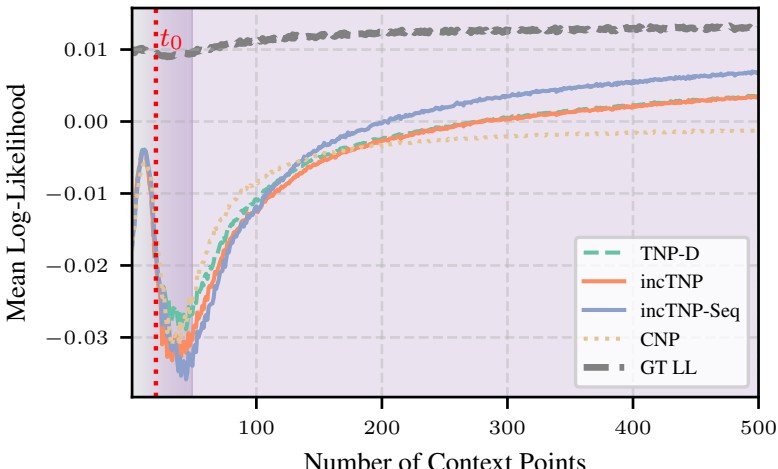

*Figure 27.* Test log-likelihood ($\uparrow$) on the distribution shift experiment (transitioning from kernel $k_1$ to $k_2$) using models trained on the mixed kernel GP task. The red line indicates the crossover point $t_0 = 20$, where both kernels $k_1$ and $k_2$ are mixed equally. The shaded purple region represents the transition from $k_1$ to $k_2$. Shaded region starts at 5% mixing interval (distribution is drawn mostly from $k_1$) and ends at 95% mixing interval (distribution is drawn mostly from $k_2$), using $\tau = 10$. The dotted grey line indicates the ground truth log-likelihood (GT LL). Results are averaged over 4,096 tasks.

Within the crossover region, all transformer-based NPs suffer a temporary drop in log-likelihood, with incremental TNPs exhibiting a slightly sharper decline as their fixed causal context history initially remains dominated by the previous regime. However, crucially, both `incTNP` and `incTNP-Seq` manage to recover as additional information arrives, with

`incTNP-Seq` ultimately outperforming `TNP-D` in the long run. This indicates that incremental TNPs can capably handle distribution shifts, holding significant potential for real-world online learning tasks.

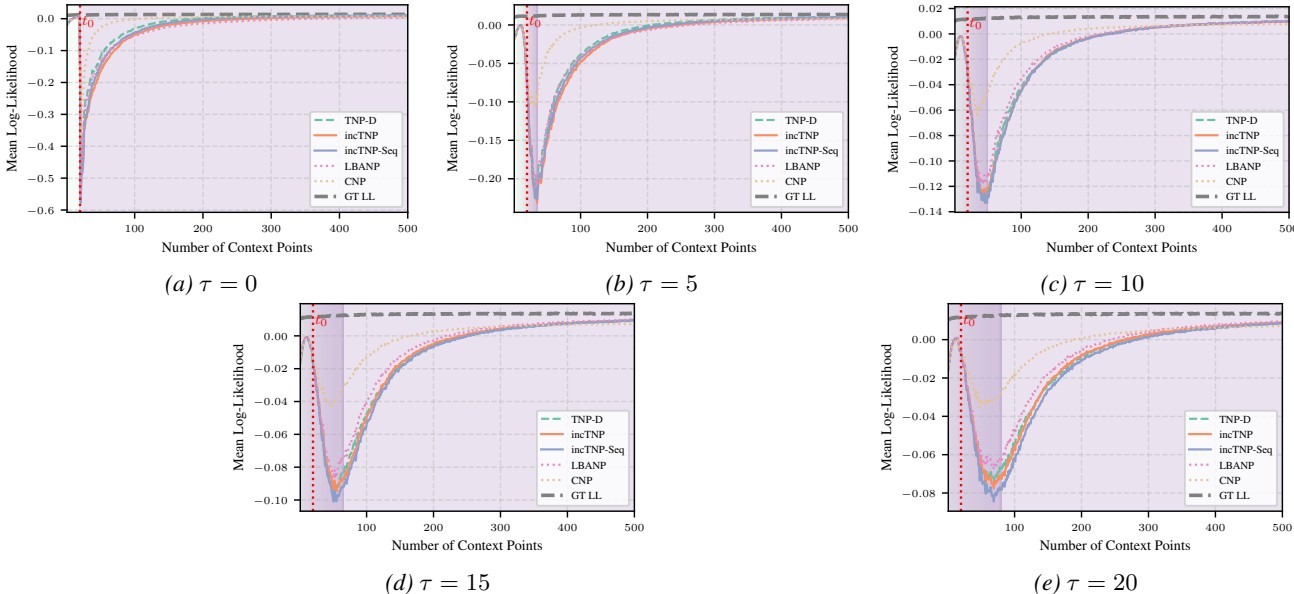

*Figure 28.* Test log-likelihood (↑) on the distribution shift task for the RBF kernel as the transition rate $\tau$ between kernels changes. Smaller $\tau$ values result in a much more abrupt distribution shift, whilst larger values lead to a smoother transition. $t_0 = 20$ is used throughout. Results are averaged over 4,096 tasks.

In Figure 28, we visualise the impact of the transition rate—controlled by $\tau$—on the RBF kernel, where $k_1$ and $k_2$ possess different randomly sampled lengthscales. We observe that all TNP-based models successfully recover strong predictive performance as context data from the new regime arrives, aligning with our findings from the mixed kernel. Whilst the `CNP` baseline adapts quickly, it remains a much weaker model due to its tendency to underfit. In contrast, incremental TNPs achieve predictive parity with, or outperform, the non-incremental `TNP-D`, fitting the data more closely as the stream progresses. As expected, smoother transitions (corresponding to larger $\tau$) result in less severe drops in log-likelihood. Notably, in these cases, the model starts to recover performance before fully transitioning into the second regime (i.e., prior to the 95% mixing threshold). This suggests that when the shift is sufficiently gradual, the model can successfully detect and adapt to the changing distribution dynamics on the fly.

This preliminary investigation demonstrates that `incTNP` and `incTNP-Seq` are capable of handling distribution shifts, rendering them promising candidates for streaming tasks in non-stationary environments too. Further work is required to quantify the rate of adaptation to such distribution shifts and how this depends on task difficulty, as well as their extrapolation behaviour as the number of context points is pushed even further beyond the maximum number of context points seen during training.

