# OpenReview forum: "Incremental Transformer Neural Processes"
_ICML.cc/2026/Conference — ICML 2026 regular_

### Official Review · Reviewer_TX2p · 2026-03-04

**Soundness:** 4
**Presentation:** 3
**Significance:** 3
**Originality:** 3
**Overall Recommendation:** 5
**Confidence:** 4

**Summary:**

This paper proposes a variant of Neural Processes that improves inference speed by relaxing the usual permutation-invariance assumption over the context points ordering. By relaxing this constraint, the proposed approach can leverage efficient incremental update techniques similar to those used in large language models, such as KV caching and causal generation.

The central claim of the paper is that this relaxation leads to a substantial improvement in inference speed while only minimally affecting predictive performance. Furthermore, the authors provide a theoretical argument based on the notion of implicit Bayesianism, suggesting that the resulting consistency property remains not too far that of the standard TNP.

**Compliance With Llm Reviewing Policy:**

Affirmed.

**Final Justification:**

Having looked at the other reviewers points and discussed with the authors, I am happy to keep my original score.

**Key Questions For Authors:**

- Have the authors considered combining incTNP’s incremental/KV-cache inference with constant-memory attention blocks [1] to address KV-cache memory growth for long streams?

- Could the authors report KL-gap under different context/stream ordering heuristics (e.g., random vs temporal vs learned/uncertainty-based), instead of only averaging over permutations, to show how KL-gap relates to order sensitivity and guide ordering choices in non-temporal tabular streams?

- Can the proposed incremental, relaxed-consistency approach be extended to row–column tokenized tabular models (e.g., TabICL/TabPFN-style architectures), and if so, could a similar relaxation be applied not only to row order (data points) but also, where appropriate, to column/feature order?

**Limitations:**

yes

**Strengths And Weaknesses:**

__Strength__:

Overall, I found the paper clear and compelling, with a novel incremental NP formulation that delivers substantial inference speedups while largely preserving predictive performance:

- Novel architecture with a well-articulated trade-off. The paper Introducing a causal structure to enable incremental updates is a bold departure from standard TNP design, and the paper is explicit about the key trade-off: improved streaming scalability at the cost of losing permutation invariance w.r.t. the context order.
- The theoretical framework used to justify the main claim is sound and useful for future related works. The discussion around exchangeability/non-exchangeability and the decomposition that motivates the KL gap provides a useful lens for analyzing “how far” the incremental prediction rule deviates from an exchangeable/Bayesian-like update rule.
- The experimental setups are good and very representative of the strength of the proposed method. Evaluation spans synthetic GP regression, tabular regression, and a real-world temperature task, and they test both factorised and AR deployment modes. The appendix discussion of TNP-A  is also a helpful addition for positioning.
- The dense autoregressive training setup is a nice design choice: by computing losses over many prefix contexts in a single forward pass.


__Weakness (Minor)__:

Eq. (3) in the main text updates the context via an MHCA form  $ Z_l^c = \mathrm{MHCA}\bigl(Z_{l-1}^c,\ \text{M-MHSA}(Z_{l-1}^c, M_{causal})\bigr),$ which seems inconsistent with the architectural description in the appendix where context is updated via masked self-attention only (Eq. 13 / Eq. 15). This looks like a notation/typing issue, but it made the model description harder to follow and should be clarified.

Additional limitation: The approach relies on KV caching; compute improves significantly, but cache memory could become a bottleneck for extremely long streams. A short memory discussion or scaling plot would strengthen the practicality story.

---

> ### Author Rebuttal · Authors · 2026-03-30
>
> We thank the reviewer for their highly constructive feedback and are pleased they found our architecture "novel" with a "well-articulated trade-off," and our theoretical contributions "useful for future related works".
>
> **Correction to Equation 3 (W1)** The reviewer is completely correct; this was a typo. We will update Eq. 3 in the main text to reflect the proper masked self-attention update: $Z_{l}^t = \text{MHCA}\left(Z_{l-1}^t, \text{M-MHSA}(Z_{l-1}^c, M^{\text{causal}})\right)$.
>
> **KV-Cache Memory Scaling & Constant-Memory Attention (W2 and Q1)** We completely agree that cache memory becomes the primary bottleneck for extremely long streams. As suggested, we will add an appendix section detailing the exact memory footprint scaling to transparently discuss this limitation. An example of memory plot we will provide is attached [here](https://anonymous.4open.science/r/ICML_Incremental_TNP-949E/Memory_Plot.png).
>
> Regarding constant-memory attention blocks, this is a fantastic suggestion and a natural next step for our framework. Having established that our causal masking strategy successfully preserves predictive performance and implicit Bayesianness, integrating constant-memory mechanisms is a natural direction for scaling incTNP to millions of data points without hitting hardware limits.
>
> **Context Ordering Heuristics (Q2)** This is another highly relevant future direction, particularly for batched streaming where we can control data ingestion order (linking closely to active learning). To test this, we conducted an experiment on the 1D-GP task using a greedy variance (uncertainty) criterion.
>
> Given an initial context, we iteratively evaluated the incremental model across all candidate locations and selected the next context point based on its predictive variance. We tested three strategies: max, median, and min variance. As shown in this [violin plot](https://anonymous.4open.science/r/ICML_Incremental_TNP-949E/Ordering_Strategies.png) (which compares these strategies against the distribution of test LLs from random permutations), the maximum variance strategy consistently leads to higher LLs, achieving performance in the upper percentiles of random permutations. Conversely, prioritising the lowest variance degrades performance.
>
> This clearly demonstrates that even simple active ordering heuristics can effectively guide streaming performance. We will include this analysis as an initial investigation into how incTNP can be used for active learning type of scenarios.
>
> **Compatibility with TabPFN and TabICL-style Architectures (Q3)** We thank the reviewer for this insightful direction. Given the recent rise in popularity of tabular foundation models, as well as the fact that one of their main bottlenecks is scaling, our technique is highly relevant, especially in streaming settings.
>
> Our incremental framework does indeed extend to state-of-the-art architectures used for tabular data prediction such as TabPFNv2 [1]. As noted by the reviewer, their architecture is based on factorised attention, separately over the rows (datapoints) and columns (features). Because this factorised attention decouples the two dimensions, we can maintain orthogonal KV caches:
> - Incremental samples (row updates): A new row queries the historical row-KV cache, followed by column-attention within that new row.
> - Incremental features (column updates): Sequentially arriving features query the historical column-KV cache, followed by row-attention.
>
> The extension to TabICL is not as straightforward because they do not use the factorised attention mechanism. The technique is still applicable over the row dimension, so we can still achieve a model that is incremental in datapoints. However, since the transformer that operates over the columns contains an induced self-attention block, KV caching becomes difficult to implement. This impedes incremental updates on the features.
>
> We will highlight this promising extension in the future work section.
>
> **References**
> 1. Hollmann, N. et al. (2025). Accurate predictions on small data with a tabular foundation model.

---

> > ### Author Rebuttal · Reviewer_TX2p · 2026-04-01
> >
> > Thanks for the clarifications, and I will keep my score unchanged

---

> > > ### Author Response · Authors · 2026-04-06
> > >
> > > We thank the reviewer for their time and feedback throughout the review process. We are very glad that our clarifications have fully addressed their concerns, and we deeply appreciate their support and positive recommendation for our work.

---

### Official Review · Reviewer_xrp7 · 2026-03-12

**Soundness:** 2
**Presentation:** 3
**Significance:** 2
**Originality:** 2
**Overall Recommendation:** 4
**Confidence:** 3

**Summary:**

This paper introduces Incremental Transformer Neural Processes (incTNP), a Neural Process variant designed for streaming data settings. The key contributions are: (1) incorporating causal masking and KV caching into Transformer NPs to enable linear-time context updates which reduces complexity from O(N²) to O(N); (2) proposing a dense autoregressive training strategy (incTNP-Seq) that treats data as a single sequence to improve data efficiency; (3)  analysis of implicit Bayesianness to quantify how causal masking affects probabilistic consistency.

**Compliance With Llm Reviewing Policy:**

Affirmed.

**Final Justification:**

These responses have effectively addressed the initial concerns raised earlier. As such, I will raise my score accordingly.

**Key Questions For Authors:**

1. incTNP-Seq is trained with dense autoregressive objective on full sequences but deployed incrementally. Is there evidence that this mismatch causes performance degradation?

2. Does causal masking affect different attention heads uniformly?

2. incTNP assumes data arrives strictly in order and is never corrected or re-ordered. In real-world streaming, data may arrive late, out-of-order, or require retroactive correction. How incTNP handles these scenarios?

**Limitations:**

1. The paper acknowledges that incTNP sacrifices permutation invariance (Section 4) but provides limited analysis of when this matters in practice.

2. The comparison between incTNP and incTNP-Seq confounds architectural differences with training strategy differences (dense autoregressive objective). It's unclear whether performance gains from incTNP-Seq stem from the architecture or the training method.

3. The paper claims "strong length generalisation" (Section 5) based on models trained on contexts up to 1024 points and evaluated on streams exceeding 30,000 points. However, only aggregate log-likelihood is reported

**Strengths And Weaknesses:**

The paper presents an adaptation of causal masking and KV caching—techniques well-established in LLMs—to the Neural Process framework for streaming applications, but it is more likely an engineering adaptation rather than a novel theoretical insight.  Section 4's theoretical framework on implicit Bayesianness also largely recapitulates existing work without extending the theory. It's better to strengthen the theoretical contribution by providing novel analysis specific to the NP context.

---

> ### Author Rebuttal · Authors · 2026-03-30
>
> We thank the reviewer for their feedback.
>
> **Theoretical Novelty and the KL-Gap (W1 and W2)** While we do leverage established sequence modelling techniques, we respectfully point to the consensus of other reviewers who recognise incTNP as a "novel architecture" (Reviewer TX2p) that is "clearly differentiated from prior works" (Reviewer iEHL) and supported by "strong empirical evidence" (Reviewer DrRu).
>
> Applying causal masking to NPs is not merely an engineering adaptation; it fundamentally challenges the core assumption of NPs: context permutation invariance. While prior works like [1] partially relax this (maintaining invariance within an initial context), we are the first to drop this constraint entirely and investigate the consequences in terms of implicit Bayesianness. Our core theoretical insight is that discarding permutation invariance does not make the prediction rule significantly less implicitly Bayesian than a standard TNP, yet it surprisingly yields better performance at a fraction of the inference cost.
>
> Furthermore, Section 4 does not simply recapitulate existing theory. Prior work [2] does not address the NP setting at all, nor do they use our metric. They measure implicit Bayesianness via the variance of log-joint predictive distributions under random permutations. We introduce a novel metric: the KL gap between a non-exchangeable prediction rule and its exchangeable counterpart.
>
> **Q1 & L2: Training Mismatch and incTNP-Seq Ablation** The dense autoregressive (AR) training actually aligns the train and test time strategies, diminishing the mismatch. The loss factorises as $\sum_{t} \log P(x_t \mid x_{<t})$. This means that during a single training pass over a sequence of length $N$, the model is explicitly optimised on $N$ different incremental deployment scenarios simultaneously (predicting the next point given a prefix of length $1, 2, \dots, N-1$). Regarding an ablation (L2), as detailed to Reviewer DrRu, separating the AR objective from causal masking is mathematically impossible. Using unmasked context would cause future information leakage, breaking the AR objective. In terms of where the gains are coming from, we hypothesise the following: while causal masking imposes a minor penalty by restricting the context view, the dense AR training signal provides significant optimisation benefits, yielding a net positive in performance.
>
> Also note that, from an architectural perspective, the differences between TNP-D, incTNP and incTNP-Seq are minimal, mostly resulting from the different masking strategies. Otherwise, for fair comparisons we maintained consistent architectural choices.
>
> **Handling Late or Out-of-Order Data (Q3)** This scenario highlights a practical advantage of our KV caching strategy.
> - Retroactive corrections: If data at index $k$ requires correction, we do not need to recompute the entire sequence. We can truncate the KV cache at $k$ and resume the incremental forward pass. This avoids recomputing the first $k-1$ steps, making it strictly more efficient than standard TNPs, which must pay the full $\mathcal{O}(N^2)$ cost to re-encode the entire sequence.
> - Late appends: Because we encode the temporal information within the input features $x$, the NP naturally learns to handle out-of-order data appended to the context without requiring strict chronological sorting.
>
> **Impact on Attention Heads (Q2)** Mechanistic interpretability of attention heads remains largely unexplored in TNPs. However, drawing parallels to standard transformers, we hypothesise the causal mask induces task-dependent specialisation: some heads likely act as global aggregators (similar to standard CNPs), while others specialise in local correlations. This represents an interesting area for future work.
>
> **Practical Impact of Permutation Invariance (L1)** Sacrificing permutation invariance implies a departure from implicit Bayesianness, which matters in sequential decision-making (e.g., Bayesian Optimisation, Active Learning, RL). As [2] demonstrate, non-implicitly Bayesian rules produce incoherent updates, making agents mathematically exploitable in adversarial betting. Our KL gap provides a practical metric: between two similarly performing models, practitioners should select the one with the smaller KL gap to ensure more coherent updates.
>
> **Length Generalisation (L3)** We do not rely solely on aggregate metrics. Figures 4, 5 (left), 14 (left), and 15 explicitly plot the log-likelihood evolving continuously over context length. The fact that log-likelihood increases monotonically for up to 30k+ points—despite the model only seeing sequences of 1024 points during training—provides evidence of length generalisation.
>
> If these explanations address your concerns, we hope you reconsider your score.
>
> **References**
> 1. Hassan, C. et al. (2025). Efficient autoregressive inference for transformer probabilistic models.
> 2. Mlodozeniec B. et al. (2024). Implicitly Bayesian Prediction Rules in Deep Learning.

---

> > ### Author Rebuttal · Reviewer_xrp7 · 2026-04-05
> >
> > Thanks to the authors for their replies. These responses have effectively addressed the initial concerns raised earlier. As such, I will raise my score accordingly.

---

> > > ### Author Response · Authors · 2026-04-06
> > >
> > > We sincerely thank the reviewer for their constructive feedback and time. We are delighted that our responses have addressed their concerns, and we deeply appreciate their support and decision to raise their score.

---

### Official Review · Reviewer_DrRu · 2026-03-13

**Soundness:** 4
**Presentation:** 3
**Significance:** 3
**Originality:** 2
**Overall Recommendation:** 5
**Confidence:** 4

**Summary:**

The paper introduces 'Incremental Transformer Neural Processes (incTNP)' a novel NP-variant based on existing Transformer Neural Processes (TNPs). Current TNPs struggle with (real time) sequential data streams due to their quadratic computational
cost for updating the context set, as they have to recompute the internal context representations from scratch for every new observation. IncTNPs address this issue by leveraging techniques from Large Language Models (LLMs),
specifically causal masking and Key-Value (KV) caching, to enable efficient incremental updates with linear time complexity. Additionally, the paper also proposes an autoregressive training strategy (incTNP-Seq) that amortizes
learning across all context sizes in a single training step, empirically improving performance and hyperparameter robustness. The paper evaluates incTNP on various synthetic and real-world tasks.
The results indicate that incTNP achieves comparable or even superior predictive performance compared to standard TNPs, while significantly accelerating sequential inference resulting in orders of magnitude speedups.
Finally, the paper examines the impact of causal masking, which results in the loss of context permutation invariance, on the model's consistency. The results show that incTNP maintains  a level of "implicit Bayesianness" comparable to standard
non-causal TNPs.

**Compliance With Llm Reviewing Policy:**

Affirmed.

**Final Justification:**

After the discussion with the authors and after reading the other reviews, I decided to keep my score of "5: Accept" and strongly recommend acceptance of the paper.

**Key Questions For Authors:**

cf. weaknesses

**Limitations:**

yes

**Strengths And Weaknesses:**

**Strenghts**:

Clarity and Motivation: The paper is exceptionally well-written, clearly stating the motivation of inc-TNPs. The overall structure is logical and easy to follow.

Strong Empirical Evidence: The authors conduct extensive experiments on a diverse set of task, including both synthetic and real-world streaming data. The results consistently demonstrate significant speedups (orders of magnitude), while
maintaining similar or even slightly better predictive performance compared to standard TNPs.

Detailed Complexity Analysis: The paper provides a very detailed and clear analysis of how the modifications affect the computational complexity, transitioning from quadratic to linear time, especially for autoregressive (AR) deployment.
This theoretical analysis is backed by an empirical evaluation of the runtime scaling.

**Weaknesses**:

Ablation Study of incTNP-Seq: The paper introduces incTNP-Seq, a new autoregressive training strategy designed to reduce gradient variance and improve performance through higher data efficiency. While the results show significant improvements over incTNP
(especially on Protein and synthetic tabular datasets) and standard TNP, the paper would benefit from a more comprehensive ablation of this strategy. Specifically, it would be insightful to see the non-causal TNP with incTNP-Seq Training, i.e.
apply the incTNP-Seq training strategy to a normal TNP (i.e. causal mask only for cross-attention). This would help disentangle the benefits of the training strategy itself from the architectural changes (specifically causal masking for the context) of incTNP.

Interpretation of KL-Gap in "Implicit Bayesianness" Analysis: While the evaluation of "implicit Bayesianness" is appreciated, the interpretation of the KL-gap metric, especially their practical significance, could be further elucidated.
Why is the KL-gap considerably smaller for incTNP-Seq compared to incTNP? Does the dense autoregressive training inherently lead to a more "Bayesian" consistent model compensating for the loss of permutation invariance or is this related to the
slightly better predictive performance of incTNP-Seq.

---

> ### Author Rebuttal · Authors · 2026-03-30
>
> We are delighted to hear that the reviewer found the motivation behind incTNPs "clear", the empirical evidence "strong", the complexity analysis "detailed", and the overall paper "exceptionally well-written".
>
> **Ablation of incTNP-Seq (W1)** We thank the reviewer for this suggestion. While disentangling architectural changes from training strategies is conceptually insightful, the requested ablation (applying the dense AR training strategy to an unmasked, non-causal TNP) is not possible due to causality violations.
>
> In our dense AR implementation, we process a joint sequence (the union of all context and target points) through two parallel streams:
> - The Context Stream: Contains the true $\mathbf{y}$ values, appended with a binary flag signifying it as context.
> - The Target Stream: Contains the exact same sequence, but the $\mathbf{y}$ values are masked (set to 0), appended with a target flag.
>
> If we applied standard bidirectional (unmasked) self-attention to the Context Stream, a representation at step $i$ would absorb the true $\mathbf{y}$ value of a future point at step $j$. When the Target Stream subsequently cross-attends to the Context Stream to predict the target at step $j$, it would indirectly receive its own true target value. This future information leakage fundamentally breaks the autoregressive objective. Therefore, causal masking on the context is a strict structural prerequisite for our dense AR strategy. We will add a brief note to the methodology clarifying this dependency.
>
> **Interpretation of KL-Gap (W2)** Interpreting the raw numerical value of a KL-gap is indeed challenging in isolation, which is why we evaluated it in comparison to baselines.
>
> - Contextualizing the KL-Gap: An exact Gaussian Process (GP) has a KL-gap of zero but cannot scale to the 30,000+ streaming data points handled by our model. We therefore benchmark against WISKI, a scalable approximate GP whose approximation is designed for streaming settings. The approximations required to make WISKI tractable introduce deviations from a purely Bayesian treatment. Surprisingly, our results demonstrate that incTNP-Seq's learned prediction rule is actually more implicitly Bayesian (smaller KL-gap) than the approximate Bayesian baseline WISKI.
> - Difference between incTNP and incTNP-Seq: As shown in Figures 3 and 13 (in the GP task), the difference in KL-gap between the two variants is often within overlapping error bars.
>
>   To investigate further, we ran [an additional evaluation](https://anonymous.4open.science/r/ICML_Incremental_TNP-949E/KL_Gap_Tabular.png) on the tabular dataset, predicting 40 target points conditioned on an initial context of 20 points. In that case TNP-D and incTNP-Seq have similar performance, while incTNP shows a larger KL Gap. All TNP variants show a significantly smaller gap than WISKI. We hypothesise that incTNP-Seq's generally smaller KL Gap stems from its AR training strategy, which better aligns with streamed deployment. However, we acknowledge these dynamics are likely task-dependent and sensitive to setup parameters (e.g., initial context size). Having established the framework to evaluate this metric, we are excited to explore its practical implications across broader scenarios in future work.
> - Practical Significance: Sacrificing permutation invariance implies a departure from implicit Bayesianness. In practice, this matters most in sequential decision-making (e.g., Bayesian Optimisation, Active Learning, RL). As [1] demonstrate, non-implicitly Bayesian rules produce incoherent updates, making agents mathematically exploitable in adversarial betting settings. Therefore, the KL-gap serves as a crucial metric for practitioners: between two similarly performing models (e.g., similar log-likelihoods), one should select the model with the smaller KL-gap to ensure more rational, coherent updates under new evidence.
>
> **References**
> 1. Mlodozeniec B. et al. (2024). Implicitly Bayesian Prediction Rules in Deep Learning.

---

> > ### Author Rebuttal · Reviewer_DrRu · 2026-04-01
> >
> > I thank the authors for their comprehensive response to my review.
> >
> > Regarding W1: This concern is fully addressed. I understand why my suggested ablation is not feasible and I appreciate the clarification provided.
> >
> > Regarding W2: I acknowledge that a smaller KL-gap is generally preferable and I appreciate the additional evaluation as well as highlighting the practical significance in sequential decision making. I am still not sure whether the smaller KL gaps in the provided evaluations are resulting from a more consistent model or rather originates from a better predictive performance. I agree with the authors that a deeper evaluation is subject for future work.

---

> > > ### Author Response · Authors · 2026-04-06
> > >
> > > We sincerely thank the reviewer for their time and for engaging with us during the rebuttal. We are very glad to hear that our responses have adequately addressed their concerns.
> > >
> > > Regarding W2, we appreciate the reviewer's insight. In this paper, our primary aim was to lay out the theoretical foundation for how one might investigate the consistency of these models (which is something that, as far as we are aware, hasn't been studied in NPs before). We completely agree that disentangling whether the smaller KL gaps originate from a more consistent model or simply from better overall predictive performance is an interesting question. We are very much looking forward to exploring this across multiple applications in our future work.

---

### Official Review · Reviewer_iEHL · 2026-03-13

**Soundness:** 3
**Presentation:** 3
**Significance:** 3
**Originality:** 3
**Overall Recommendation:** 4
**Confidence:** 4

**Summary:**

This paper introduces `incTNP`, a Transformer Neural Process (TNP) variant for streaming settings. By applying causal masking and KV caching, it reduces the per-update context cost from $\\mathcal{O}(N_c^2)$ to $\\mathcal{O}(N_c)$. A companion dense autoregressive training strategy (`incTNP-Seq`) supervises all prefix contexts in a single forward pass, improving both performance and hyperparameter robustness. The paper further adapts the implicit Bayesianness framework of Mlodozeniec et al. (2024) [3] to show that `incTNP`'s causal structure does not noticeably degrade probabilistic consistency. Experiments on 1D GP regression, tabular data, and spatiotemporal temperature prediction show `incTNP-Seq` matching or exceeding `TNP-D` under equal compute budgets while delivering up to $3.5\\times$ inference speedups in AR mode.

**Compliance With Llm Reviewing Policy:**

Affirmed.

**Final Justification:**

**Soundness.** The architecture is technically sound, and the near-parity with bidirectional TNP-D despite causal masking is a surprising finding. My primary concern (W2) — that the proof of Proposition 4.1 was invalid — was resolved.

**Originality.** Causal masking for TNPs was explored in TNP-A (Nguyen et al., 2022); the contribution here is extending it to context tokens to enable KV caching. Both KV caching and the dense training strategy (incTNP-Seq) are novel engineering adoptions from the LLM literature to NPs — cleanly executed and well-motivated.

**Significance.** The streaming capability and speedups address a genuine bottleneck. The implicit Bayesianness analysis provides a useful theoretical lens, though not fully substantiated experimentally.

**Clarity.** Well-written with a solid, transparent experimental scope. Presentation concerns (W1, W3, W5–W8) were addressed satisfactorily in the rebuttal.

**How the rebuttal changed my evaluation.** The rebuttal resolved W2 and all other weaknesses, leading me to raise my score from 3 to 4. The main reason I am not scoring higher is that implicit Bayesianness is presented as a cornerstone contribution, yet the teacher-forced experimental protocol evaluates the KL gap at ground-truth conditioning — a zero-probability event in practice. This means the reported gaps are strictly limited to one-step-ahead predictions in streaming cases where ground-truth data keep arriving, and do not capture the compounding effects where consistency failures would manifest. The reported gap is necessary but not sufficient for the claim. I refer the AC to my detailed comment for the full discussion of this point and its implications for the length generalisation results.

**Key Questions For Authors:**

**Questions for the Authors**

1. **Target and context ordering.** AR prediction is order-sensitive (Bruinsma et al., 2023 [2]), yet the paper does not state what ordering is used for the $N_t$ target points during AR decoding in any experiment. In tabular streaming experiments, there is no natural context ordering, and the chosen ordering is also unspecified. Could the authors (i) report the orderings used, and (ii) provide at least a brief sensitivity analysis or ablation, since ordering is a potentially confounding variable?

2. **CNP outperforming `incTNP` on Temperature (Forecast).** In Table 1a, `CNP` ($+0.181$) and `LBANP` ($+0.268$) both outperform `incTNP` ($+0.030$) on Temperature (Forecast), directly contradicting the claim in Section 5.1 that `incTNP` significantly outperforms `CNP`. Could the authors (1) confirm whether these numbers are correct, and if so, (2) explain why incTNP underperforms comparatively simple baselines on this task, and (3) clarify whether the claim in Section 5.1 should be qualified or retracted?

3. **Proof of Proposition 4.1 and exchangeability mismatch (W2).** The proof of Proposition 4.1 fails at two independent steps under any realistic assumption on \\(p\\):

- At step (28)\\(\\to\\)(29), cancelling the permutation inside \\(p\\) requires output-only exchangeability, i.e. \\(p(y_{\\pi(1)}, \\ldots, y_{\\pi(n)} \\mid x_1, \\ldots, x_n) = p(y_1, \\ldots, y_n \\mid x_1, \\ldots, x_n)\\). A GP with a non-trivial kernel does not satisfy this.

- At step (29)\\(\\to\\)(30), pushing \\(\\frac{1}{n!}\\sum_{\\pi \\in \\Pi_n}\\) inside the integral requires the log-ratio \\(\\log \\frac{\\hat{q}\_{1:n}}{p}\\) to be independent of \\(\\pi\\). This again requires output-only exchangeability of \\(p\\). Assuming conditional exchangeability of \\(p\\) instead does not resolve this: conditional exchangeability only controls the joint under simultaneous permutation of input-output pairs, but the denominator here is \\(p(y_{\\pi(1)}, \\ldots, y_{\\pi(n)} \\mid x_1, \\ldots, x_n)\\) with input held fixed, and this still depends on \\(\\pi\\). The summation therefore cannot be factored out of the integral.

Could the authors:

- (i) identify which assumption in the proof they believe covers both steps  and provide a formal justification; or

- (ii) if they agree with this assessment, explain how they propose to repair the proposition — keeping in mind that restricting to output-only exchangeable \\(p\\) makes the result incompatible with any realistic process; rederiving under conditional exchangeability still does not resolve the failure at step (29)\\(\\to\\)(30); and reframing the KL gap as a heuristic requires substantially revising the paper's claimed theoretical contributions throughout?


4. **Number of seeds.** Are results from a single training run per model or averaged over multiple seeds? This should be explicitly stated in the experimental setup, as readers may otherwise assume — following convention in the NPs literature — that results are averaged over multiple runs.

**Limitations:**

The conclusion discusses distributional shift robustness, KV cache memory, and context-size scaling as open problems. However, two limitations are absent: (1) ordering sensitivity for non-temporal and AR settings (see Q1), and (2) the conceptual scope of the implicit Bayesianness analysis (W2--W3). Both should be addressed.

**Strengths And Weaknesses:**

**Strengths.**

- The problem is well-motivated. Streaming inference is a genuine bottleneck for TNPs and PFN-style models, and the proposed solution — applying causal masking and KV caching uniformly across the entire context stream — is clean, technically sound, and clearly differentiated from prior works.
- **The near-parity of `incTNP` with `TNP-D` is surprising and important.** With causal masking, each context token $i$ attends only to predecessors $j \\leq i$, strictly limiting its representation relative to bidirectional `TNP-D`. That this incurs only a small performance penalty raises a broader question about how much cross-token interaction is actually needed for effective in-context learning in NPs.
- **`incTNP-Seq`'s dense training strategy** demonstrably improves performance beyond what is explained by compute alone (Figure 3), is FlashAttention-compatible, and improves hyperparameter robustness. The observation that dense supervision over all prefix
contexts simultaneously yields more informative gradients is well-supported empirically.
- **The empirical scope is solid and commendably transparent.** Experiments span diverse domains with both factorised and AR deployment, both static and streaming evaluation, and wall-clock cost reported alongside predictive performance. The appendix provides full hyperparameter sweeps, optimiser settings, dataset splits, architecture specifications, and hardware configurations and code for all experiments is released, further supporting reproducibility and future research.

---

**Weaknesses.**

**W1 --- Notational inconsistency in Equation 3.** Eq. 3 writes $\mathbf{Z}\_{l}^c = \text{MHCA}(\mathbf{Z}\_{l-1}^c ,\text{M-MHSA}(\mathbf{Z}\_{l-1}^c, M^{\text{causal}}))$ as the context update, but the outer MHCA is unmasked: token $i$ can attend to the M-MHSA output of future token $j > i$, undermining causality.

**W2 — The proof of Proposition 4.1 is incorrect, invalidating the paper's theoretical consistency analysis [Primary concern].** The theoretical contribution in Section 4.4 is not valid as presented. The critical step is the move from Equation (28) to Equation (29) in the proof, where the authors apply a change of variables \\((y_1, \\ldots, y_n) \\mapsto (y_{\\pi(1)}, \\ldots, y_{\\pi(n)})\\) and then cancel the permutation inside both \\(\\hat{q}_{1:n}\\) and \\(p\\) simultaneously. Concretely, this requires:

-  $$p(y_{\\pi(1)}, \\ldots, y_{\\pi(n)} \\mid x_1, \\ldots, x_n) = p(y_1, \\ldots, y_n \\mid x_1, \\ldots, x_n)$$

that is, \\(p\\) must satisfy **output-only exchangeability**: the joint distribution is invariant to permuting outputs at fixed input locations (Korshunova et al., 2018 [4]). However, a GP with a non-trivial kernel satisfies only **conditional exchangeability**:

- $$p(y_{\\pi(1)}, \\ldots, y_{\\pi(n)} \\mid x_{\\pi(1)}, \\ldots, x_{\\pi(n)}) = p(y_1, \\ldots, y_n \\mid x_1, \\ldots, x_n)$$

where input-output pairs are permuted together (Korshunova et al., 2020 [5]). These are distinct properties. A GP with a non-trivial kernel does not satisfy output-only exchangeability: the joint marginal \\(p(y_1, \\ldots, y_n \\mid x_1, \\ldots, x_n)\\) depends on the assignment of outputs to input locations through the kernel, and permuting only the outputs while holding inputs fixed changes this assignment. The proposition is therefore not proven for the GP experimental setting.

Even if the authors were to assume conditional exchangeability of \\(p\\) as a proposed fix to step (28)\\(\\to\\)(29)**, the proof would still
fail at step (29)\\(\\to\\)(30). At this step, one needs to push the sum \\(\\frac{1}{n!}\\sum_{\\pi \\in \\Pi_n}\\) inside the integral and recognise the result as \\(\\hat{q}_{1:n}\\). This factorisation requires the log-ratio

$$\\log \\frac{\\hat{q}\_{1:n}(y_{\\pi(1)}, \\ldots, y_{\\pi(n)})}{p(y_{\\pi(1)},
\\ldots, y_{\\pi(n)} \\mid x_1, \\ldots, x_n)}$$

to be independent of \\(\\pi\\), so that the sum acts only on \\(q_{1:n}\\) and can be recognised as \\(\\hat{q}\_{1:n}\\). While
\\(\\hat{q}\_{1:n}(y_{\\pi(1)}, \\ldots, y_{\\pi(n)}) = \\hat{q}\_{1:n}(y_1, \\ldots, y_n)\\) holds by its construction, the denominator \\(p(y_{\\pi(1)}, \\ldots, y_{\\pi(n)} \\mid x_1, \\ldots, x_n)\\) still depends on \\(\\pi\\) through the permuted outputs at fixed inputs, and this dependence is only removed under output-only exchangeability of \\(p\\) which, as noted above, holds only for degenerate processes.

The practical consequence is that the KL gap reported in Figures 3 and 13 currently has no valid theoretical interpretation as a measure of implicit Bayesianness. The quantity being computed is meaningful as an empirical diagnostic, but the paper has not established that it bounds or proxies the performance gap between \\(q_{1:n}\\) and a valid Bayesian posterior.

**W3 --- Imprecise framing of conditional inconsistency.**
Section 4.4 states that TNP-D "becomes conditionally inconsistent" in streaming settings, implying the deployment regime induces this property. In fact, conditional inconsistency is inherent to almost all C/NP variants, independent of deployment; streaming merely makes its consequences more visible and practically significant. This should be clarified to avoid misrepresenting the relationship between the model class and the deployment setting (see Bruinsma (2024), Section 2.3).

**W4 — Clarifications needed for the hyperparameter robustness analysis.** The variance reduction visible in Figure 3 is suggestive, but three clarifications would strengthen the claim in Section 5.1:

- *Outlier filtering*: how are failed runs defined, and is the criterion applied uniformly across all models? Removing them risks underestimating variance for less-robust baselines, potentially weakening the robustness claim for incTNP-Seq.
- *Mechanism*: lower gradient variance is asserted but not measured. A smoother loss landscape or implicit regularisation from denser supervision are equally plausible alternatives and should be acknowledged.
- *Variance vs. peak performance*: the compute-matched comparison (stars in Figure 3) tests peak performance, not hyperparameter sensitivity. These are distinct claims and should not be conflated in the discussion.

**W5 — "Length generalisation" framing requires clarification.**
Scaling from \\(N_c = 1024\\) (training) to \\(N_s > 30{,}000\\) (evaluation) is a useful and impressive result, but "length generalisation" can carry different meanings in the community. In the NP context, one might interpret it as observing samples at progressively more time steps, eventually exceeding the maximum seen during training. The paper should clarify which sense of generalisation is claimed.

**W6 --- `TNP-A` should appear in the main AR table.**
`TNP-A` (Nguyen et al., 2022) is the closest existing baseline --- explicitly designed for AR target prediction with the same backbone as `TNP-D`. The appendix AR table shows `TNP-A` outperforming or matching `incTNP-Seq` on HadISD Interpolation and 1D GP, with `incTNP-Seq` leading on the forecasting task. Including TNP-A in Table 1b — with a note on its streaming cost — would give a more complete picture.

**W7 --- Presentation suggestion: brief exposition of KV caching.**
The paper's description of KV caching is concise but may leave readers unfamiliar with the LLM literature without a clear understanding of (a) why causal masking is a prerequisite for the optimisation to be valid, (b) how cached matrices are used when a new token arrives, and (c) why bidirectional attention precludes it. A brief exposition would improve accessibility.


**W8 --- Minor presentation issues.**
Two typos: "log-likelihod" (Figure 6 caption) and "training stategy" (Section 4.2 heading). Moving the definitions of context/target tokens and $M^{\\text{causal}}$ into the main text would make Section 3 self-contained and help readers less familiar with TNPs. The main 1D GP results could also benefit from stating the kernel and context size (RBF, $N_c^{(\\text{max})}=64$) rather than relying on the appendix.

---

**Note on Naming Convention (Non-Binding Suggestion)**

As a minor editorial suggestion (not affecting my score): Bruinsma (2024)[1] (Section 2.2) proposes a useful convention distinguishing model *classes* (lowercase, plural abbreviation, e.g., "conditional neural processes" / CNPs) from specific *instances* (capitalised, singular, e.g., "the Conditional Neural Process" / CNP). This paper capitalises class references throughout (e.g., "Conditional Neural Processes," "Transformer Neural Processes"), while correctly handling instances like "the Incremental Transformer Neural Process (`incTNP`)." Given how arbitrarily NP terminology is currently used across the literature, I encourage the authors---and the community more broadly---to adopt this convention to improve precision and consistency.

---

**References**

**[1]** Bruinsma, W. P. (2024). Convolutional Conditional Neural Processes. *arXiv preprint arXiv:2408.09583*.

**[2]** Bruinsma, W. P., Markou, S., Requeima, J., Foong, A. Y. K., Andersson, T. R., Vaughan, A., Buonomo, M., Hosking, J. S., & Turner, R. E. (2023). Autoregressive Conditional Neural Processes. *ICLR*.

**[3]** Mlodozeniec, B., Requeima, J., & Turner, R. E. (2024). Implicitly Bayesian Prediction Rules in Deep Kernel Learning. *NeurIPS*.

**[4]** Korshunova, I., Degrave, J., Huszár, F., Gal, Y., Gretton, A., & Dambre, J. (2018). BRUNO: A Deep Recurrent Model for Exchangeable Data. *NeurIPS*, 31.

**[5]** Korshunova, I., Gal, Y., Gretton, A., & Dambre, J. (2020). Conditional BRUNO: A Neural Process for Exchangeable Labelled Data. *Neurocomputing*, 416, 305--309.

---

> ### Author Rebuttal · Authors · 2026-03-30
>
> We thank the reviewer for finding our method "clean, technically sound", "clearly differentiated from prior works", and our empirical scope "solid and commendably transparent".
>
> **W1, W6, W7, W8** We will correct Eq. 3 to $Z_{l}^t = \text{MHCA}\left(Z_{l-1}^t, \text{M-MHSA}(Z_{l-1}^c, M^{\text{causal}})\right)$, fix typos, and add the TNP-A results in Table 1b. We will explain KV caching, its need for causal (not bidirectional) attention, and add a section on its memory footprint.
>
> **Proof of Prop 4.1 & Exchangeability (W2 & Q3)** The proof is indeed for a general ‘unsupervised’ setting $(z_1, \dots, z_n)$. You're right that it's too large of an implicit leap we took in presenting the adaptation to regression, and the $y_i$ notation in the appendix that was intended to represent target-input pairs $(x_i, y_i)$ is misleading. We'll correct this, with a summary of how the gap is adapted below:
>
> Defining $z_i=(x_i, y_i)$, the gap _can_ be directly applied to the regression setting in the case of _i.i.d._ inputs $p(x_1, \dots, x_n)$ and conditionally exchangeable $p(y_1, \dots|x_1, \dots)$. Namely, the exchangeability gap directly applies to $q(z_1, \dots, z_n) := q(y_1,\dots,y_n|x_1, \dots, x_n)p(x_1, \dots, x_n)$, with $q(z_1, \dots, z_n)$ exchangeable _if and only if_ $q(y_1,\dots,y_n|x_1, \dots, x_n)$ is conditionally exchangeable (assuming $p$ has full support). One can easily verify that $\hat{q}(z_1, \dots, z_n)$ is exchangeable, and so $\hat{q}(y_1, \dots, y_n|x_1, \dots, x_n)$ is conditionally exchangeable, and will be equal to $q$ if and only if $q$ is already conditionally exchangeable. Lastly, the gap can be rewritten as:
> $$KL[q(z_1, \dots, z_n) || \hat{q}(z_1, \dots, z_n)] =\mathbb{E}_{p(x_1, \dots, x_n)}\left[ KL[ q(y_1, \dots, y_n|x_1,\dots, x_n)||\hat{q}(y_1, \dots, y_n|x_1,\dots, x_n) ] \right],$$
> where the latter is precisely what we report in the experiments. We always evaluate on iid input data.
>
> Separately, a similar (but different) gap can be derived for the case of non-_i.i.d._ input distribution $p(x)$, with the exchangeabilified distribution $\hat{q}(y|x)$ taking a different form (that doesn't depend on the true $p(y|x)$). We are happy to include this if it is of interest, but did not present it here due to space constraints.
>
> **Conditional Inconsistency Framing (W3)** We will clarify that deployment mode doesn't introduce inconsistency; rather, in settings like streaming data, TNP-D's conditional inconsistency manifests explicitly.
>
> **Robustness Analysis (W4)** We excluded the largest learning rate because TNP-D/incTNP failed to converge without a LR scheduler. Further filtering relied on Seaborn's default functionality. Unfiltered plots are [here](https://anonymous.4open.science/r/ICML_Incremental_TNP-949E/Boxplot_hyperparameters_no_outliers.png). Regarding gradient variance, [gradient noise-to-signal ratio plots](https://anonymous.4open.science/r/ICML_Incremental_TNP-949E/GNSR_RBF.png) support our hypothesis, though we value the reviewer's alternative explanations and will include them.
>
> **Length Generalisation (W5)** Models trained on ≤ 1024 points successfully predicted >30k points at inference without OOD degradation (LL increased monotonically). We will clarify this.
>
> **Q1**
> - Target orderings: We randomly order targets. We also provide the [standard deviation of the LL](https://anonymous.4open.science/r/ICML_Incremental_TNP-949E/Target_Ordering.png) - incTNP-Seq shows similar sensitivity to target ordering as TNP-D, while incTNP is slightly more sensitive.
> - Context orderings: As incTNP breaks context invariance, we evaluated a context KL gap to measure sensitivity to context permutations. [Results](https://anonymous.4open.science/r/ICML_Incremental_TNP-949E/Context_Ordering_RBF.png) show incTNP/incTNP-Seq exhibit a context KL Gap comparable to approximate Bayesian methods (WISKI). We randomly order tabular context. Also see Q2 from reviewer TX2p for ordering strategies.
>
> **Q2** We will clarify our best variant (incTNP-Seq) consistently outperforms CNP across all experiments, though incTNP underperforms on the zero-shot Forecast task.
>
> Predicting step 8 given the previous 7 frames introduces a distribution shift compared to standard NP training (randomly sampling contexts/targets across all 8 frames). This train-test mismatch, alongside highly correlated targets, hurts traditional factorised predictions. TNP-D also underperforms CNP here, indicating this stems from the standard NP training regime, not causal masking. CNP's better performance is likely due to underfitting.
>
> However, deploying models in AR mode at inference (which is the appropriate choice given the task), or using our densely AR-trained incTNP-Seq, recovers performance and significantly surpasses CNP. We will include this explanation in the updated manuscript.
>
> **Q4**
> Results reflect a single training seed; we will clarify this.
>
> If our updates address your concerns, we would appreciate a score reconsideration.

---

> > ### Author Rebuttal · Reviewer_iEHL · 2026-04-03
> >
> > I thank the authors for their thorough and constructive responses across all raised concerns. The clarifications on W1, W3–W8, and Q1–Q2, Q4 are well taken and, if incorporated into the revision as described, would meaningfully improve the paper.
> >
> > Regarding **W2**, I appreciate the authors' acknowledgement of the implicit leap in the original proof and their willingness to correct it. The clarification that the proof operates on joint objects $z_i = (x_i, y_i)$ and that the notation was intended to permute input-output pairs jointly is helpful — thank you for engaging with this carefully. That said, I think it would be valuable to see the corrected argument written out precisely. To that end, I would be grateful if the authors could provide the following:
> >
> > 1. **The precise definition of $\hat{q}$**: Could you write out the exchangeabilised prediction rule $\hat{q}(z_1, \ldots, z_n)$ explicitly (the analogue of Equation 2 in the revised setting), clarifying how it relates to $q(y_1, \ldots, y_n \mid x_1, \ldots, x_n)$ under the i.i.d. input assumption?
> >
> > 2. **The revised statement of Proposition 4.1**: Could you state the corrected proposition with its assumptions made explicit? This would help make clear exactly what the proposition claims and under what conditions.
> >
> > 3. **A sketch proof**: A concise but rigorous sketch of the revised proof would be very helpful, making explicit where each symmetry property is invoked — specifically, (a) where conditional exchangeability of $p$ enters, (b) where the i.i.d. assumption on inputs is used, and (c) how the KL decomposes into the conditional form $\mathbb{E}_{p(\mathbf{x})}[\mathrm{KL}(q \| \hat{q})]$ reported in experiments.
> >
> > Given that Proposition 4.1 is an important part of the paper's contribution, having the corrected version made fully explicit before the discussion period concludes would go a long way toward resolving my remaining concern.

---

> > > ### Author Response · Authors · 2026-04-05
> > >
> > > We sincerely thank the reviewer again for their suggestions, and agree that they will lead to an improved manuscript. We address the remaining concern – how we will adapt the presentation of Proposition 4.1 – below:
> > >
> > > **Proposition 4.1 Adaptation**
> > >
> > > Proposition 4.1 would remain mostly unchanged, but we would change the notation from $y_i$ to $z_i$ to explicate that it does not refer to targets in a regression setting. We would clarify in the lead-up that it is stated for a general unsupervised setting.
> > >
> > > We would then add a Corollary that states the adaptation to the regression setting that reads as follows:
> > >
> > > > Corollary: Let $p_{Y_{1:n},X_{1:n}}:\mathcal{Y}^n \times \mathcal{X}^n \to \mathbb{R}$ be the ‘true’ data generating density, such that the distribution is conditionally exchangeable, i.e.:
> > > > $$p_{Y_{1:n}|X_{1:n}}(y_{1:n}|x_{1:n}) = p_{Y_{1:n}|X_{1:n}}(y_{\pi_{1:n}}|x_{\pi_{1:n}}) \quad \forall \pi, x_{1:n}, y_{1:n},$$
> > > > and the distribution over the inputs $p_{X_1, \dots, X_n}: \mathcal{X}^n \to \mathbb{R}$ is _i.i.d._.  For any conditional density $q_{Y_{1\colon n}|X_{1\colon n}}: \mathcal{Y}^n \times \mathcal{X}^n \to \mathbb{R}$, define the exchangeable-ified conditional density as:
> > > > $$q̂(y_{1:n}|x_{1:n}):= \frac{1}{n!}\sum_{\pi  \in \Pi} q_{Y_{1\colon n}|X_{1\colon n}}(y_{\pi_{1:n}}|x_{\pi_{1:n}}).$$
> > > > Then, the following decomposition holds:
> > > $$
> > > E_{X_{1:n} \sim p_{X_{1:n}}}[KL[q_{Y_{1:n}|X_{1:n}}(\cdot|X_{1:n} )\mid p_{Y_{1:n}|X_{1:n}}(\cdot|X_{1:n} )]] = E_{X_{1:n} \sim p_{X_{1:n}}}[KL[q̂_{Y_{1:n}|X_{1:n}}(\cdot|X_{1:n} )\mid p_{Y_{1:n}|X_{1:n}}(\cdot|X_{1:n} )]] + \underbrace{E_{X_{1:n} \sim p_{X_{1:n}}}[KL[q_{Y_{1:n}|X_{1:n}}(\cdot|X_{1:n} )\mid q̂_{Y_{1:n}|X_{1:n}}(\cdot|X_{1:n})]]}_{Gap}
> > > $$
> > >
> > > The above gap is the quantity we computed in the experiments.
> > >
> > > A summary of the proof is as follows:
> > >
> > > - We can note that when we consider the space $\mathcal{Z}=\mathcal{X}\times \mathcal{Y}$ of input-target pairs, the true joint distribution of input-target pairs in the corollary is exchangeable. Writing $z_i = (x_i, y_i)$ we have that $p(z_{1:n}) = p((x_1, y_1), \dots) = p(y_{1:n}|x_{1:n}) p(x_{1:n}) = p(y_{\pi_{1:n}}|x_{\pi_{1:n}}) p(x_{\pi_{1:n}}) = p(z_{\pi_{1:n}})$, where we use the conditional exchangeability of the targets and the *i.i.d.* assumption on the inputs.
> > >
> > > - We can define $q_{Z_{1:n}}(z_{1:n}):=q(y_{1:n} | x_{1:n}) p(x_{1:n})$.
> > >
> > > - Proposition 4.1 clearly applies to $q(z_{1:n})$ and $p(z_{1:n})$ above, since the latter is exchangeable. Hence, we have the gap:
> > > $$KL[q(z_{1:n}) || p(z_{1:n})] = KL[q̂(z_{1:n}) || p(z_{1:n})] + KL[q(z_{1:n}) || q̂(z_{1:n})]$$
> > >
> > > - Next, we expand the symmetrised joint distribution $q̂(z_{1:n})$ defined in Proposition 4.1. Because $p(x_{1:n})$ is *i.i.d.*, it is invariant to permutations ($p(x_{\pi_{1:n}}) = p(x_{1:n})$), allowing us to factor it out of the sum: $$q̂(z_{1:n}) = \frac{1}{n!}\sum_{\pi \in \Pi} q(y_{\pi_{1:n}} | x_{\pi_{1:n}}) p(x_{\pi_{1:n}}) = \left( \frac{1}{n!}\sum_{\pi \in \Pi} q(y_{\pi_{1:n}} | x_{\pi_{1:n}}) \right) p(x_{1:n}) = q̂(y_{1:n}|x_{1:n}) p(x_{1:n})$$
> > > - Finally, by the standard chain rule for KL divergence, taking the KL divergence between two joint distributions that share the exact same marginal distribution over $x_{1:n}$ simply yields the expected KL divergence of their conditionals. Applying this to all three terms in the gap equation above gives exactly the expected conditional decomposition stated in the Corollary.
> > >
> > > **Extension to non-iid inputs**
> > >
> > > We also want to point out that we can expand our analysis to non-iid inputs. We can include a section on this in the appendix or main body if the reviewer finds it a valuable addition. Defining the exchangeable-ified conditional distribution as
> > >
> > > $$
> > > q̂(y_{1:n}|x_{1:n}):= \frac{1}{n!}\sum_{\pi  \in \Pi} q_{Y_{1:n}|X_{1:n}}(y_{\pi(1:n)}|x_{\pi(1:n)}) \frac{p_{X_{1:n}}(x_{\pi(1:n)})}{p̂_{X_{1:n}}(x_{{1:n}})}
> > > $$
> > >
> > > where $\hat{p}(x_{1:n}):= \frac{1}{n!}\sum_{\pi \in \Pi} p_{X_{1:n}}(x_{\pi(1:n)})$, for any conditionally exchangeable $p(y_{1:n}|x_{1:n})$, the following decomposition holds:
> > >
> > > $$E_{x_{1:n}}[KL[q(y_{1:n}|x_{1:n})\mid p(y_{1:n}|x_{1:n})]] = E_{x_{1:n}}[KL[q̂(y_{1:n}|x_{1:n})\mid p(y_{1:n}|x_{1:n})]] + \underbrace{E_{x_{1:n}}[KL[q(y_{1:n}|x_{1:n})\mid q̂(y_{1:n}|x_{1:n})]]}_{Gap}$$
> > >
> > > Let us know if this describes how we would change Proposition 4.1 in sufficient detail, or whether can clarify anything else. We thank the reviewer once again for their time and input.

---

### Decision · Program_Chairs · 2026-04-30

**Decision:**

Accept (regular)

**Comment:**

This paper introduces Incremental Transformer Neural Processes (incTNP), adapting causal masking and KV caching to enable linear-time context updates for streaming data.
All reviewers recommend accepting the paper stating that it tackles a well-motivated bottleneck in Neural Processes with a clean, technically sound solution.
The reviewers praised the strong empirical evidence demonstrating orders-of-magnitude inference speedups without sacrificing predictive performance.
Additionally, the proposed dense autoregressive training strategy (incTNP-Seq) was highlighted as a highly effective contribution that improves both data efficiency and hyperparameter robustness.

During the review process, the primary point  centered on the theoretical claims regarding "implicit Bayesianness."
It was noted that the proof for Proposition 4.1 incorrectly relied on output-only exchangeability, which does not hold for non-trivial Gaussian Processes.
The authors engaged constructively during the rebuttal and discussion phases, successfully resolving this theoretical gap by deriving a revised Corollary tailored to the conditionally exchangeable regression setting.
The discussion also clarified that while the empirical KL gap computed under a teacher-forced protocol is a useful diagnostic of one-step ordering sensitivity, it does not fully establish the implicit Bayesianness of the model's full joint predictive distribution.

For the camera-ready version, the authors have committed to several important updates to reflect the discussion.
They promised to update Section 4 and Proposition 4.1 with the corrected Corollary and proof sketch, and to explicitly state the limitations of the teacher-forced KL gap metric, particularly regarding compounding errors and length generalization.
Furthermore, the authors agreed to fix the contextual update notation in Equation 3, include the TNP-A baseline results in the main tables, refine the text to clarify that conditional inconsistency is inherent to the model class rather than the deployment setting, and add a dedicated section detailing the KV cache memory footprint scaling.